# Rab4A-directed endosome traffic shapes pro-inflammatory mitochondrial metabolism in T cells via mitophagy, CD98 expression, and kynurenine-sensitive mTOR activation

Activation of the mechanistic target of rapamycin (mTOR) is a key metabolic checkpoint of pro-inflammatory T-cell development that contributes to the pathogenesis of autoimmune diseases, such as systemic lupus erythematosus (SLE), however, the underlying mechanisms remain poorly understood. Here, we identify a functional role for Rab4A-directed endosome traffic in CD98 receptor recycling, mTOR activation, and accumulation of mitochondria that connect metabolic pathways with immune cell lineage development and lupus pathogenesis. Based on integrated analyses of gene expression, receptor traffic, and stable isotope tracing of metabolic pathways, constitutively active Rab4A[Q72L] exerts cell type-specific control over metabolic networks, dominantly impacting CD98-dependent kynurenine production, mTOR activation, mitochondrial electron transport and flux through the tricarboxylic acid cycle and thus expands CD4[+] and CD3[+]CD4[−]CD8[−] double-negative T cells over CD8[+] T cells, enhancing B cell activation, plasma cell development, antinuclear and antiphospholipid autoantibody production, and glomerulonephritis in lupus-prone mice. Rab4A deletion in T cells and pharmacological mTOR blockade restrain CD98 expression, mitochondrial metabolism and lineage skewing and attenuate glomerulonephritis. This study identifies Rab4A-directed endosome traffic as a multilevel regulator of T cell lineage specification during lupus pathogenesis.

Systemic lupus erythematosus (SLE) is a potentially fatal autoimmune disease of unknown etiology[1]. Its pathogenesis has been characterized by the production of antinuclear autoantibodies (ANA) with both T cells[2] and B cells being essential for disease development[3,4]. Activation of the mechanistic target of rapamycin (mTOR) within T cells was earlier uncovered as a therapeutic target in patients with systemic lupus erythematosus (SLE)[5,6]. Subsequently, mTOR was identified as a central regulator of lineage development in the immune system[7,8]. mTOR is considered to be an integrator of genetic and environmental cues, which confer predisposition to SLE[9–11]. In turn, blockade of mTOR with rapamycin has shown preliminary evidence for remarkable therapeutic efficacy both in mice[12–14] and patients with SLE[5,15–17]. mTOR may be activated by oxidative stress[18,19] and localization to the lysosomal membranes where it senses amino acid sufficiency[20]. Skewed

✉ e-mail: perla@upstate.edu

tryptophan metabolism with characteristic accumulation of kynurenine (KYN) has been identified as the top metabolic biomarker[21,22] and predictor of therapeutic mTOR blockade in patient with SLE[21]. KYN triggered the activation of mTORC1 in primary T cells of patients[21] and mice with SLE[23,24]. Metabolome analyses demonstrated that lupus T cells processed tryptophan (TRP) differently, suggesting a contribution of T-cell intrinsic factors[24]. However, the precise mechanism of KYN-sensitive mTOR activation during T-cell development and lupus pathogenesis has been unknown.

mTOR has been localized to endosomes[25] along with traffic regulator small GTPases, Rab4A[6] and Rab5[26]. Both of these GTPases are overexpressed in T cells of patients[6] and mice with SLE[13]. Notably, the overexpression of Rab4A, but not Rab5, precedes mTOR activation, ANA production and disease onset in SLE[13]. Rab4A is encoded by the *HRES-1/Rab4* human endogenous retroviral element[27], which is centrally positioned within the 1q42 lupus susceptibility locus[28]. Endogenous retroviruses, such as *HRES-1/Rab4*, share regulatory DNA elements with exogenous viruses and serve as sensors of infections[27] and mediators of autoimmunity[29,30]. Polymorphic alleles of *HRES-1/Rab4* have been associated with autoantibody production and predisposition to autoimmune diseases, such as SLE[31,32] and multiple sclerosis (MS)[33,34]. *HRES-1* haplotypes influence autoantibody production and organ involvement, including glomerulonephritis (GN), in patients with SLE[32]. *HRES-1* polymorphisms influence the expression of *HRES-1/Rab4* or *Rab4A*[27,35], a small GTPase that regulates endosomal recycling of surface receptors and organelles, including mitochondria[36]. Rab4A restrains mitophagy and promotes the accumulation of oxidative-stress-generating mitochondria[13,37]. Further upstream, the transcription of *Rab4A* is controlled by redox-sensitive transcription factors, NRF1 and USF1[35]. Downstream, Rab4A itself regulates mTOR activation in Jurkat human leukemic T cells and primary human T lymphocytes in vitro[35]. However, the role of Rab4A-mediated endosome traffic beyond mTOR activation, T-cell lineage development and autoimmunity remain unknown.

Here, we show that constitutive activation of Rab4A in C57Bl/6 (B6) and lupus-prone B6 SLE1.2.3. triple-congenic (B6.TC) mice exert dominant control over pro-inflammatory signal transduction networks at multiple levels in vivo: i) Rab4A enhances mitochondrial metabolism by triggering the accumulation of mitochondria, mitochondrial hyperpolarization (MHP), increased mitochondrial ATP production and enhanced tricarboxylic acid (TCA) and pentose phosphate pathway (PPP) fluxing in CD4+ T cells; ii) Rab4A prominently accelerates the recycling and surface expression of CD98 that serves as receptor for branched chain and aromatic amino acids and pro-inflammatory metabolites, such as KYN. In turn, KYN may spread inflammation through the bloodstream by eliciting mTOR activation, the expansion of CD4+ T cells at the expense of CD8+ T cells, promotes CD3+CD4−CD8− double-negative (DN) T cell and B cell activation and plasma cell expansion; iii) Rab4A-driven mTOR activation promotes ANA production and GN, while the inactivation of Rab4A in T cells restrains CD98 expression, KYN accumulation, mTOR activation, mitochondrial metabolism and lineage skewing and blocks ANA and anti-phospholipid antibody (aPL) production and GN. Moreover, CD98 expression is elevated in T cells and it predicts therapeutic response to mTOR blockade within the context of a controlled clinical trial in patients with SLE[15]. These results establish Rab4A as a cell type-specific controller of mitochondrial metabolism and CD98-dependent and KYN-sensitive mTOR activation that mediate therapeutically targetable pro-inflammatory lineage specification in SLE.

## Results

### Activation of Rab4A promotes autoimmunity and glomerulonephritis in female lupus-prone B6.TC mice

Rab4A is overexpressed in T cells of SLE patients[6] and, prior to the onset of ANA production or any sign of disease, in lupus-prone mice[13].

To determine its causative involvement in disease pathogenesis, we replaced the wild-type *Rab4A* alleles in C57Bl/6 (B6) and lupus-prone B6 SLE123 triple congenic (B6.TC) mice with constitutively active *Rab4A*^Q72L alleles surrounded by loxP sites (B6/Rab4A^Q72L; Figures S1A, B, and C). *Rab4A* was selectively deleted in T cells of B6/Rab4A^Q72L−KO and B6.TC/Rab4A^Q72L−KO mice by crossing of B6/Rab4A^Q72L and B6.TC/Rab4A^Q72L strains with B6.CD4^Cre mice[38] (Figures S1D and E). Cre was not expressed in peripheral CD8+ T cells as the deletion of Rab4A had occurred at the developmental stage of double-positive T cells in the thymus, as originally described in this targeted deletion model[38]. The production of ANA marked the onset of autoimmunity in male and female B6.TC mice from 20-29 weeks of age (Fig. 1A). Constitutive Rab4A activation enhanced ANA production in both female and male B6.TC/Rab4A^Q72L mice over B6.TC controls matched for age and gender (Fig. 1A). Interestingly, ANA production was also increased B6/Rab4A^Q72L over B6 controls, suggesting that activation of Rab4A alone also triggers autoimmunity on the B6 background (Fig. 1A). Inactivation of Rab4A in T cells blocked ANA production in female but not male B6.TC/Rab4A^Q72L−KO mice over B6.TC/Rab4A^Q72L controls (Fig. 1A).

Elevated levels of antiphospholipid antibodies (aPL), such as anti-apolipoprotein H (anti-ApoH) or anti-β2 glycoprotein I (anti-β2GPI) (Fig. 1B) and anti-cardiolipin antibodies (ACLA) were also observed in male and female B6.TC mice over B6 controls (Fig. 1C). Male B6.TC mice had fold increases of 4.7, 2.3, and 2.5 in the production of ANA, anti-ApoH and ACLA, respectively, over B6 controls. ANA, anti-ApoH, and ACLA production were further elevated in B6.TC/Rab4A^Q72L mice by 26%, 38% and 42%, respectively, over B6.TC controls. Male B6.TC/Rab4A^Q72L-KO mice were protected from autoimmunity as ANA, anti-ApoH and ACLA production were not increased over B6/Rab4A^Q72L-KO controls (Fig. 1A–C). Female B6.TC mice also showed elevated production of ANA, anti-ApoH and ACLA, 1.9-fold ($p = 0.0251$), 2.1-fold ($p = 0.0141$) and 2.0-fold ($p = 0.0001$) over B6 controls, respectively (Fig. 1A–C). B6.TC/Rab4A^Q72L mice did not significantly deviate in autoantibody production from B6.TC controls. However, B6.TC/Rab4A^Q72L-KO mice produced significantly less ANA, anti-ApoH and ACLA relative to B6.TC/Rab4A^Q72 mice with drops of 33% ($p = 0.0008$), 31% ($p = 0.0441$) and 27% ($p = 0.0064$), respectively (Fig. 1A–C).

Proteinuria was increased in young adult male ($1.12 \pm 0.18$ μg/μl) over female B6 mice at 10–19 weeks of age ($0.65 \pm 0.96$ μg/μl; $p = 0.0395$). Proteinuria declined in males to 0.54 μg/μl by 50 weeks of age ($p = 0.0395$; Fig. S2A), which was consistent with earlier findings of sex and age-related differences[39–41]. Non-autoimmune B6 males also had greater proteinuria ($1.01 \pm 0.14$ μg/μl) than B6 female controls at 20–29 weeks of age ($0.49 \pm 0.07$ μg/μl, $p = 0.0299$; Fig. S2). 

With the onset of ANA at 20–29 weeks of age (Fig. S2B), there was increased proteinuria in B6.TC/Rab4A^Q72L females (1.11 μg/μl) relative to B6.TC females (0.58 μg/μl; $p = 0.0443$) (Fig. 1D). Inactivation of *Rab4A* in T cells abrogated proteinuria in female B6.TC/Rab4A^Q72L-KO mice ($0.56 \pm 0.11$ μg/μl) in comparison to B6.TC/Rab4A^Q72L female controls ($1.11 \pm 0.19$ μg/μl; Fig. 1D). Proteinuria was reduced in B6.TC/Rab4A ^Q72L-KO females relative to B6.TC/Rab4A^Q72L females with the development of GN at ages of 40–49 and >50 weeks (Fig. S2B).

Kidney tissues were scored for GN, glomerulosclerosis (GS), and % of glomeruli with sclerosis or hyalinosis at 50 weeks of age by an expert renal pathologist who was blinded to genotypes or therapeutic interventions, as earlier described[13]. Both female (Fig. 1E) and male lupus-prone B6.TC mice had significant increases in GN scores relative to B6 controls at 50 weeks of age (Fig. 1F). As expected[42], females had more severe GN scores ($1.31 \pm 0.23$) than male B6.TC mice ($0.61 \pm 0.19$; $p = 0.0282$). Importantly, female B6.TC/Rab4A^Q72L lupus-prone mice carrying constitutively active Rab4A^Q72L alleles had more severe GN than B6.TC lupus-prone mice with wild-type Rab4A alleles (Fig. 1E; $p = 0.0295$). By contrast, inactivation of Rab4A in T cells completely abrogated GN in B6.TC/Rab4A^Q72L-KO female mice relative to B6.TC/Rab4A^Q72L controls (Fig. 1E; $p < 0.0001$). GN scores of B6.TC/Rab4A^Q72L-

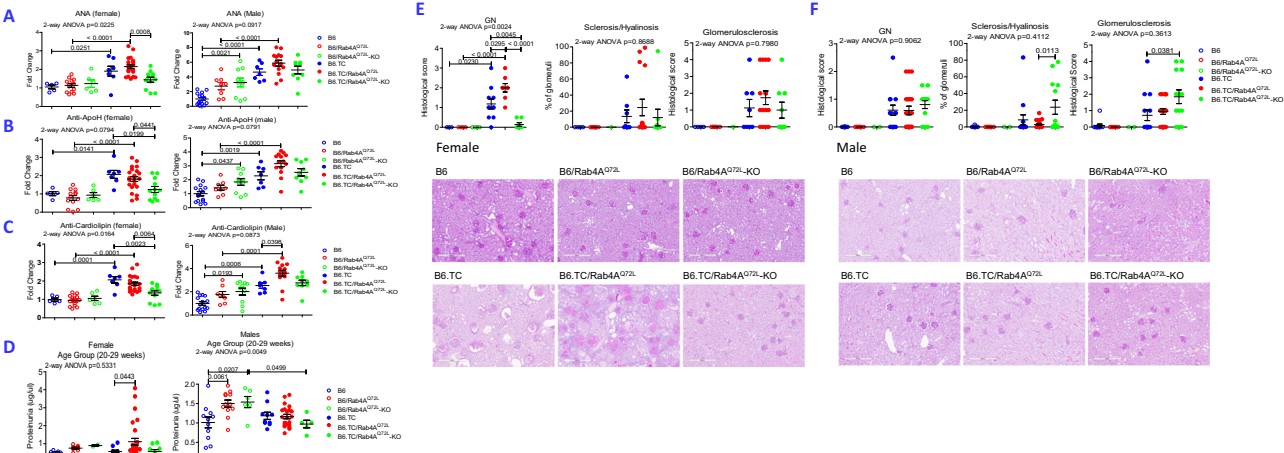

**Fig. 1 | Rab4A activation promotes ANA, aPL, proteinuria, and GN in female B6.TC mice.** Effect of Rab4A activation and T-cell-specific inactivation were examined on antinuclear autoantibody (ANA; **A**), β2-glycoprotein I or apolipoprotein H (Apo-H; **B**), and anti-cardiolipin autoantibody (ACLA; **C**) production and proteinuria (**D**) in age-matched, 20- to 29-week-old, male and female mice homozygous for constitutively active Rab4A^Q72L or lacking Rab4A in T cells Rab4A^Q72L-KO, respectively. **E** Effect of Rab4A on the development of GN, GS, and % hyalinosis in female mouse sets at ~50 weeks of age. Scale bars are embedded into each representative microscopic image. Dot plots present individual mice. **F** Effect of Rab4A

on the development of GN, GS, and % hyalinosis in male mouse sets at ~50 weeks of age. Kidneys were scored by an experienced renal pathologist blinded to mouse genotypes. Scale bars are embedded into each representative microscopic image. Dot plots present individual mice. 2-way ANOVA and Sidak's post-hoc test *p* values are displayed for multiple comparisons of Rab4 WT, Rab4A^Q72L, and Rab4A^Q72L-KO mice within B6 control and B6.TC SLE strains. Overall 2-way ANOVA *p* values are shown in the header of each figure panel, while Sidak's post-hoc test *p* values < 0.05 over brackets reflect comparison between experimental groups.

KO mice were also reduced relative to B6.TC controls (Fig. 1E; *p* = 0.0045). While Rab4A activation triggered autoimmunity both in females and males (Fig. 1A–C), it failed to influence proteinuria and GN in male B6.TC mice (Fig. 1F), which may be attributed to gender differences in end-organ resistance in SLE[43]. Interestingly, male B6.TC/Rab4A^Q72L-KO mice developed severe glomerulosclerosis with greater percentage of glomeruli with sclerosis or hyalinosis relative to B6.TC mice with normal Rab4A alleles (Fig. 1F). These findings are consistent with earlier observations that males have elevated glomerulosclerosis index, mean glomerular volume, and proteinuria (3.1-, 1.7-, and 1.8-fold, respectively) over age-matched females[40]. Importantly, estrogen regulates the biosynthesis of geranylgeranyl isoprene units which allow for posttranslational modification of Rab GTPases, such as Rab4[44]. While geranylgeranylation is required for binding of Rab4A to endosome membranes, pharmacological blockade of this enzymatic process inhibits the development of SLE in female mice[13]. Therefore, Rab4A-directed therapy may have different outcomes in males relative to females. SLE has a 9:1 increased prevalence in female over male patients with GN being a leading cause of mortality[45]. Given the overexpression of Rab4A in SLE patients, predisposition to GN in female B6.TC/Rab4A^Q72L mice, and blockade of GN upon inactivation of Rab4A in T cells, its role in immune system activation and disease pathogenesis were further investigated in female mice.

## Rab4A expands CD4+ T cells at the expense of CD8+ T cells both in B6 and B6.TC mice

In order to determine the mechanisms by which Rab4A promotes the immunopathogenesis of SLE, lymphocyte subsets were examined in female mice before the onset of autoantibody production and proteinuria at 20 weeks of age. T cells were overall expanded at the expense of B cells in the spleen of B6.TC/Rab4A^Q72L mice over B6/Rab4A^Q72L controls (2-way ANOVA *p* < 0.0001; Fig. 2A). Within T cells, Rab4A activation expanded CD4+ T cells but depleted CD8+ T cells both in B6/Rab4A^Q72L and B6.TC/Rab4A^Q72L mice over B6 and B6.TC controls, respectively (Fig. 2B). By contrast, CD4+ T cells were depleted in Rab4A^Q72L-KO mice with respect to Rab4A^Q72L parental controls both in the B6 and B6.TC strains, reversing the CD4:CD8 ratio back to baseline.

These findings suggest that the activation of Rab4A causes an expansion of CD4+ cells at the expense of CD8+ cells within the T cell compartment in both B6 and B6.TC mice. The inactivation of Rab4A in T cells of B6/Rab4A^Q72L–KO and B6.TC/Rab4A^Q72L–KO mice contracted the relative and absolute numbers of CD4+ T cells in comparison to B6/Rab4A^Q72L and B6.TC/Rab4A^Q72L mice, respectively (Figs. 2B, S3). In contrast, relative rather than absolute numbers of CD8+ T cells were expanded by the inactivation of Rab4A in T cells of B6.TC/Rab4A^Q72L–KO mice over B6.TC/Rab4A^Q72L controls (Figs. 2B, S3). These findings indicate that the skewing of CD4:CD8 T-cell abundance by Rab4A may be driven by the absolute depletion of CD4+ T cells rather than the expansion of CD8+ T cells in B6.TC/Rab4A^Q72L–KO mice over B6.TC and B6.TC/Rab4A^Q72L controls (Fig. S3).

In accordance with earlier findings that Rab4A limited mitophagy and promoted the accumulation of mitochondria in HeLa and Jurkat cells[13,37], the metabolic basis of lineage skewing was characterized by an increase of mitochondrial mass in CD4+ T cells of B6.TC/Rab4A^Q72L mice over B6.TC controls, which were reversed upon Rab4A deletion in B6.TC/Rab4A^Q72L-KO mice (Fig. 2C). Relative to the mitochondrial mass, the mitochondrial transmembrane potential (ΔΨm) was elevated in CD4+ T cells of B6.TC/Rab4A^Q72L mice, indicating MHP (Fig. 2C), which is a hallmark of mitochondrial dysfunction of T cells of patients with SLE[46]. Activation of Rab4A failed to augment mitochondrial mass or elicited MHP in CD8+ T cells of B6/Rab4A^Q72L mice (Fig. 2D).

Similar to SLE patients[13], the accumulation of mitochondrial mass was attributed to Rab4A-mediated Drp1 depletion in CD4+ T cells of B6/Rab4A^Q72L (Fig. S4A) and B6.TC/Rab4A^Q72L mice (Fig. S4B). Involvement of mTOR in a positive feedback loop was supported by the increased expression of Rab4A and the depletion of Drp1 and pDrp1^S616, which restrain mitochondrial fission and mitophagy[47,48] and thus elicit the accumulation of mitochondria, in mouse embryonic fibroblasts (MEFs) lacking a tuberous sclerosis complex 1 (TSC1-/-; Fig. S4C). Of note, patients with genetically enforced mTOR activation due to TSC mutations may develop SLE[49–51], including severe nephritis[52–55]. Alternatively, Drp1 and pDrp1 were accumulated and mitochondrial mass was reduced in MEFs lacking mLST8, a component shared by mTORC1

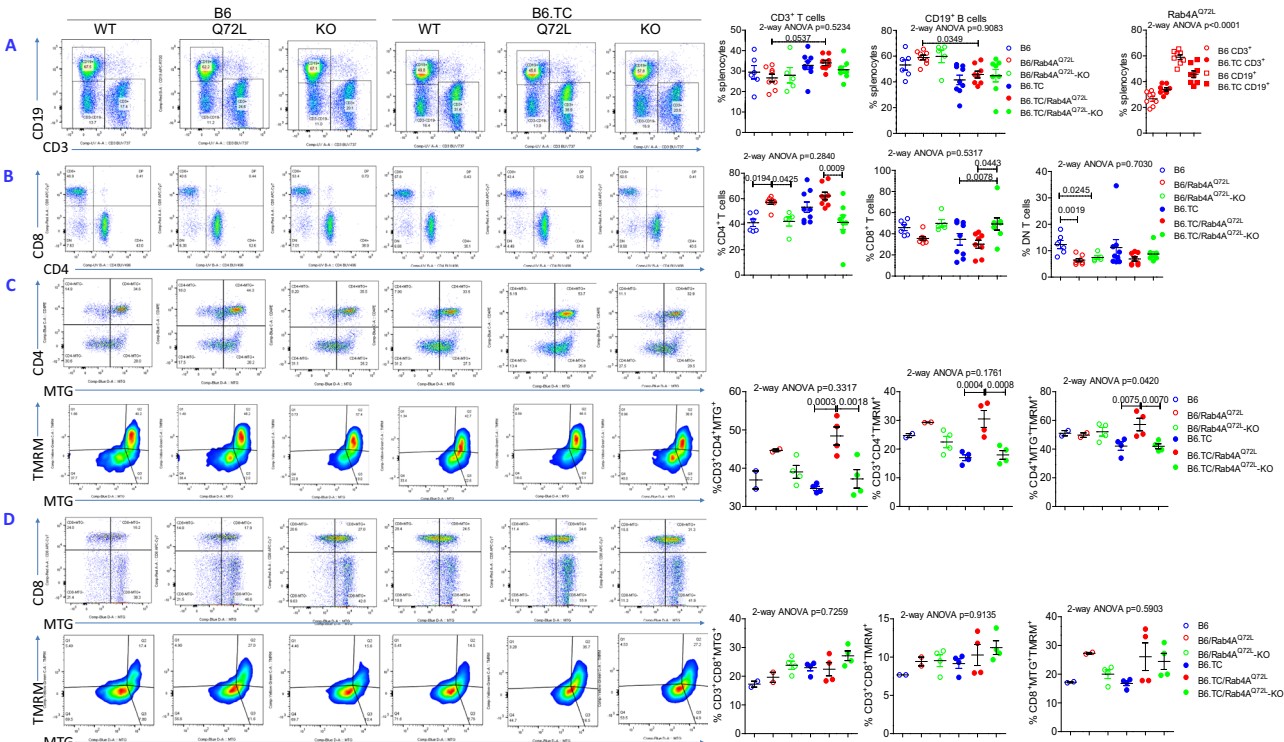

**Fig. 2 | Rab4A expands CD4⁺ T cells at the expense of CD8⁺ T cells both in B6 and B6.TC mice. A** Expansion of T cells at the expense of B cells in the spleens B6.TC/Rab4A^Q72L mice over B6/Rab4A^Q72L controls in 20-week-old female mice. The numbers (*n*) of mice in each experimental group were as follows: B6 (*n* = 6), B6/Rab4A^Q72L (*n* = 8), B6/Rab4A^Q72L-KO (*n* = 5), B6.TC (*n* = 9), B6.TC/Rab4A^Q72L (*n* = 8), B6.TC/Rab4A^Q72L-KO (*n* = 8); CD3⁺ T cells 2-way ANOVA *p* = 0.5234; Sidak's post-hoc test *p* values corrected for multiple comparisons: B6/Rab4A^Q72L vs B6.TC/Rab4A^Q72L *p* = 0.0537. CD19⁺ B cells 2-way ANOVA *p* = 0.9083; Sidak's post-hoc test *p* values corrected for multiple comparisons: B6/Rab4A^Q72L vs B6.TC/Rab4A^Q72L *p* = 0.0349. CD3⁺ vs CD19⁺ B cells in B6/Rab4A^Q72L vs B6.TC/Rab4A^Q72L mice comparison by 2-way ANOVA *p* < 0.0001. **B** Rab4A activation elicits the expansion of CD4⁺ T cells at the expense of CD8⁺ T cells within the T-cell compartment in B6/Rab4A^Q72L and B6.TC/Rab4A^Q72L mice. Left panel, representative flow cytometry dot plots; right panel, cumulative analysis of abundance of CD4⁺ and CD8⁺ T cells. The numbers (*n*) of mice in each experimental group were as follows: B6 (*n* = 6), B6/Rab4A^Q72L (*n* = 8), B6/Rab4A^Q72L-KO (*n* = 5), B6.TC (*n* = 9), B6.TC/Rab4A^Q72L (*n* = 8), B6.TC/Rab4A^Q72L-KO (*n* = 8); **C** Accumulation of mitochondrial mass and elevation of the

mitochondrial transmembrane potential (ΔΨm) in CD4⁺ T cells of B6/Rab4A^Q72L and B6.TC/Rab4A^Q72L mice and their reversal upon Rab4A deletion in B6/Rab4A^Q72L-KO and B6.TC/Rab4A^Q72L-KO mice. Relative to the mitochondrial mass, ΔΨm was elevated in CD4⁺ T cells of B6.TC/Rab4A^Q72L mice, indicating mitochondrial hyperpolarization (MHP). The numbers (*n*) of mice in each experimental group were as follows: B6 (*n* = 2), B6/Rab4A^Q72L (*n* = 2), B6/Rab4A^Q72L-KO (*n* = 4), B6.TC (*n* = 4), B6.TC/Rab4A^Q72L (*n* = 4), B6.TC/Rab4A^Q72L-KO (*n* = 4); **D** MHP in CD8⁺ T cells of B6/Rab4A^Q72L mice is reversed upon Rab4A deletion in B6/Rab4A^Q72L-KO mice. The numbers (*n*) of mice in each experimental group were as follows: B6 (*n* = 2), B6/Rab4A^Q72L (*n* = 2), B6/Rab4A^Q72L-KO (*n* = 4), B6.TC (*n* = 4), B6.TC/Rab4A^Q72L (*n* = 4), B6.TC/Rab4A^Q72L-KO (*n* = 4); Dot plots represent individual mice matched for genotype, age, and sex within each experimental group and processed in parallel. 2-way ANOVA *p* values are displayed for multiple comparisons of Rab4 WT, Rab4A^Q72L, and Rab4A^Q72L-KO mice within B6 control and B6.TC SLE strains. Overall 2-way ANOVA *p* values are shown in the header of each figure panel, while Tukey's post-hoc test *p* values < 0.05 over connecting bars reflect comparison between experimental groups.

and mTORC2 (Fig. S4D). Along these lines, Drp1 and pDrp1 were accumulated and mitochondrial mass was also reduced in MEFs lacking mTORC2 component Rictor and mTORC1 effector 4E-BP1 (Fig. S4D). These findings indicated the involvement of mTOR in Rab4A-driven accumulation of mitochondria.

### Rab4A activation induces splenomegaly, thrombocytopenia, and GN that are reversible by therapeutic mTOR blockade with rapamycin and N-acetylcysteine

In response to Rab4A-driven lupus pathogenesis, mTORC1 was consistently activated in CD3⁺, CD4⁺, and CD8⁺ T cells of 20-week-old B6.TC/Rab4A^Q72L mice relative to B6/Rab4A^Q72L controls (Fig. S5). mTORC1 activation of CD3⁺, CD4⁺, and CD8⁺ T cells of lupus-prone B6.TC/Rab4A^Q72L mice was reversed by the inactivation of Rab4A in T cells of B6.TC/Rab4A^Q72L-KO mice (Fig. S5). Since B6.TC/Rab4A^Q72L mice exhibited advanced GN at 50 weeks of age (Fig. 1E), we investigated whether disease pathogenesis could be blocked by 10-week treatment with rapamycin or N-acetylcysteine (NAC) from ~30 weeks of age. Similar to 50-week-old mice, GN scores were enhanced by the activation of Rab4A in a second set of B6.TC/Rab4A^Q72L mice over

B6.TC controls at 40 weeks of age (Fig. 3A). Inactivation of Rab4A in T cells of B6.TC/Rab4A^Q72L-KO mice repeatedly blocked GN as compared to B6.TC/Rab4A^Q72L mice (Fig. 3A). Enhanced GN of B6.TC/Rab4A^Q72L mice was characterized by inflammatory cellular infiltrates comprised of mTORC1⁺CD3⁺ T cells and B220⁺ B cells (Fig. 3B). Similar to patients with lupus nephritis (LN)[56], activation of mTORC1 was also noted in vascular endothelial cells (Fig. 3B). Deletion of Rab4A in T cells eliminated the infiltration by mTORC1⁺ T cells without affecting mTORC1 activity in blood vessels (Fig. 3B). However, similar to LN patients[56], rapamycin treatment blocked mTORC1 expression in vascular endothelial cells (Fig. 3B). NAC also reduced renal infiltration by T and B cells without affecting mTORC1 expression in vascular endothelial cells (Fig. 3B). In accordance with GN, IgG immune complex deposition was enhanced in B6.TC/Rab4A^Q72L mice and reversed in B6.TC/Rab4A^Q72L-KO mice (Fig. S6).

While total body and heart weights were unaffected, splenomegaly was markedly increased in B6.TC/Rab4A^Q72L mice over B6.TC controls at 40 weeks of age, which was completely reversed by the inactivation of Rab4A in T cells of B6.TC/Rab4A^Q72L-KO mice (Fig. 3C). B6.TC/Rab4A^Q72L mice had an average spleen weight of 1.19 g relative to

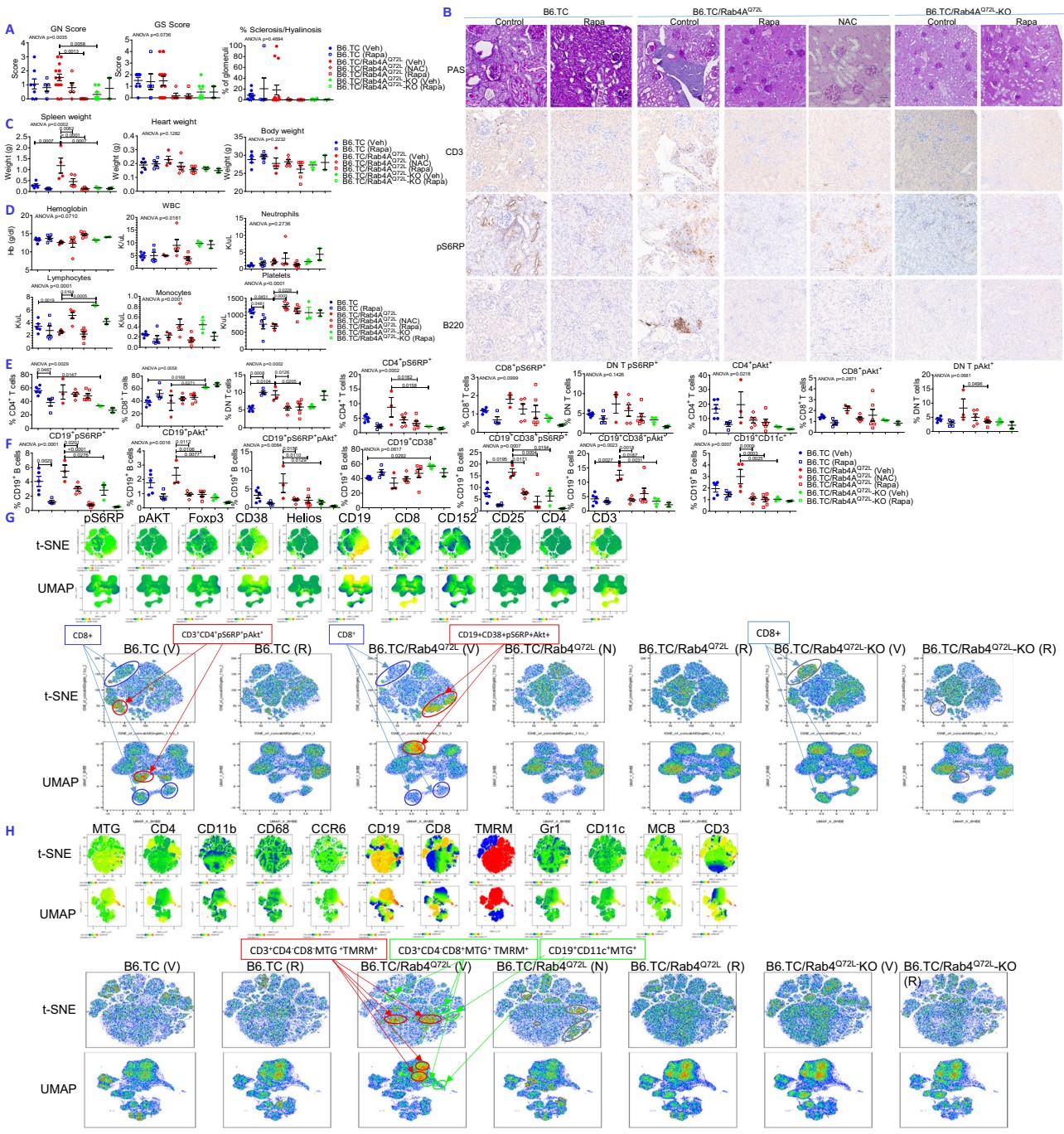

0.29 g in B6.TC mice ($p = 0.0007$) or 0.17 g in B6.TC/Rab4A$^{Q72L}$-KO mice ($p = 0.0007$). Rapamycin reduced spleen sizes from 1.19 to 0.13 g in B6.TC/Rab4A$^{Q72L}$ mice ($p < 0.0001$; Fig. 3C). NAC also abrogated the massive splenomegaly of B6.TC/Rab4A$^{Q72L}$ mice ($p = 0.0062$; Fig. 3C). Lymphocyte counts were elevated in B6.TC/Rab4A$^{Q72L}$-KO mice over B6.TC ($p = 0.0019$) and B6.TC/Rab4A$^{Q72L}$ controls ($p = 0.0005$; Fig. 3D). NAC also increased lymphocyte counts in B6.TC/Rab4A$^{Q72L}$ mice ($p = 0.0164$; Fig. 3D). Thrombocytopenia was noted in B6.TC/Rab4A$^{Q72L}$ mice over B6.TC controls ($p = 0.0461$) that was reversed by treatment with NAC ($p = 0.0003$) or rapamycin ($p = 0.0228$); Fig. 3D). Thrombocytopenia was not detected in B6.TC/Rab4A$^{Q72L}$-KO mice over B6.TC controls (Fig. 3D). Thus, the inactivation of Rab4A improved GN[l], lymphopenia, and thrombocytopenia, which are commonly found in patients and mice with SLE[57].

Similar to 20-week-old B6 and B6.TC females (Fig. 2B), CD4$^+$ T cells were depleted to 32.5% in B6.TC/Rab4A$^{Q72L}$-KO mice from 56.1%

in B6.TC mice, while CD8$^+$ T cells were expanded to 61.8% in B6.TC/Rab4A$^{Q72L}$-KO mice relative to 38.2% in B6.TC controls at 40 weeks of age as well (2-way ANOVA $p < 0.0001$; Fig. 3E). Thus, Rab4A exerted dominant control over the relative CD4:CD8 abundance independent of SLE or age. Moreover, DN T cells were markedly expanded in B6.TC/Rab4A$^{Q72L}$ mice over B6.TC controls ($p = 0.0104$) which were reversed by treatment with NAC ($p = 0.0126$) or rapamycin ($p = 0.0205$) but no longer detected in B6.TC/Rab4A$^{Q72L}$-KO mice (Fig. 3E). CD4$^-$CD8$^-$ DN thymocytes, preceding the later stages of CD4$^+$CD8$^+$ double-positive (DP) and CD4$^+$ or CD8$^+$ single-positive (SP) cells during T-cell development in the thymus, were depleted in B6/Rab4A$^{Q72L}$ females over B6 and B6/Rab4A$^{Q72L}$-KO mice but expanded in B6.TC/Rab4A$^{Q72L}$ females over B6.TC and B6.TC/Rab4A$^{Q72L}$-KO mice (Fig. S7). Intracellular staining demonstrated coordinate changes in mTORC1 and mTORC2, which underlay the skewed abundance of DN thymocytes (Fig. S8). These results suggest that Rab4A and lupus synergistically activate mTORC1

**Fig. 3 | Rab4A-mediated pro-inflammatory T and B cell development during lupus pathogenesis is responsive to treatment by rapamycin and N-acetylcysteine (NAC) in vivo.** Treatment with rapamycin (Rapa) and NAC was implemented in female B6.TC/Rab4A$^{Q72L}$ mice beginning at $27 \pm 1.4$ weeks of age. 3 mg/kg rapamycin was dissolved in phosphate-buffered saline (PBS) with 0.2% carboxymethylcellulose (CMC) solvent vehicle (Veh) and administered intraperitoneally (ip) three times weekly, while 10 g/l of NAC was provided in drinking water for 12 weeks. Control mice were treated ip three times weekly with 0.2% CMC solvent control alone. Age-matched female B6.TC and B6.TC/Rab4A$^{Q72L.CD4Cre}$-KO mice were also treated with rapamycin or solvent control. **A** Rapamycin, NAC, and inactivation of Rab4A block GN, GS, and glomerular hyalinosis. Kidneys were scored by an experienced renal pathologist blinded to mouse genotypes and treatments. Overall one-way ANOVA p values are shown in the header of each figure panel, while Sidak's post-hoc test p values < 0.05 over brackets reflect comparison between experimental groups. The numbers (n) of mice in each experimental group were as follows: B6.TC Veh (n = 8), B6.TC Rapa (n = 5), B6.TC/Rab4A$^{Q72L}$ Veh (n = 15), B6.TC/Rab4A$^{Q72L}$ NAC (n = 5), B6.TC/Rab4A$^{Q72L}$ Rapa (n = 6), B6.TC/Rab4A$^{Q72L}$-KO Veh (n = 8), B6.TC/Rab4A$^{Q72L}$-KO Rapa (n = 2); **B** Effect of rapamycin and inactivation of Rab4A on renal infiltration by CD3$^+$ T cells and B220$^+$ B cells and expression of pS6RP were assessed by immunohistochemistry. **C** Rapamycin and NAC, and inactivation of Rab4A block splenomegaly and cardiomegaly in lupus-prone mice. Overall one-way ANOVA p values are shown in the header of each figure panel, while Sidak's post-hoc test p values < 0.05 over brackets reflect comparison between experimental groups. The numbers (n) of mice in each experimental group were as follows: B6.TC (Veh) (n = 5), B6.TC Rapa (n = 5), B6.TC/Rab4A$^{Q72L}$ (Veh) (n = 4), B6.TC/Rab4A$^{Q72L}$ (NAC) (n = 5), B6.TC/Rab4A$^{Q72L}$ (Rapa) (n = 6), B6.TC/

Rab4A$^{Q72L}$-KO (Veh) (n = 3), B6.TC/Rab4A$^{Q72L}$-KO (Rapa) (n = 2). Spleen weights one-way ANOVA p = 0.0002, Sidak's post-hoc test p values corrected for multiple comparisons: B6.TC (Veh) vs B6.TC/Rab4A$^{Q72L}$ (Veh) p = 0.0007, B6.TC/Rab4A$^{Q72L}$ (Veh) vs B6.TC/Rab4A$^{Q72L}$-KO (Veh) p = 0.0007, B6.TC/Rab4A$^{Q72L}$ (Veh) vs B6.TC/Rab4A$^{Q72L}$ (NAC) p = 0.0062, B6.TC/Rab4A$^{Q72L}$ (Veh) vs B6.TC/Rab4A$^{Q72L}$ (Rapa) p < 0.0001; heart weights one-way ANOVA p = 0.1282, body weights one-way ANOVA p = 0.2232. **D** Effect of rapamycin and NAC and inactivation of Rab4A on anemia, leukopenia, and thrombocytopenia of lupus-prone mice. Overall one-way ANOVA p values are shown in the header of each figure panel, while Sidak's post-hoc test p values < 0.05 over brackets reflect comparison between experimental groups. The numbers (n) of mice in each experimental group were as follows: B6.TC Veh (n = 5), B6.TC Rapa (n = 5), B6.TC/Rab4A$^{Q72L}$ Veh (n = 3), B6.TC/Rab4A$^{Q72L}$ NAC (n = 5), B6.TC/Rab4A$^{Q72L}$ Rapa (n = 6), B6.TC/Rab4A$^{Q72L}$-KO Veh (n = 3), B6.TC/Rab4A$^{Q72L}$-KO Rapa (n = 2); **E** Effect of rapamycin and NAC, and inactivation of Rab4A on the abundance and mTOR activation of CD4$^+$, CD8$^+$, and DN T cells of lupus-prone mice. Overall one-way ANOVA p values are shown in the header of each figure panel, while Sidak's post-hoc test p values < 0.05 over brackets reflect comparison between experimental groups. The numbers (n) of mice in each experimental group were as follows: B6.TC Veh (n = 5), B6.TC Rapa (n = 4), B6.TC/Rab4A$^{Q72L}$ Veh (n = 3), B6.TC/Rab4A$^{Q72L}$ NAC (n = 5), B6.TC/Rab4A$^{Q72L}$ Rapa (n = 6), B6.TC/Rab4A$^{Q72L}$-KO Veh (n = 3), B6.TC/Rab4A$^{Q72L}$-KO Rapa (n = 2); **F** Effect of rapamycin and NAC, and inactivation of Rab4A on the activation of mTORC1 and mTORC2 in CD19$^+$ and CD19$^+$CD38$^+$ B cells and abundance of CD19$^+$CD11c$^+$ B cells. Overall one-way ANOVA p values are shown in the header of each figure panel, while Sidak's post-hoc test p values < 0.05 over brackets reflect comparison between experimental groups.

---

and mTORC2 and expand DN thymocytes in the thymus of B6.TC/Rab4A$^{Q72L}$ female mice (Figs. S7 and S87). DN T cells in the spleen and DN thymocytes of B6.TC/Rab4A$^{Q72L}$ females displayed markers of exhaustion; pAkt$^+$T-bet$^+$ splenocytes (Fig. S9A) and PD-1$^+$T-bet$^+$ and CTLA-4$^+$TIGIT$^+$ thymocytes were expanded in B6.TC/Rab4A$^{Q72L}$ female mice (Fig. S9), along with coordinated mTORC1 and mTORC2 activation and elevated expression of CD98 (Fig. S10). These findings indicate that the Rab4A/CD98/mTOR axis may cause exhaustion in the early DN stage of T-cell development within the thymus.

Surprisingly, rapamycin expanded DN T cells in B6.TC mice while it depleted DN T cells in B6.TC/Rab4A$^{Q72L}$ mice; NAC also depleted DN T cells in B6.TC/Rab4A$^{Q72L}$ mice (Fig. 3E). 10-week treatment with rapamycin in vivo most prominently blocked activation of mTORC1 in CD4$^+$ T cells of B6.TC/Rab4A$^{Q72L}$ mice (Fig. 3E). Inactivation of Rab4A also blocked mTORC1 activation in CD4$^+$ T cells of B6.TC/Rab4A$^{Q72L}$-KO mice as compared to B6.TC/Rab4A$^{Q72L}$ controls. These findings unveiled a tightly controlled positive feedback loop between Rab4A and mTOR in CD4$^+$ T cells.

Interestingly, mTORC1 and mTORC2 were also activated in CD19$^+$ and CD19$^+$CD38$^+$ B cells of B6.TC/Rab4A$^{Q72L}$ mice, and these changes were also reversed by the inactivation of Rab4A in T cells of B6.TC/Rab4A$^{Q72L}$-KO mice (Fig. 3F). Likewise, CD19$^+$CD11c$^+$ age-associated B cells (ABCs) were expanded in B6.TC/Rab4A$^{Q72L}$ mice, which were restrained in B6.TC/Rab4A$^{Q72L}$-KO mice (Fig. 3F). Rapamycin and NAC also reversed the activation of mTORC1 and mTORC2 and the expansion of ABCs in B6.TC/Rab4A$^{Q72L}$ mice (Fig. 3F).

Dimensionality reduction analyses by t-SNE and UMAP highlighted the expansion of mTORC1$^+$mTORC2$^+$CD4$^+$ T cells and mTORC1$^+$mTORC2$^+$CD19$^+$CD38$^+$ B cells and the depletion of CD8$^+$ T cells in B6.TC/Rab4A$^{Q72L}$ mice over B6.TC and B6.TC/Rab4A$^{Q72L}$-KO mice (Fig. 3G). Dependence of pro-inflammatory lineage skewing on Rab4A-mediated mTOR activation in B6.TC/Rab4A$^{Q72L}$ mice was substantiated by reversal via mTOR blockade with rapamycin or NAC or inactivation of Rab4A in B6.TC/Rab4A$^{Q72L}$-KO mice (Fig. 3G). Moreover, t-SNE and UMAP analyses highlighted the expansion of metabolically active MTG$^+$TMRM$^+$ DN T cells and MTG$^+$CD19$^+$CD11c$^+$ ABCs and the depletion of metabolically active MTG$^+$TMRM$^+$ CD8$^+$ T cells in B6.TC/Rab4A$^{Q72L}$ mice, all of which were reversed by the inactivation of Rab4A B6.TC/Rab4A$^{Q72L}$-KO mice or treatment with NAC or rapamycin

(Fig. 3H). As opposed to 20-week-old mice (Fig. 2C), MTG$^+$TMRM$^+$ CD4$^+$ T cells were not expanded in B6.TC/Rab4A$^{Q72L}$ mice over B6.TC controls following disease onset at 40 weeks of age (Fig. 3H).

## Rab4A distorts mitochondrial respiration and carbon flux through the TCA cycle in opposite directions between CD4$^+$ and CD8$^+$ T cells

Given the overall expansion of CD4$^+$ T cells and depletion of CD8$^+$ T cells by Rab4A activation in B6.TC/Rab4A$^{Q72L}$ mice, which were reversed by Rab4A inactivation in T cells of B6.TC/Rab4A$^{Q72L}$-KO mice, we examined the molecular bases of such skewing in lineage specification on the level of gene expression. RNA sequencing (RNAseq) unveiled strikingly discordant effects of Rab4A between CD4$^+$ T cells and CD8$^+$ T cells from B6.TC, B6.TC/Rab4A$^{Q72L}$, and B6.TC/Rab4A$^{Q72L}$-KO mice (Fig. S11A, S11B). Amongst the top 40 pathways affected by Rab4A at FDR p < 0.05 across all 12,777 pathways in the Partek database, 31 pathways involved the mitochondrial function-endosome-autophagy traffic cluster, while 9 pathways involved nucleotide biosynthesis and DNA-chromatin complex formation (Fig. S11C). Mitochondrial and endosome traffic and autophagy pathways were distorted by Rab4A into sharply opposite directions between CD4$^+$ and CD8$^+$ T cells (Fig. S11C). Functional studies using a Seahorse analyzer demonstrated markedly greater mitochondrial metabolism in CD4$^+$ T cells of B6.TC/Rab4A$^{Q72L}$ mice over those of B6.TC controls (ANOVA p = 0.0001). Rab4A activation increased basal respiration and mitochondrial ATP production in CD4$^+$ cells of B6.TC/Rab4A$^{Q72L}$ mice over B6.TC controls (Fig. 4A, B). Mitochondrial function was overall reduced in CD8$^+$ T cells upon activation of Rab4A (Fig. 4C, D). Rapamycin exerted opposite effects on basal respiration in CD4$^+$ relative to CD8$^+$ T cells. While rapamycin failed to impact the respiration of CD4$^+$ T cells (Fig. 4B), it increased basal respiration and mitochondrial ATP production in CD8$^+$ T cells of B6.TC/Rab4A$^{Q72L}$ mice (Fig. 4D). Upon rapamycin treatment in vivo, CD8$^+$ T cells of B6.TC/Rab4A$^{Q72L}$ mice exhibited greater basal respiration and mitochondrial ATP production over those of B6.TC controls (Fig. 4D). Thus, rapamycin increased ATP production in CD8$^+$ but not CD4$^+$ T cells of B6.TC/Rab4A$^{Q72L}$ mice (Fig. 4D), indicating that rapamycin selectively enhanced the mitochondrial metabolic fitness of CD8$^+$ T cells of lupus-prone B6.TC/Rab4A$^{Q72L}$ mice (Fig. 4D). Rab4A activation skewed basal respiration and mitochondrial ATP production into opposite directions

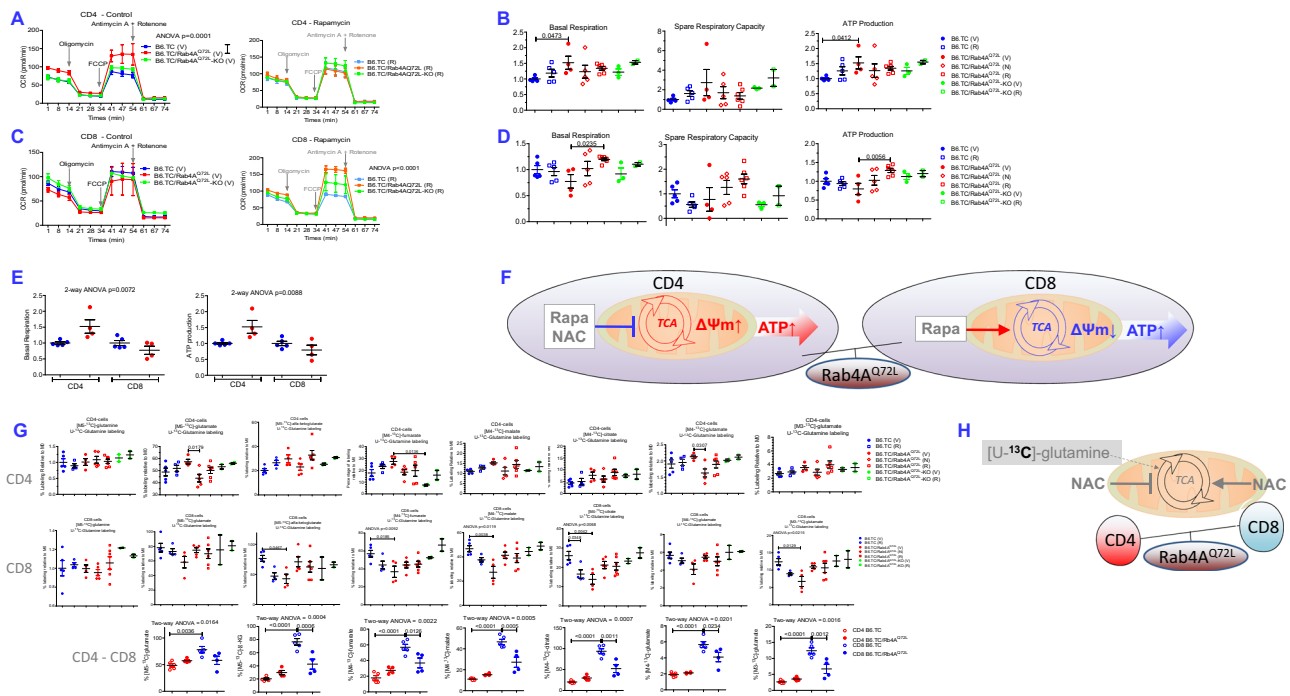

between CD4$^+$ and CD8$^+$ T cells in B6.TC/Rab4A$^{Q72L}$ mice as compared to B6.TC controls (Fig. 4E, F). By contrast, Rab4A deletion and rapamycin restored basal respiration and ATP production in CD8$^+$ T cells (Fig. 4D, F).

To delineate the metabolic bases of Rab4A-driven distortion of mitochondrial oxygen consumption rates (OCR) between CD4$^+$ and CD8$^+$ T cells, we traced the flux of [U-$^{13}$C]-glutamine using high-resolution mass spectrometry. Baseline metabolic flux through the TCA cycle was increased in CD8$^+$ T cells over CD4$^+$ T cells of B6.TC mice carrying wild-type (WT) Rab4A alleles (Fig. 4G). Enrichment of TCA substrates, [M5-$^{13}$C]-glutamate, [M5-$^{13}$C]-α-keto-glutarate (αKG), [M4-$^{13}$C]-fumarate, and [M4-$^{13}$C]-malate, was moderately accelerated in CD4$^+$ T cells of B6.TC/Rab4A$^{Q72L}$ mice over those of B6.TC controls. The inactivation of Rab4A blocked the accumulation of [M4-$^{13}$C]-fumarate in B6.TC/Rab4A$^{Q72L}$-KO mice relative to B6.TC/Rab4A$^{Q72L}$ controls (Fig. 4G). By contrast, metabolic flux through the TCA cycle was reduced in CD8$^+$ T cells of B6.TC/Rab4A$^{Q72L}$ mice over B6.TC controls, as indicated by the diminished enrichment of [M5-$^{13}$C]-αKG, [M5-$^{13}$C]-citrate, [M4-$^{13}$C]-fumarate, [M4-$^{13}$C]-malate, and [M3-$^{13}$C]-glutamate (Fig. 4G). Inactivation of Rab4A restored the TCA flux in CD8$^+$ T cells of B6.TC/Rab4A$^{Q72L}$-KO mice relative to B6.TC/Rab4A$^{Q72L}$ controls (Fig. 4G). Thus, the activation of Rab4A promoted metabolic flux through the TCA cycle and enhanced mitochondrial ATP production in CD4$^+$ T cells while it reduced metabolic flux through the TCA cycle and mitochondrial ATP production in CD8$^+$ T cells of B6.TC/Rab4A$^{Q72L}$ mice (Fig. 4E, G). Treatment with NAC blocked the accumulation of [M5-$^{13}$C]-glutamate and [M4-$^{13}$C]-glutamate in CD4$^+$ T cells of B6.TC/Rab4A$^{Q72L}$ mice (Fig. 4G).

### Rab4A enhances glucose flux and R5P production through the PPP in CD4$^+$ but not in CD8$^+$ T cells

Unlike mitochondrial respiration, glycolysis was unaffected by Rab4A activation alone in CD4$^+$ or CD8$^+$ T cells when measured with the Seahorse metabolic analyzer (Fig. S12A). However, in vivo treatment with rapamycin imposed markedly opposite effects on glycolysis of CD4$^+$ and CD8$^+$ T cells, which were influenced by Rab4A (Fig. S12A). Upon rapamycin treatment alone, both glycolysis and glycolytic capacity were enhanced in CD4$^+$ T cells but reduced in CD8$^+$ T cells of B6.TC mice (Fig. S12B). Rapamycin also increased glycolysis and

glycolytic capacity in CD4$^+$ T cells of B6.TC/Rab4A$^{Q72L}$-KO mice (Fig. S12A). Following rapamycin treatment in vivo, glycolysis and glycolytic capacity were enhanced in CD8$^+$ T cells of B6.TC/Rab4A$^{Q72L}$ mice relative to those from B6.TC controls (ANOVA $p = 0.0004$; Fig. S12A). Thus, Rab4A exerted cell type-specific changes in glycolysis between CD4$^+$ and CD8$^+$ T cells in the setting of mTOR blockade.

Metabolic flux through glycolysis was traced with [U-$^{13}$C]-labeled glucose. While the uptake of [U-$^{13}$C]-glucose was not affected by Rab4A in CD4$^+$ or CD8$^+$ T cells (Fig. S12C), [M1-$^{13}$C]-pyruvate was accumulated in CD4$^+$ T cells but not in CD8$^+$ T cells of B6.TC/Rab4A$^{Q72L}$-KO mice upon [1,2-$^{13}$C]-glucose labeling (Fig. S12C). This suggests that Rab4A regulates carbon flux between the non-oxidative branch of the PPP and glycolysis in CD4$^+$ but not in CD8$^+$ T cells. Upon [U-$^{13}$C]-glucose labeling, PPP substrates [M6-$^{13}$C]-glucose/fructose 6-phosphate (G/F6P) and [M6-$^{13}$C]-sedoheptulose 7-phosphate (S7P) and [M4-$^{13}$C]-S7P were accumulated in CD4$^+$ T cells of B6.TC/Rab4A$^{Q72L}$-KO mice over B6.TC controls (Fig. S12D). In contrast, the enrichment of [M4-$^{13}$C]-S7P and [M5-$^{13}$C]-ribose 5-phosphate (R5P) was enhanced by the activation of Rab4A in CD4$^+$ T cells of B6.TC/Rab4A$^{Q72L}$ mice over B6.TC and B6.TC/Rab4A$^{Q72L}$-KO mice (Fig. S12D). Glucose flux through the PPP was unaffected in CD8$^+$ T cells since neither R5P was enriched by Rab4A activation in B6.TC/Rab4A$^{Q72L}$ mice nor S7P was accumulated by Rab4A inactivation in B6.TC/Rab4A$^{Q72L}$-KO mice (Fig. S12D). Thus, enhanced glucose metabolism through the non-oxidative branch of the PPP supports greater proliferative capacity of CD4$^+$ T cells of B6.TC/Rab4A$^{Q72L}$ mice[10,58].

### The Rab4A/mTOR axis promotes KYN accumulation in CD8$^+$ T cells and sera of B6.TC/Rab4A$^{Q72L}$ mice

Given that robust Rab4A-driven mTOR-dependent metabolic changes in T cells were accompanied by mTOR activation and expansion of B cells of B6.TC/Rab4A$^{Q72L}$ mice, all of which were reversed in B6.TC/Rab4A$^{Q72L}$−KO mice, we investigated whether metabolites may have transmitted inflammation via the bloodstream. Serum metabolome analysis showed an accumulation of creatinine in B6.TC/Rab4A$^{Q72L}$ mice at 38.5 weeks of age, reflecting accelerated GN (Fig. 5A). In these sera, TRP metabolites KYN, kynurenic acid (KYNA), and anthranilate; valine (VAL) and lysine (LYS); PPP-connected sugars ribose, deoxyribose, 5-aminoimidazole-4-carboxamide ribonucleotide (AICAR), S-ribosyl-homocysteine, purine, deoxyadenosine, adenosine phosphosulfate;

**Fig. 4 | Rab4A exerts cell type-specific control over mitochondrial metabolism between CD4+ and CD8+ T cells. A** Measurement of mitochondrial $O_2$ consumption rate (OCR) in CD4+ T cells. OCR curves of B6.TC, B6.TC/Rab4A$^{Q72L}$ and B6.TC/Rab4A$^{Q72L}$-KO mice were compared upon treatment with 0.2% CMC (Vehicle panel) or rapamycin dissolved in 0.2% CMC (Rapamycin panel). Exact $p$ values are displayed for analyses by repeated measures ANOVA. The numbers ($n$) of mice in each experimental group were as follows: B6.TC Veh ($n = 5$), B6.TC Rapa ($n = 5$), B6.TC/Rab4A$^{Q72L}$ Veh ($n = 4$), B6.TC/Rab4A$^{Q72L}$ NAC ($n = 5$), B6.TC/Rab4A$^{Q72L}$ Rapa ($n = 6$), B6.TC/Rab4A$^{Q72L}$-KO Veh ($n = 3$), B6.TC/Rab4A$^{Q72L}$-KO Rapa ($n = 2$). **B** Cumulative analysis of individual mitochondrial functional checkpoints, basal respiration and mitochondrial ATP production. Dot plot charts reflect mean ± SE of each experimental group normalized to vehicle-treated control B6.TC (V) mice. The numbers ($n$) of mice in each experimental group were as follows: B6.TC Veh ($n = 5$), B6.TC Rapa ($n = 5$), B6.TC/Rab4A$^{Q72L}$ Veh ($n = 4$), B6.TC/Rab4A$^{Q72L}$ NAC ($n = 5$), B6.TC/Rab4A$^{Q72L}$ Rapa ($n = 6$), B6.TC/Rab4A$^{Q72L}$-KO Rapa ($n = 2$). One-way ANOVA Sidak's post-hoc test $p$ values corrected for multiple comparisons; basal respiration B6.TC (Veh) vs B6.TC/Rab4A$^{Q72L}$ (Veh) $p = 0.0473$; ATP production B6.TC (Veh) vs B6.TC/Rab4A$^{Q72L}$ (Veh) $p = 0.0412$. **C** Mitochondrial OCR in CD8+ T cells. The numbers ($n$) of mice in each experimental group were as follows: B6.TC Veh ($n = 5$), B6.TC Rapa ($n = 5$), B6.TC/Rab4A$^{Q72L}$ Veh ($n = 4$), B6.TC/Rab4A$^{Q72L}$ NAC ($n = 5$), B6.TC/Rab4A$^{Q72L}$ Rapa (n = 6), B6.TC/Rab4A$^{Q72L}$-KO Veh ($n = 3$), B6.TC/Rab4A$^{Q72L}$-KO Rapa ($n = 2$); B6.TC Veh vs B6.TC/Rab4A$^{Q72L}$ Veh vs B6.TC/Rab4A$^{Q72L}$-KO Veh two-way repeated-measures ANOVA p = 0.9997, B6.TC Rapa vs B6.TC/Rab4A$^{Q72L}$ Rapa vs B6.TC/Rab4A$^{Q72L}$-KO Rapa two-way repeated-measures ANOVA $p < 0.0001$. **D** Cumulative analysis of individual mitochondrial functional checkpoints, basal respiration and mitochondrial ATP production, in CD8+ T cells. Measurements were carried out, as described in (**B**). The numbers ($n$) of mice in each experimental group were as follows: B6.TC Veh ($n = 5$), B6.TC Rapa ($n = 5$), B6.TC/Rab4A$^{Q72L}$ Veh ($n = 4$), B6.TC/Rab4A$^{Q72L}$ NAC ($n = 5$), B6.TC/Rab4A$^{Q72L}$ Rapa (n = 6), B6.TC/Rab4A$^{Q72L}$-KO Veh ($n = 3$), B6.TC/Rab4A$^{Q72L}$-KO Rapa ($n = 2$). One-way ANOVA Sidak's post-hoc test $p$ values corrected for multiple comparisons; basal respiration B6.TC/Rab4A$^{Q72L}$ (Veh) vs B6.TC/Rab4A$^{Q72L}$ (Rapa) $p = 0.0235$; ATP production B6.TC/Rab4A$^{Q72L}$ (Veh) vs B6.TC/Rab4A$^{Q72L}$ (Rapa) $p = 0.0056$. **E** Opposite effects of Rab4A activation on basal respiration and mitochondrial ATP production between CD4+ and CD8+ T cells. Normalized values in B6.TC mice were compared to those of B6.TC/Rab4A$^{Q72L}$ mice with two-way ANOVA. The numbers ($n$) of mice in each experimental group were as follows: B6.TC Veh ($n = 5$), B6.TC/Rab4A$^{Q72L}$ Veh ($n = 4$); **F** Schematic diagram of Rab4A-mediated cell type-specific changes in mitochondrial metabolism between CD4+ and CD8+ T cells. Rab4A increased mitochondrial respiration and ATP production in CD4+ T cells while it exerted an opposite effect in CD8+ T cells. NAC and rapamycin treatment in vivo reversed the changes in mitochondrial metabolism in CD4+ T cells, while rapamycin reversed these changes in CD8+ T cells. **G** Assessment of metabolic flux of [U-$^{13}$C]-glutamine through the mitochondrial TCA cycle in CD4+ T (top row) and CD8+ T cells (middle row). Dot plot charts show mean ± SE of % enrichment of TCA substrates [M5-$^{13}$C]-glutamate, [M5-$^{13}$C]-α-ketoglutarate ([M5-$^{13}$C]-αKG), [M4-$^{13}$C]-

fumarate, and [M4-$^{13}$C]-malate. Overall one-way ANOVA $p$ values are shown in the header of each figure panel, while Sidak's post-hoc test $p$ values < 0.05 over brackets reflect comparison between experimental groups. Effects of Rab4A activation and inactivation within CD4+ and CD8+ T cells were compared by two-way ANOVA. The numbers ($n$) of mice in each experimental group were as follows: B6.TC Veh ($n = 5$), B6.TC Rapa ($n = 4$), B6.TC/Rab4A$^{Q72L}$ Veh ($n = 4$), B6.TC/Rab4A$^{Q72L}$ NAC ($n = 5$), B6.TC/Rab4A$^{Q72L}$ Rapa ($n = 6$), B6.TC/Rab4A$^{Q72L}$-KO Veh ($n = 2$), B6.TC/Rab4A$^{Q72L}$-KO Rapa ($n = 2$). One-way ANOVA Sidak's post-hoc test $p$ values corrected for multiple comparisons; CD4+ T cells [M5-$^{13}$C]-glutamate enrichment B6.TC/Rab4A$^{Q72L}$ (Veh) vs B6.TC/Rab4A$^{Q72L}$ (NAC) $p = 0.0179$; CD4+ T cells [M4-$^{13}$C]-fumarate enrichment B6.TC/Rab4A$^{Q72L}$ (Veh) vs B6.TC/Rab4A$^{Q72L}$–KO (Veh) $p = 0.0136$; CD4+ T cells [M4-$^{13}$C]-glutamate enrichment B6.TC/Rab4A$^{Q72L}$ (Veh) vs B6.TC/Rab4A$^{Q72L}$ (NAC) $p = 0.0307$; CD8+ T cells [M5-$^{13}$C]-α-ketoglutarate enrichment B6.TC (Veh) vs B6.TC/Rab4A$^{Q72L}$ (Veh) $p = 0.0457$; CD8+ T cells [M4-$^{13}$C]-fumarate enrichment B6.TC (Veh) vs B6.TC/Rab4A$^{Q72L}$ (Veh) $p = 0.0185$; CD8+ T cells [M4-$^{13}$C]-malate enrichment B6.TC (Veh) vs B6.TC/Rab4A$^{Q72L}$ (Veh) $p = 0.0038$; CD8+ T cells [M4-$^{13}$C]-citrate enrichment B6.TC (Veh) vs B6.TC/Rab4A$^{Q72L}$ (Veh) $p = 0.0042$; CD8+ T cells [M4-$^{13}$C]-citrate enrichment B6.TC (Veh) vs B6.TC (Rapa) $p = 0.0348$; CD8+ T cells [M3-$^{13}$C]-glutamate enrichment B6.TC (Veh) vs B6.TC/Rab4A$^{Q72L}$ (Veh) $p = 0.0215$. Two-way ANOVA of CD4 vs CD8 and B6.TC vs B6.TC/Rab4A$^{Q72L}$: [M5-$^{13}$C]-glutamate enrichment two-way ANOVA p = 0.0164, CD4 B6.TC vs CD8 B6.TC Sidak's post-hoc test $p = 0.0036$; [M5-$^{13}$C]-αKG enrichment two-way ANOVA $p = 0.0004$, CD4 B6.TC vs CD8 B6.TC Sidak's post-hoc test $p < 0.0001$, CD8 B6.TC vs CD8 B6.TC/Rab4A$^{Q72L}$ Sidak's post-hoc test $p = 0.0006$; [M4-$^{13}$C]-fumarate enrichment two-way ANOVA p = 0.0022, CD4 B6.TC vs CD8 B6.TC Sidak's post-hoc test $p < 0.0001$, CD8 B6.TC vs CD8 B6.TC/Rab4A$^{Q72L}$ Sidak's post-hoc test $p = 0.0126$; [M4-$^{13}$C]-malate enrichment two-way ANOVA $p = 0.0005$, CD4 B6.TC vs CD8 B6.TC Sidak's post-hoc test $p < 0.0001$, CD8 B6.TC vs CD8 B6.TC/Rab4A$^{Q72L}$ Sidak's post-hoc test $p = 0.0005$; [M4-$^{13}$C]-citrate enrichment two-way ANOVA $p = 0.0007$, CD4 B6.TC vs CD8 B6.TC Sidak's post-hoc test $p < 0.0001$, CD8 B6.TC vs CD8 B6.TC/Rab4A$^{Q72L}$ Sidak's post-hoc test $p = 0.0011$; [M4-$^{13}$C]-glutamate enrichment two-way ANOVA $p = 0.0201$, CD4 B6.TC vs CD8 B6.TC Sidak's post-hoc test $p < 0.0001$, CD8 B6.TC vs CD8 B6.TC/Rab4A$^{Q72L}$ Sidak's post-hoc test $p = 0.0234$; [M3-$^{13}$C]-glutamate enrichment two-way ANOVA $p = 0.0016$, CD4 B6.TC vs CD8 B6.TC Sidak's post-hoc test $p < 0.0001$, CD8 B6.TC vs CD8 B6.TC/Rab4A$^{Q72L}$ Sidak's post-hoc test $p = 0.0012$. **H** Schematic diagram of Rab4A-mediated cell type-specific changes in mitochondrial metabolism between CD4+ and CD8+ T cells. Rab4A increased mitochondrial respiration and ATP production in CD4+ T cells while it exerted an opposite effect in CD8+ T cells. Metabolic flux in the TCA cycle was increased in CD4+ T cells but reduced in CD8+ T cells; which were highlighted by red circular arrows and red TCA designation in CD4+ T cells and blue circular arrows and blue TCA designation in CD8+ T cells, respectively· within schematic mitochondria. NAC and rapamycin treatment in vivo reversed the changes in mitochondrial metabolism in CD4+ T cells, while rapamycin reversed these changes in CD8+ T cells.

---

fuels of mitochondrial metabolism (pyruvate, succinyl-carnitine, oxaloacetate, acetoacetate, 2-keto-butyrate, hydroxymethylbutyrate, dodecanedioate, 3-methylglutaconic acid) and products of oxidative stress malondialdehyde (MDA), dihydrothymine, acrylic acid, carboxymethyllysine, arachidonic acid were accumulated (Fig. 5A). KYN levels were elevated in CD8+ over CD4+ T cells of B6 mice (3.2-fold, FDR $p = 0.0071$). Moreover, KYN was further accumulated in CD8+ T cells of B6.TC/Rab4A$^{Q72L}$ mice over B6.TC controls (Fig. 5B). In addition to KYN, other precursors of de novo pyridine nucleotide biosynthesis, i.e., KYNA, 3-OH-KYN, quinolinate (QUIN), and nicotinate were all accumulated in CD8+ T cells but depleted in CD4+ T cells of B6.TC/Rab4A$^{Q72L}$ mice (Fig. 5B). Inactivation of Rab4A distorted the accumulation of [M5-$^{13}$C]-KYN into opposite directions in [11-$^{13}$C]-TRP-labeled CD4+ and CD8+ T cells of 20-week-old B6.TC/Rab4A$^{Q72L}$-KO mice over age-matched female B6.TC/Rab4A$^{Q72L}$ controls (2-way ANOVA $p = 0.0315$; Fig. 5C). Rab4A promoted the accumulation of [M8-$^{13}$C]-KYNA but reduced the enrichment of [M2-$^{13}$C]-αKG in [10-$^{13}$C]-KYN-labeled CD8+ T cells of B6.TC/Rab4A$^{Q72L}$ mice, which was reversed by the inactivation of Rab4A in B6.TC/Rab4A$^{Q72L}$-KO mice (Fig. 5D). Importantly, KYN → KYNA accumulation occurred with the depletion of αKG (Fig. 4G), indicating that

KYN metabolism may siphon off this substrate from the TCA cycle (Fig. 5E).

## KYN induces CD4+ T-cell expansion, CD8+ T-cell depletion, B-cell activation, and plasma cell expansion amongst mouse splenocytes

Among the metabolites accumulated in the sera of B6.TC/Rab4A$^{Q72L}$ mice (Fig. 5A), KYN in and of itself elicited the accumulation of mitochondria and the production of reactive oxygen intermediates (ROS) (Fig. 6A) and increased CD98 expression in CD4+ and CD8+ T cells (Fig. 6B). As measured by mean fluorescence intensity (MFI), the extent of CD98 expression was greater on CD8+ T cells (ANOVA p < 0.0001) that occurred with the contraction CD8+ T cells relative to CD4+ T cells upon treatment with KYN and concurrent CD3/CD28 co-stimulation (ANOVA $p = 0.0059$; Fig. 6B). The cell type-specific differences in KYN accumulation and KYN-induced contraction of CD8+ T cells may be attributed to markedly elevated expression of CD98 on CD8+ T cells over CD4+ T cells upon CD3/CD28 co-stimulation (ANOVA $p = 0.0100$; Fig. 6B). KYN increased mTORC1 and mTORC2 activity both in CD4+ (Fig. 6C) and CD8+ T cells (Fig. 6D).

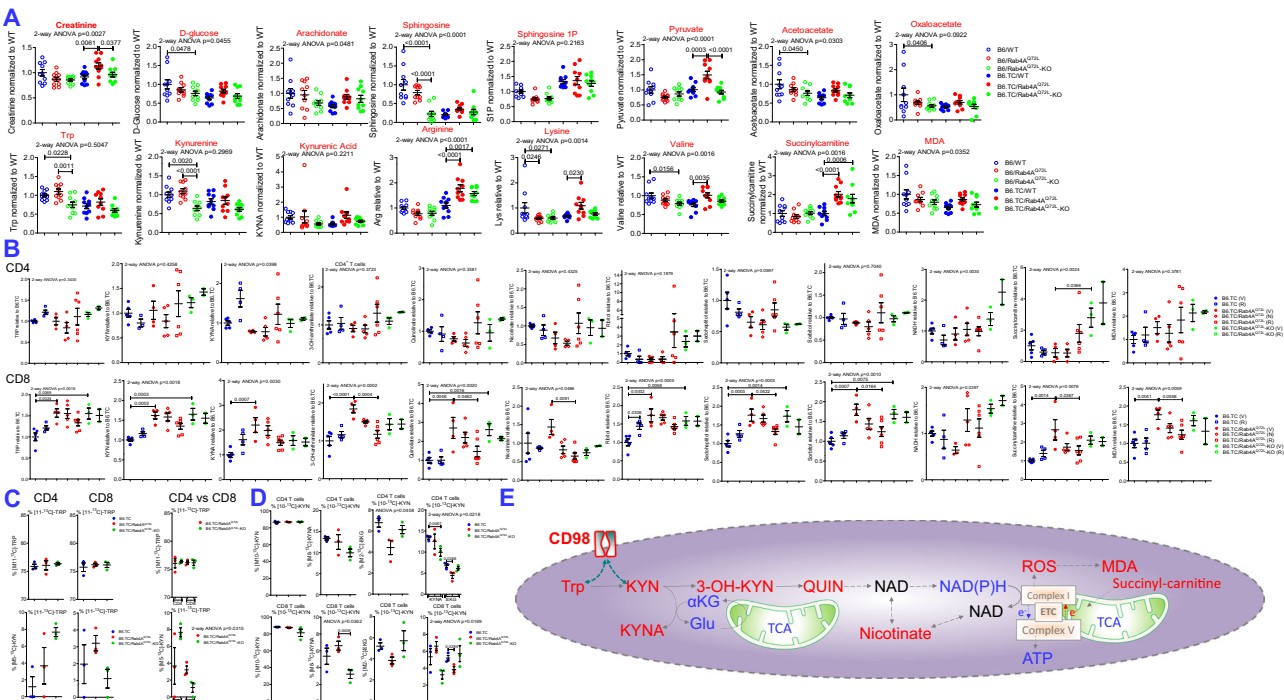

**Fig. 5 | Rab4A/mTOR axis promotes C5-C7 polyol and KYN accumulation in CD8⁺ T cells and sera of B6.TC/Rab4A^Q72L mice. A** Effect of Rab4A activation on serum metabolite concentrations in female B6 and lupus-prone mice B6.TC mice. A total of 60 mice were analyzed, 10 mice per each of six genotypes: B6, B6/Rab4A^Q72L, B6/Rab4A^Q72L-KO, B6.TC, B6.TC/Rab4A^Q72L, B6.TC/Rab4A^Q72L-KO. The mice were 39.1 ± 1.1 weeks of age and matched for age amongst the genotypes. Overall two-way ANOVA *p* values are shown in the header of each figure panel, while Tukey's post-hoc test *p* values < 0.05 over brackets reflect comparison between experimental groups. The numbers of mice was *n* = 10 in each experimental group. Charts show mean ± SEM for each experimental group. **B** Effect of Rab4A activation and treatment by rapamycin and NAC on TRP, KYN, pyridine nucleotide, and C5-C7 polyol concentrations in CD4⁺ and CD8⁺ T cells of B6.TC, B6.TC/Rab4A^Q72L, and B6/Rab4A^Q72L-KO mice described in Fig. 3. Data represent concentration values normalized to those of B6.TC control mice. The numbers (*n*) of mice in each

experimental group were as follows: B6.TC Veh (*n* = 5), B6.TC Rapa (*n* = 4), B6.TC/Rab4A^Q72L Veh (*n* = 4), B6.TC/Rab4A^Q72L NAC (*n* = 5), B6.TC/Rab4A^Q72L Rapa (*n* = 6), B6.TC/Rab4A^Q72L-KO Veh (*n* = 3), B6.TC/Rab4A^Q72L-KO Rapa (*n* = 2). Charts show mean ± SEM for each experimental group. **C** Inactivation of Rab4A distorted the accumulation of [M5-¹³C]-KYN into opposite directions in [11-¹³C]-TRP-labeled CD4⁺ and CD8⁺ T cells of 20-week-old B6.TC/Rab4A^Q72L-KO mice over age-matched female B6.TC/Rab4A^Q72L controls. The number of mice was *n* = 3 in each experimental group. Charts show mean ± SEM for each experimental group. **D** Rab4A promoted the accumulation of [M8-¹³C]-KYNA but reduced the enrichment of [M2-¹³C]-αKG in [10-¹³C]-KYN-labeled CD8⁺ T cells of B6.TC/Rab4A^Q72L mice, which was reversed by the inactivation of Rab4A in B6.TC/Rab4A^Q72L-KO mice. The number of mice was *n* = 3 in each experimental group. Charts show mean ± SEM for each experimental group. **E** Mechanistic diagram of metabolic pathways underlying increased production of KYN in CD8⁺ T cells of B6.TC/Rab4A^Q72L mice.

Next, we evaluated whether the accumulation of KYN in sera of B6.TC/Rab4A^Q72L mice may have contributed to B-cell activation. Similar to T cells, KYN directly activated mTORC1 and mTORC2 and increased CD98 expression in CD19⁺ (Fig. S13A), CD19⁺CD38⁺ B cells (Fig. S13B), CD19⁺CD11c⁺ ABCs (Fig. S13C), and CD138⁺ plasma cells (Fig. S13D). During concurrent lipopolysaccharide (LPS) stimulation, KYN continued to robustly activate mTORC1 and mTORC2 and increased CD98 expression in CD19⁺ (Fig. S13A), and CD19⁺CD38⁺ B cells (Fig. S13B) and selectively activated mTORC2 in CD19⁺CD11c⁺ ABCs (Fig. S13C) and mTORC1 and CD98 in CD138⁺plasma cells (Fig. S13D). Concurrent LPS and KYN stimulation markedly expanded pS6RP⁺CD138⁺ and CD98⁺CD138⁺ plasma cells (Fig. S13E). While KYN or LPS alone failed to influence the overall abundance of plasma cells, KYN and LPS together expanded CD138⁺ plasma cells (Fig. S13F) that exhibited increased mTORC1 activity and CD98 expression (Figure S13E). These findings indicate that the accumulation of KYN may underlie B-cell activation in B6.TC/Rab4A^Q72L mice.

**Rab4A-directed endosome traffic mediates discordant CD98 expression between CD4⁺ and CD8⁺ T cells during lupus pathogenesis**

Given that Rab4A impacts cellular function through endocytic recycling, we systematically assessed its influence on the expression of surface receptors that traffic via endosomes, such as CD71[6,27],

CD38[59,60], CD68[61,62], CD98[63,64], SERT[65,66], and CD152[67,68] (Fig. 7A). Each of these receptors, CD38[69], CD68[70], CD71[71], CD98[72], CD152[73], and SERT, transmit signals that regulate mitochondrial metabolism[74]. The tracking of recycled receptors delineated five cell clusters in B6.TC mice which were impacted by altered expression of Rab4A and therapeutic intervention by rapamycin and NAC: cluster 1, CD71⁺CD98⁺CD152⁺ T cells; cluster 2, CD4⁺ T cells; cluster 3, CD71⁺CD98⁺SERT⁺ DN T cells; cluster 4, CD8⁺ T cells; cluster 5, CD71⁺CD98⁺CD152⁻ T cells (Fig. 7B). The trafficked receptors comprised distinct subsets within CD4⁺, CD8⁺, and DN T cells. Across all T cells, two clusters were skewed into opposite directions by the activation over inactivation by Rab4A: CD68⁺CD71⁺CD98⁺SERT⁺CD152⁻ (cluster 3) and CD68⁻CD71⁻CD98⁺SERT⁻CD152⁻ T cells (cluster 4), respectively, (Fig. 7B). Clusters 3 and 5 were expanded, while cluster 4 was depleted in B6.TC/Rab4A^Q72L mice (Fig. 7C). These changes were reversed by rapamycin treatment of B6.TC/Rab4A^Q72L mice in vivo or by the inactivation Rab4A in B6.TC/Rab4A^Q72L–KO mice (Fig. 7C).

Both CD71 and CD98 were most abundantly expressed on DN T cells and the majority of CD98⁺ T cells co-expressed CD71 in B6.TC mice (Fig. 8A). CD71⁺CD98⁺ double-positive cells constituted 0.7 ± 0.1% of CD4⁺ (*p* = 0.0050 relative to DN T cells), 0.3 ± 0.1% of CD8⁺ (*p* = 0.0050 relative to DN T cells) and 14.8 ± 3.8% of DN T cells, respectively (Fig. 8A, B). These CD71⁺CD98⁺ cells were expanded in CD4⁺, CD8⁺, and DN T cells by activation of Rab4A in B6.TC/Rab4A^Q72L

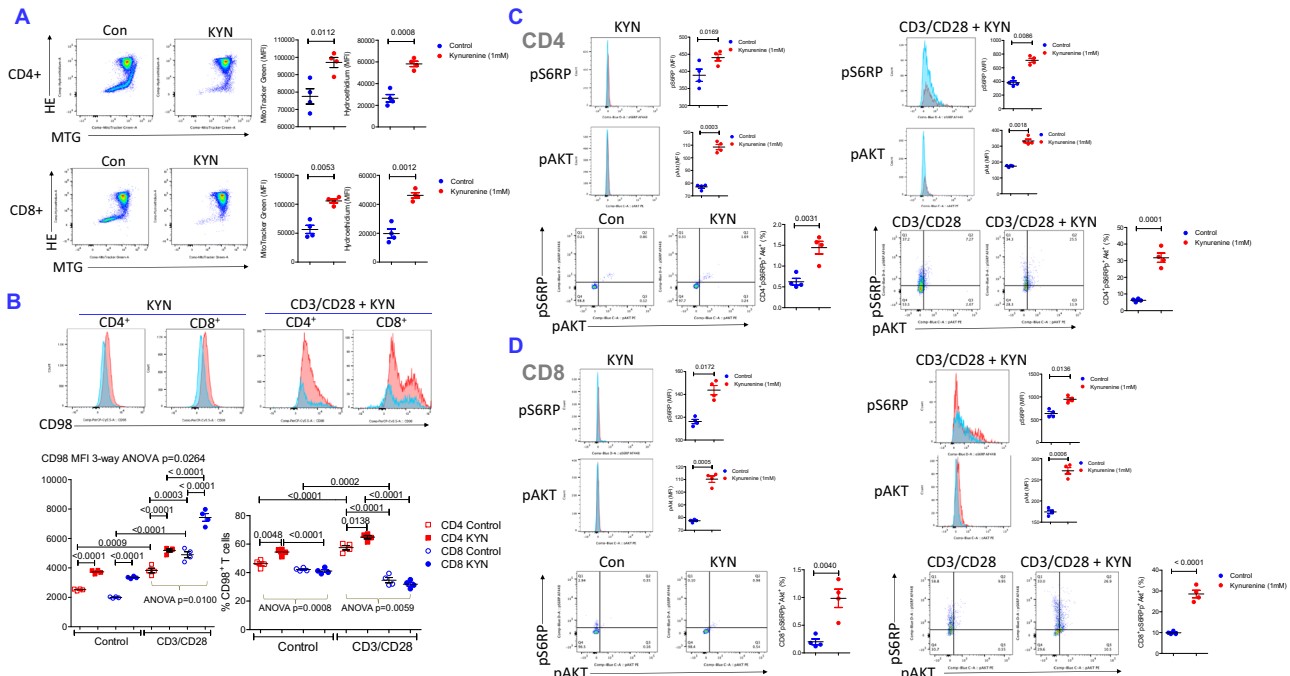

**Fig. 6 | Contrasting effects of KYN on activation of mTORC1 and mTORC2, expression of CD98 and relative abundance of primary CD4⁺ and CD8 mouse T⁺ cells.** Splenocytes from four female B6 mice were stimulated with 1 mM KYN alone or together with CD3/CD28 for 72 h in vitro, as indicated for each panel. Phenotyping for expression of CD3, CD19, CD4, CD8, and CD98 was performed by flow cytometry. **A** Measurement of mitochondrial mass and ROS production by MTG and HE fluorescence, respectively. Left panels show representative flow cytometry dot plots, while right panels show GraphPad charts of cumulative analyses. Brackets represent $p$ values < 0.05 by comparison using two-tailed paired $t$ test. The number of mice was $n = 4$ in each experimental group. Charts show mean ± SEM for each experimental group. **B** Effect of KYN on the expression of CD98 and the prevalence of CD4⁺ and CD8⁺ T cells with and without concurrent CD3/CD28 co-stimulation. Representative flow cytometry histograms and dot plots and mean ± SE four independent experiments are shown. Brackets show $p$ values < 0.05 using 3-way ANOVA and Sidak's post-hoc tests to correct for multiple comparisons. CD98 MFI three-way ANOVA $p = 0.0264$; Sidak's post-hoc test p values corrected for multiple comparisons: KYN vs control unstimulated CD4⁺ T cells $p < 0.0001$, KYN vs control unstimulated CD8⁺ T cells $p < 0.0001$, KYN vs control CD3CD28-stimulated CD4⁺ T cells $p < 0.0001$, KYN vs control CD3CD28-stimulated CD8⁺ T cells $p < 0.0001$, CD3CD28-stimulated vs unstimulated CD4⁺ T cells $p = 0.0009$, CD3CD28-stimulated vs unstimulated CD8⁺ T cells $p < 0.0001$, control CD3CD28-stimulated CD4⁺ vs control CD3CD28-stimulated CD8⁺ T cells $p = 0.0003$, KYN-treated CD3CD28-stimulated CD4⁺ vs KYN-treated CD3CD28-stimulated CD8⁺ T cells $p < 0.0001$. Two-way ANOVA of CD3CD28-stimulated control and KYN-treated CD4⁺ and CD8⁺ T cells $p = 0.0100$. %CD98⁺ cells three-way ANOVA p = 0.6602; Sidak's post-hoc test p values corrected for multiple comparisons: KYN vs control unstimulated CD4⁺ T cells $p = 0.0048$, KYN-treated unstimulated CD4⁺ T cells vs KYN-treated unstimulated CD8⁺ T cells

$p < 0.0001$, KYN vs control CD3CD28-stimulated CD4⁺ T cells $p = 0.0138$, CD3CD28-stimulated vs unstimulated CD4⁺ T cells $p < 0.0001$, CD3CD28-stimulated vs unstimulated CD8⁺ T cells $p = 0.0002$, control CD3CD28-stimulated CD4⁺ vs control CD3CD28-stimulated CD8⁺ T cells $p < 0.0001$, KYN-treated CD3CD28-stimulated CD4⁺ vs KYN-treated CD3CD28-stimulated CD8⁺ T cells $p < 0.0001$. Two-way ANOVA of unstimulated control and KYN-treated CD4⁺ and CD8⁺ T cells $p = 0.0008$. Two-way ANOVA of CD3CD28-stimulated control and KYN-treated CD4⁺ and CD8⁺ T cells $p = 0.0059$. **C** Effect of KYN on mTORC1 and mTORC2 activities in CD8⁺ T cells with and without concurrent CD3/CD28 co-stimulation. Following staining for surface expression of CD3, CD19, CD4, CD8, and CD98, cells were fixed and permeabilized and activities of mTORC1 and mTORC2 were measured by intracellular staining for pS6RP and pAkt, respectively. Representative flow cytometry histograms and dot plots and mean ± SE four independent experiments are shown. Brackets represent $p$ values < 0.05 by comparison using two-tailed paired t-test: KYN-treated vs control unstimulated cells pS6RP MFI $p = 0.0169$, pAkt MFI $p = 0.0003$, CD4⁺pS6RP⁺pAkt⁺ (%) $p = 0.0031$, KYN-treated vs control CD3CD28-stimulated cells pS6RP MFI $p = 0.0086$, pAkt MFI $p = 0.0018$, CD4⁺pS6RP⁺pAkt⁺ (%) $p = 0.0001$. **D** Effect of KYN on mTORC1 and mTORC2 activities in CD8⁺ T cells with and without concurrent CD3/CD28 co-stimulation. Following staining for surface expression of CD3, CD19, CD4, CD8, and CD98, cells were fixed and permeabilized and activities of mTORC1 and mTORC2 were measured by intracellular staining for pS6RP and pAkt, respectively. Representative flow cytometry histograms and dot plots and mean ± SE four independent experiments are shown. Brackets represent $p$ values < 0.05 by comparison using two-tailed paired $t$ test: KYN-treated vs control unstimulated cells pS6RP MFI $p = 0.0172$, pAkt MFI $p = 0.0005$, CD8⁺pS6RP⁺pAkt⁺ (%) $p = 0.0040$, KYN-treated vs control CD3CD28-stimulated cells pS6RP MFI $p = 0.0136$, pAkt MFI $p = 0.0006$, CD8⁺pS6RP⁺pAkt⁺ (%) $p < 0.0001$.

mice (Fig. 8A, B). The specificity of these changes was confirmed by their reversal upon inactivation of Rab4A in T cells of B6.TC/Rab4A^Q72L-KO mice (Fig. 8A and B). The CD71⁺CD98⁻ and CD71⁻CD98⁺ single-positive subsets were also expanded in DN T cells of Rab4A B6.TC/Rab4A^Q72L mice over B6.TC and B6.TC/Rab4A^Q72L-KO controls (Fig. 8B). The expansion of CD71⁺CD98⁺ double-positive cells was reversed by rapamycin in CD8⁺ and DN T cell subsets of B6.TC/Rab4A^Q72L mice (Fig. 8B). The CD71⁺CD98⁻ and CD71⁻CD98⁺ single-positive and CD71⁻CD98⁻ subsets of DN T cells in B6.TC/Rab4A^Q72L mice were also restrained by rapamycin (Fig. 8B).

CD71 expression by MFI alone was also highest in DN T cells relative to CD4⁺ (1.5-fold; $p = 0.0247$) and CD8⁺ T cells (1.9-fold; $p = 0.0030$),

respectively (Fig. 8C). Elevated CD71 expression in DN T cells of B6.TC/Rab4A^Q72L mice was reduced by in vivo rapamycin treatment or the inactivation of Rab4A in B6.TC/Rab4A^Q72L-KO mice (Fig. 8C). CD98 expression was also highest in DN T cells relative to CD4⁺ (2.2-fold; $p = 0.0081$) and CD8⁺ T cells (2.0-fold; $p = 0.0087$), respectively, most increased in B6.TC/Rab4A^Q72L mice, and reversed by rapamycin treatment or the deletion of Rab4A in T cells of B6.TC/Rab4A^Q72L-KO mice (Fig. 8D). Given that Rab4A impacts cellular function through endocytic traffic, we examined the influence of Rab4A on recycling of CD71 (Fig. 8E) and CD98 in 20-week-old female B6 and B6.TC mice, thus preceding the onset of SLE (Fig. 8F). Interestingly, CD71 expression was only elevated CD4⁺ T cells of B6.TC/Rab4A^Q72L mice over B6/Rab4A^Q72L

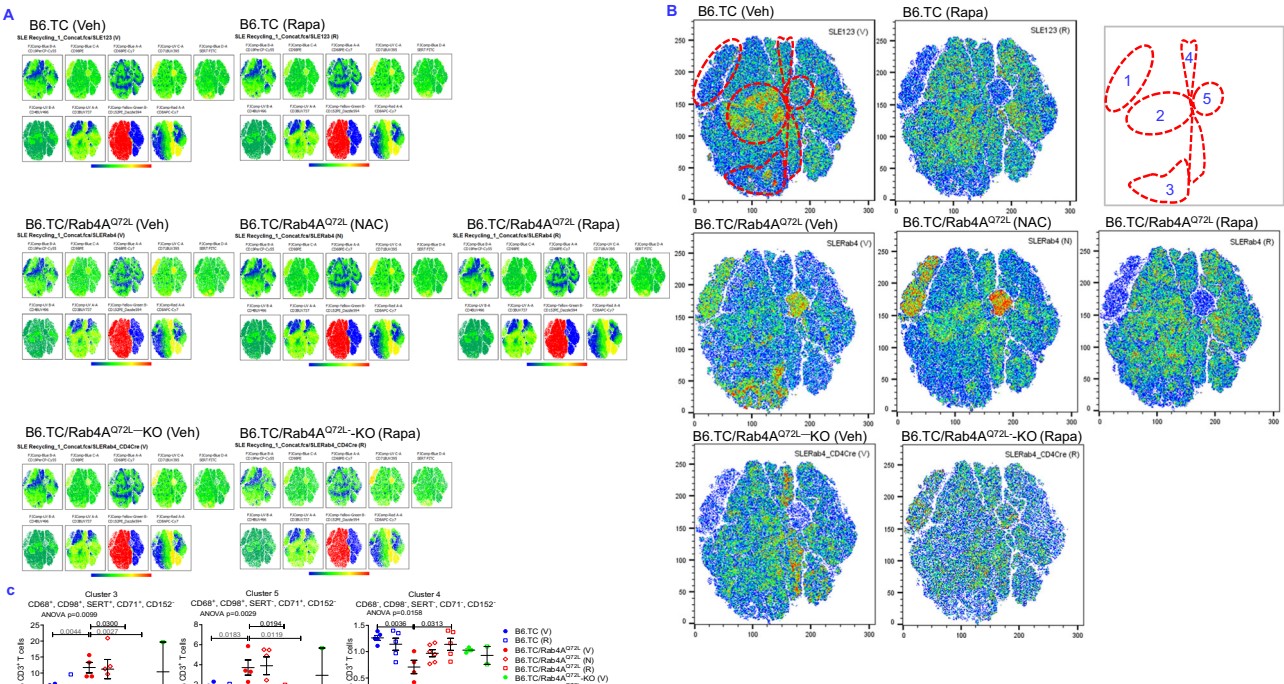

**Fig. 7 | Flow cytometry analysis of the impact of Rab4A and treatment by rapamycin and NAC on the surface expression of trafficked receptors in splenocyte subsets from lupus-prone mice.** B6.TC, B6.TC/Rab4A$^{Q72L}$, and B6/Rab4A$^{Q72L}$-KO mice were treated intraperitoneally (ip) three times weekly with 0.2% carboxymethylcellulose (CMC) vehicle solvent control (V), 3 mg/kg rapamycin (R), or 10 g/l of NAC (N) in drinking water. **A** Dimensionality reduction analyses of surface receptors by t-SNE in all mice combined. Color axis: blue (0) → red (max). Color scale from blue to red: −1622 to 262856. **B** Dimensionality reduction analyses by t-SNE delineated five cell clusters in response to altered expression of Rab4A and therapeutic intervention by rapamycin and NAC: cluster 1, CD71$^+$CD98$^+$CD152$^+$

T cells; cluster 2, CD4$^+$ T cells; cluster 3, CD71$^+$CD98$^+$SERT$^+$ DN T cells; cluster 4, CD8$^+$ T cells; cluster 5, CD71$^+$CD98$^+$CD152$^-$ T cells. **C** Statistical analysis of the impact of Rab4A and treatment with rapamycin and NAC on the abundance of clusters 3, 5, and 4. Overall one-way ANOVA p values are shown in the header of each figure panel, while Sidak's post-hoc test *p* values < 0.05 over brackets reflect comparison between experimental groups. The numbers (*n*) of mice in each experimental group were as follows: B6.TC Veh (*n* = 5), B6.TC Rapa (*n* = 5), B6.TC/Rab4A$^{Q72L}$ Veh (*n* = 4), B6.TC/Rab4A$^{Q72L}$ NAC (*n* = 5), B6.TC/Rab4A$^{Q72L}$ Rapa (*n* = 6), B6.TC/Rab4A$^{Q72L}$-KO Veh (*n* = 3), B6.TC/Rab4A$^{Q72L}$-KO Rapa (*n* = 2). Charts show mean ± SEM for each experimental group.

controls (Fig. 8E). Treatment with phorbol 12, 13-dibutyrate (PDBu) triggered a canonical Rab4-dependent internalization of CD71 in all T cells (Fig. 8E), as earlier described[27]. Of note, Rab4A exerted the most robust effect on the recycling of CD98 in CD8$^+$ T cells both in B6 and B6.TC mice (2-way RM ANOVA *p* < 0.0001; Fig. 8F). Moreover, CD98 exhibited strikingly discordant expression and traffic patterns in CD4$^+$, CD8$^+$, and DN T cells. CD98 expression was reduced in all T cells of B6/Rab4A$^{Q72L}$ mice but increased in all T cells of B6.TC/Rab4A$^{Q72L}$ mice, while these changes were consistently reversed in B6/Rab4A$^{Q72L}$-KO and B6.TC/Rab4A$^{Q72L}$-KO mice, respectively (Fig. 8F). CD98 expression in CD4$^+$ T cells was prominently increased by PDBu, which was followed by internalization and recycling after PDBu withdrawal (Fig. 8F). After recycling for 120 min, CD98 expression exceeded baseline levels in CD4$^+$ T cells (Fig. 8F). Since de novo protein synthesis was inhibited with cycloheximide during the recycling assay, CD98 was being stored in and recycled from the endocytic compartment. Unlike CD4$^+$ T cells, CD98 expression was not augmented by PDBu in CD8$^+$ T cells, however, it was also recycled in CD8$^+$ T cells of B6.TC and B6.TC/Rab4A$^{Q72L}$ mice. In striking contrast, CD98 was internalized by PDBu but failed to recycle in CD8$^+$ T cells of B6.TC/Rab4A$^{Q72L}$-KO mice (Fig. 8F). These findings indicate that CD98 expression is regulated by Rab4A via endocytic traffic in mTOR-dependent and cell type-specific manners.

## CD98 controls KYN uptake, TCA metabolism, and redox homeostasis in HeLa cells

CD98 is a ubiquitously expressed surface protein[75] that mediates the transport of branched-chain (VAL, LEU, ILE) and aromatic amino acids

(PHE, TYR, TRP)[76], and therefore, we examined the effects of its siRNA-mediated knockdown on these metabolic pathways and signal transduction in HeLa cells. CD98, also called SLC3A2, is a disulfide-linked heterodimer composed of a glycosylated heavy chain and a non-glycosylated light chain, large amino acid transporter 1 (LAT1), also called SLC7A5[77]. CD98 is detected as a 70–125 kD protein depending on the extent of glycosylation. CD98 has been localized to the cell surface[64], endosomes and lysosomes[63], where it has been implicated in activating mTORC1[78]. As shown in Fig. S14A, the selective knockdown of CD98, but not LAT1, reduced Akt phosphorylation (Fig. S14B) and distorted metabolic pathways led by changes in branched-chain and aromatic amino acid biosynthesis and degradation as well as the TCA cycle (Fig. S14C, D). LAT1 protein levels were not affected by Rab4A in CD4$^+$ or CD8$^+$ T cells of age-matched female B6.TC, B6.TC/Rab4A$^{Q72L}$, and B6.TC/Rab4A$^{Q72L}$-KO mice (Fig. S15A) or in Jurkat human CD4$^+$ T cell lines with altered expression of Rab4A (Fig. S15B). However, CD98 knockdown markedly reduced intracellular levels of branched-chain (VAL, LEU, ILE) and aromatic amino acids (PHE, TYR, TRP) as well as KYN. Moreover, CD98 knockdown exerted antioxidant effects, as evidenced by increased GSH/GSSG ratio and reduced homocysteine levels (Fig. S14F). While KYN is a driver of mitochondrial oxidative stress[79], its metabolism is dependent on the availability of a TCA metabolite, αKG[80–83]. Accordingly, the depletion of KYN was accompanied by the accumulation of αKG and other TCA metabolites upon CD98 knockdown (Fig. S14G). These findings suggest that CD98 expression impacts metabolic cross-talk between KYN metabolism and the TCA cycle.

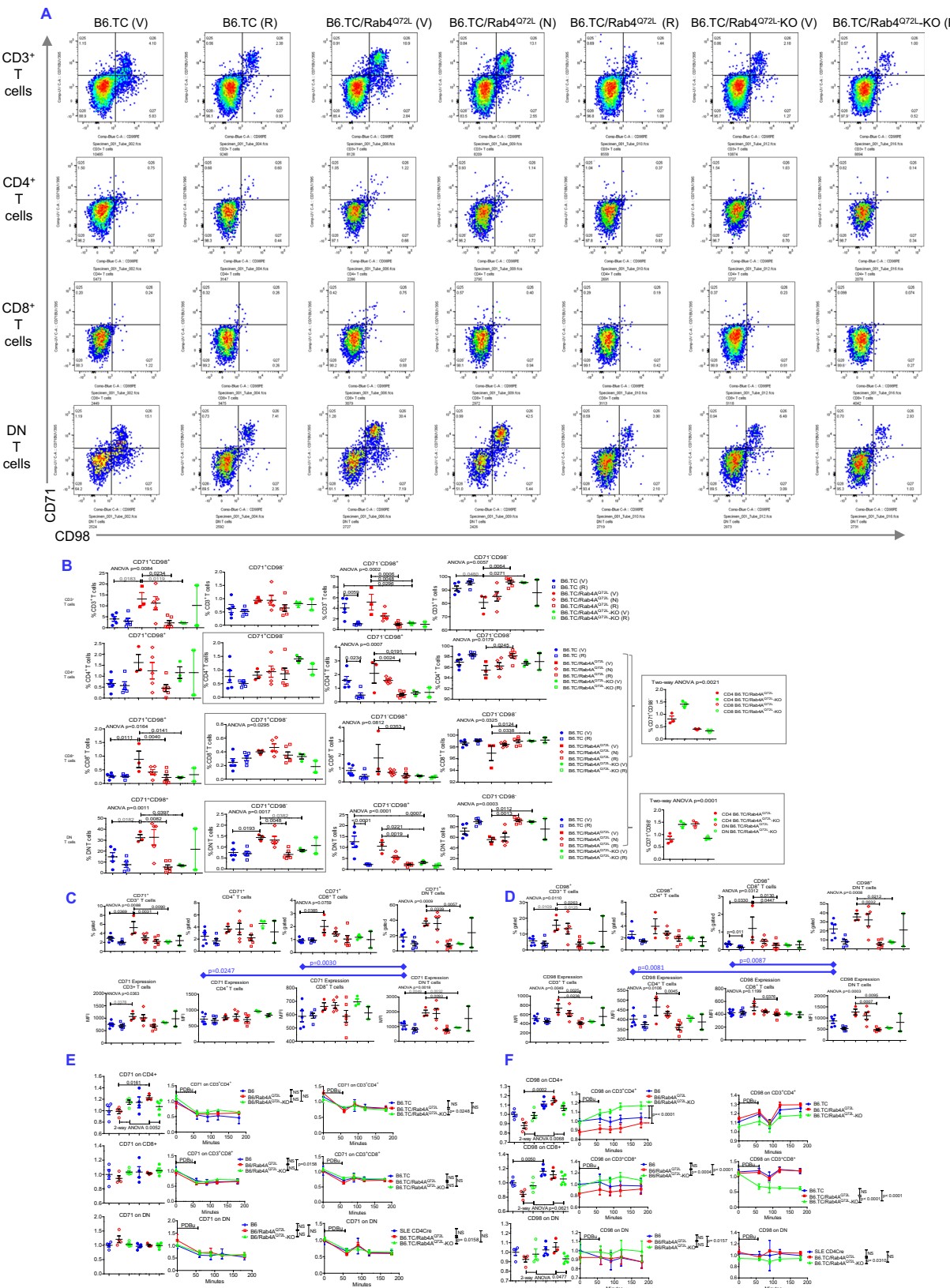

## Rab4A forms a positive feed-back loop with CD98 and mTOR in patients with SLE

Similar to lupus-prone B6.TC/Rab4A$^{Q72L}$ mice, flow cytometry of SLE and control participants, who had been matched for age and gender in the context of a clinical trial[15], unveiled an expansion of CD98$^+$ DN T cells in SLE patients (Fig. 9A, B). mTORC1 activation was confined to

CD98$^+$ T cells in SLE patients (Fig. 9B). Knockdown of CD98 by siRNA blocked CD3/CD28-induced expression of CD98 and restrained the activation of mTORC1 in primary human T cells (Fig. 9C). Over-expression of Rab4A enhanced the total protein levels (Fig. 9D) and increased the surface expression of CD98 in Jurkat cells (Fig. 9E). Thus, Rab4A-induced overexpression of CD98 may contribute to the

**Fig. 8 | The effect of Rab4A-directed endosome traffic and treatment by rapamycin and NAC on the surface expression of CD71 and CD98 in the CD3+, CD4+, CD8+, and DN T cell subsets of age-matched female lupus-prone mice.**
**A** Representative flow cytometry dot plots show the effect of Rab4A and treatment by rapamycin and NAC on the surface expression of CD71 and CD98 in the CD3+, CD4+, CD8+, and DN T cell subsets of age-matched female lupus-prone mice.
**B** Cumulative flow cytometry analyses represent the mean ± SE of the effect of Rab4A and treatment by rapamycin and NAC on concurrent surface expression of CD71 and CD98 in the CD3+, CD4+, CD8+, and DN T cell subsets of age-matched female lupus-prone mice. Overall one-way ANOVA p values are shown in the header of each figure panel, while Sidak's post-hoc test p values < 0.05 over brackets reflect comparison between experimental groups. The effects of Rab4A deletion between CD4+ and CD8+ T and between CD4+ and DN T cells were compared by two-way ANOVA. The numbers (n) of mice in each experimental group were as follows: B6.TC Veh (n = 5), B6.TC Rapa (n = 5), B6.TC/Rab4A^Q72L Veh (n = 3), B6.TC/Rab4A^Q72L NAC (n = 5), B6.TC/Rab4A^Q72L Rapa (n = 6), B6.TC/Rab4A^Q72L-KO Veh (n = 3), B6.TC/Rab4A^Q72L-KO Rapa (n = 2). Charts show mean ± SEM for each experimental group.
**C** The impact of Rab4A and treatment by rapamycin and NAC on the surface expression of CD71 in age-matched female B6.TC, B6.TC/Rab4A^Q72L, and B6/Rab4A^Q72L-KO mice. Dot plot charts represent cumulative assessment of the percentage of receptor-positive cells (top panels) and MFI of CD71 expression in the CD3+, CD4+, CD8+, and DN T cell compartments (bottom panels). Overall one-way ANOVA p values are shown in the header of each figure panel, while Sidak's post-hoc test p values < 0.05 over brackets reflect comparison between experimental groups. The numbers (n) of mice in each experimental group were as follows: B6.TC Veh (n = 5), B6.TC Rapa (n = 5), B6.TC/Rab4A^Q72L Veh (n = 3), B6.TC/Rab4A^Q72L NAC (n = 5), B6.TC/Rab4A^Q72L Rapa (n = 6), B6.TC/Rab4A^Q72L-KO Veh (n = 3), B6.TC/Rab4A^Q72L-KO Rapa (n = 2). Charts show mean ± SEM for each experimental group.
**D** The impact of Rab4A and treatment by rapamycin and NAC on the surface expression of CD98 in age-matched female B6.TC, B6.TC/Rab4A^Q72L, and B6/Rab4A^Q72L-KO mice. Dot plot charts represent cumulative assessment of the percentage of receptor-positive cells (top panels) and MFI of CD71 expression in the CD3+, CD4+, CD8+, and DN T cell compartments (bottom panels). Overall one-way ANOVA p values are shown in the header of each figure panel, while Sidak's post-hoc test p values < 0.05 over brackets reflect comparison between experimental groups; p values in blue reflect comparison between CD4+ and DN T cells or CD8+ DN T cells in B6.TC control mice. The numbers (n) of mice in each experimental

group were as follows: B6.TC Veh (n = 5), B6.TC Rapa (n = 5), B6.TC/Rab4A^Q72L Veh (n = 3), B6.TC/Rab4A^Q72L NAC (n = 5), B6.TC/Rab4A^Q72L Rapa (n = 6), B6.TC/Rab4A^Q72L-KO Veh (n = 3), B6.TC/Rab4A^Q72L-KO Rapa (n = 2). Charts show mean ± SEM for each experimental group. **E** Effect of Rab4A on endocytic traffic of CD71 in CD4+, CD8+, and DN T cells in B6/Rab4A^Q72L and B6.TC/Rab4A^Q72L mice (carrying constitutively active Rab4A^Q72L), B6/Rab4A^Q72L-KO and B6.TC/Rab4A^Q72L-KO mice (lacking Rab4A in T cells), and B6 and B6.TC controls. Left panel, baseline expression in all genotypes; middle panel, assessment of the effect of Rab4A on receptor recycling in B6 control mice; right panel, assessment of the effect of Rab4A in B6.TC SLE mice. 2-way ANOVA p values are shown within each figure panel, while Sidak's post-hoc test p values < 0.05 over brackets reflect comparison between experimental groups. The numbers (n) of mice in each experimental group were as follows: B6 (n = 4), B6/Rab4A^Q72L (n = 4), B6/Rab4A^Q72L-KO (n = 4), B6.TC (n = 4), B6.TC/Rab4A^Q72L (n = 3), B6.TC/Rab4A^Q72L-KO (n = 4). Charts show mean ± SEM for each experimental group. **F** Effect of Rab4A on endocytic traffic of CD98 in CD4+, CD8+, and DN T cells in B6/Rab4A^Q72L and B6.TC/Rab4A^Q72L mice (carrying constitutively active Rab4A^Q72L), B6/Rab4A^Q72L-KO and B6.TC/Rab4A^Q72L-KO mice (lacking Rab4A in T cells), and B6 and B6.TC controls. Left panel, baseline expression in all genotypes; middle panel, an assessment of the effect of Rab4A on receptor recycling in B6 control mice; right panel, assessment of the effect of Rab4A in B6.TC SLE mice. 2-way ANOVA p values are shown within each figure panel, while Sidak's post-hoc test p values < 0.05 over brackets reflect comparison between experimental groups. CD98 on CD4+ T cells 2-way ANOVA p = 0.0068, B6/Rab4A^Q72L vs B6.TC/Rab4A^Q72L Sidak's post-hoc test p = 0.0002, B6 vs B6/Rab4A^Q72L-KO repeated-measures ANOVA p < 0.0001, B6/Rab4A^Q72L vs B6/Rab4A^Q72L-KO repeated-measures ANOVA p < 0.0001, CD98 on CD8+ T cells 2-way ANOVA p = 0.0477, B6/Rab4A^Q72L vs B6.TC/Rab4A^Q72L Sidak's post-hoc test p = 0.0050, B6 vs B6/Rab4A^Q72L-KO repeated-measures ANOVA p < 0.0001, B6/Rab4A^Q72L vs B6/Rab4A^Q72L-KO repeated-measures ANOVA p = 0.0004, B6.TC vs B6.TC/Rab4A^Q72L-KO repeated-measures ANOVA p < 0.0001, B6.TC/Rab4A^Q72L vs B6.TC/Rab4A^Q72L-KO repeated-measures ANOVA p < 0.0001, CD98 on DN T cells 2-way ANOVA p = 0.0621, B6/Rab4A^Q72L vs B6.TC/Rab4A^Q72L Sidak's post-hoc test p = 0.0050, B6 vs B6/Rab4A^Q72L-KO repeated-measures ANOVA p = 0.0157, B6.TC/Rab4A^Q72L vs B6.TC/Rab4A^Q72L-KO repeated-measures ANOVA p = 0.0310. The numbers (n) of mice in each experimental group were as follows: B6 (n = 4), B6/Rab4A^Q72L (n = 4), B6/Rab4A^Q72L-KO (n = 4), B6.TC (n = 4), B6.TC/Rab4A^Q72L (n = 3), B6.TC/Rab4A^Q72L-KO (n = 4). Charts show mean ± SEM for each experimental group.

---

activation of mTORC1 in SLE. Similar to lupus T cells[6], FKBP12 expression was also induced by Rab4A in Jurkat cells (Fig. 9D). Further, elevated expression of CD98 expression was predictive of therapeutic efficacy of sirolimus, as measured by the SLE responder index (Fig. 9F).

As earlier uncovered, expression of Rab4A was enhanced by activation of mTORC1[6] and, reciprocally, Rab4A also promoted mTORC1 activation in Jurkat and primary human T cells[35]. Of note, the overexpression of wild-type Rab4A enhanced mTOR localization to the lysosomes, whereas dominant-negative Rab4A^S27N inhibited mTOR traffic to the lysosomes in Jurkat cells (Fig. S16A, B). mTOR activation depends on its localization to the lysosomal membranes where it senses amino acid sufficiency[20]. These findings thus suggest that enhanced traffic to the lysosome underlies Rab4A-dependent mTOR activation in Jurkat cells (Fig. S16C).

### Rab4A accelerates MOG-induced EAE
Since HRES-1/Rab4 polymorphism has also been linked to MS[33,84], we examined the impact of Rab4A in experimental autoimmune encephalitis (EAE), a mouse model for MS that is mediated by antigen-specific CD4+ T cells[85]. MOG-induced EAE was markedly enhanced in 12-week-old B6/Rab4A^Q72L mice over B6/WT controls, which was reversed upon deletion of Rab4A in T cells in B6/Rab4A^Q72L-KO mice (Fig. S17). Enhanced EAE was characterized greater lymphocytic infiltration as well as vasculitis, resulting in hemorrhage of the spinal cord in B6/Rab4A^Q72L mice (Fig. S17).

## Discussion
The present study provides evidence that Rab4A influences the traffic of multiple cargos that modulate T-cell homeostasis (Fig. S18). Rab4A promotes the development of CD4+ T cells at the expense of CD8+ T cells both in control B6 and lupus-prone B6.TC mice through regulating endosome traffic of the following cargos: 1) Rab4A elicits the depletion of mitophagy-initiator Drp1, and causes the accumulation of mitochondria in CD4+ T cells of B6/Rab4A^Q72L mice and B6.TC/Rab4A^Q72L mice; increased mitochondrial metabolism and metabolic flux through the TCA cycle is coupled with enhanced ATP production and increased metabolic flux through the PPP, which generates R5P, an essential precursor of de novo nucleotide biosynthesis required for cell proliferation; 2) Synergistically with SLE, Rab4A enhances the endocytic recycling of CD98 that mediates the uptake of KYN, which accumulates in CD8+ T cells and sera of B6.TC/Rab4A^Q72L mice and spreads mTOR activation to B cells and plasma cells; and 3) Rab4A directly promotes the localization of mTOR to lysosomes where it senses amino acid sufficiency[20]. Independent experimental approaches using i) flow cytometry; ii) mitochondrial O2 consumption assays; and iii) stable isotope tracing of metabolic pathways all indicate that the expansion of CD4+ T cells is fueled by increased mitochondrial mass, increased mitochondrial O2 consumption and ATP synthesis, metabolic flux through the mitochondrial TCA cycle and increased production of nucleotide precursor R5P via the PPP (Fig. S19). While mitochondrial mass was not increased by Rab4A activation in CD8+ T cells, their contraction occurred with MHP, reduced mitochondrial O2 consumption and reduced ATP production, and reduced carbon flux

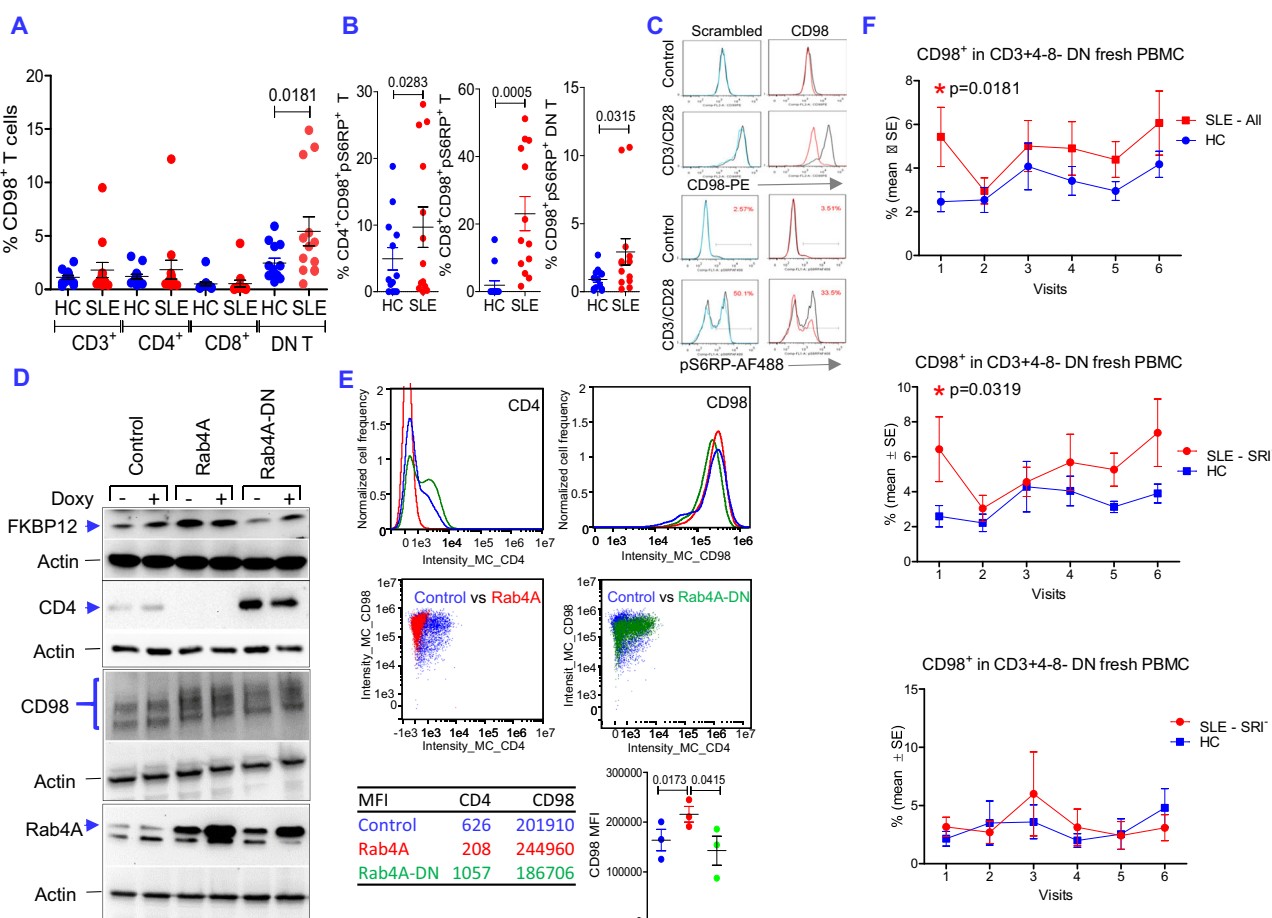

**Fig. 9 | CD98 expression promotes mTOR activation and predicts clinical response to sirolimus in patients with SLE. A** Quantitation of CD98+ T cells in 12 paired SLE and healthy control (HC) participants; two-tailed *t* test *p* = 0.0181. Charts show mean ± SEM for each experimental group. **B** mTORC1 activation is increased within CD98+ T lymphocytes in 12 SLE patients relative to 12 HC participants. Charts show mean ± SEM for each experimental group. *p* values < 0.05 are shown as determined by 2-tailed paired t-test. **C** Knockdown by siRNA indicates CD98 involvement in TCR-induced mTOR activation. Alexa647-conjugated CD98 or scrambled control siRNA was electroporated into 2 × 10^6 HC peripheral blood lymphocytes (PBL), as earlier described[6,35]. CD98PE and pS6RP-AF488 histograms were gated on Alexa 647+/CD3-APC-Cy7+ dual-positive cells. mTORC1 activation was assessed by the bracketed pS6RP-AF488+ cells, as earlier described[19,102]. **D** Rab4A promotes the surface expression of CD98. The effect of Rab4A was examined by western blot on the expression of FKBP12 and CD98 in Jurkat cells carrying doxycycline-inducible adeno-associated virus (AAV) expression vectors[27]. Western blots represent 5 or more similar experiments. Jurkat cells with construct 6678 overexpressed wild-type Rab4A while those with construct 9035 overexpressed dominant-negative Rab4A^S26N (Rab4A-DN), as earlier described[27]. Control cells carried "empty" vector construct 4480[27]. Rab4A, FKBP12, CD4, and actin were detected by antibodies described earlier[6]. CD98 was detected with antibody sc-9160 from Santa Cruz Biotechnology (Santa Cruz, CA). **E** Effect of Rab4A on surface

expression of CD98 was detected by flow cytometry. CD4 was detected as a control antigen that is targeted for lysosomal degradation by Rab4A, which is blocked by overexpression of Rab4A-DN[27]. Of note, the expression vectors confer moderate overexpression of Rab4A and Rab4A-DN in the absence of doxycycline, which are sufficient to exert opposing changes on CD4 expression in Jurkat cells of primary CD4 T cells, as earlier described[27]. Top and middle panels show histogram and dot plot overlays of CD4 and CD98 expression. Bottom panels show mean channel fluorescence intensity (MFI) of representative histograms of the top panel, while bar charts show mean ± SE of three independent experiments. *P* values represent comparison using two-tailed paired *t* test, connecting bars indicate *P* < 0.05, which reflect hypothesis testing and have not been corrected for multiple comparisons. **F** Effect of sirolimus (rapamycin) treatment on expression of CD98 in DN T cells in SLE patients during 12-month intervention. The prevalence of CD98+ DN cells was determined in thirteen freshly isolated PBL of SLE and HC participants matched for age within 10 years (top panel). Nine patients met criteria for SLE Responder Index (SRI+, middle panel), while 4 patients were SRI non-responders (SRI−, lowest panel). CD98+ DN cells were assessed before treatment (visit 1) and after treatment for 1 month (visit 2), 3 months (visit 3), 6 months (visit 4), 9 months (visit 5), and 12 months (visit 6). Effects of sirolimus were also assessed by 2-tailed paired *t* test relative to HC participants tested in parallel; *, *p* < 0.05. Charts show mean ± SE of patients and controls for each time point.

---

through the TCA cycle. These CD8+ T cells exhibited oxidative stress characterized by the accumulation of MDA, and the depletion of NADH, which is required for activity of ETC complex I (Fig. 5). Rab4A activation in CD8+ T cells increased the production of pyridine nucleotide precursors, such as KYN, 3OH-KYN, QUIN, and nicotinate. Moreover, independent lines of evidence support a pro-inflammatory role for KYN: 1) the accumulation of KYN in sera and CD8+ T cells preceded the onset of autoantibody production and GN disease onset in SLE in B6.TC/Rab4A^Q72L mice; 2) KYN itself enhanced CD4+ over CD8+ T-cell development in primary B6 mouse splenocytes; 3) KYN activated mTOR in B cells and expanded plasma cells in primary B6 mouse splenocytes; and

4) Rab4A formed a positive feed-back loop with mTOR activation and expression of metabolite-transporting CD98 receptor during lupus pathogenesis in mice and patients with SLE. Furthermore, KYN → KYNA accumulation occurred with the depletion of αKG in CD8+ but not CD4+ T cells of B6.TC/Rab4A^Q72L mice, indicating a Rab4A-driven cell type-specific crosstalk between KYN metabolism and the mitochondrial TCA cycle (Figs. 5, S19). These findings identify CD98-dependent accumulation of KYN, as a pro-inflammatory metabolite that may contribute to Rab4A/mTOR-driven autoimmunity in SLE.

As also unveiled by this study, Rab4A elicited mTOR activation in CD4+ and CD8+ T cells of B6/Rab4A^Q72L and B6.TC/Rab4A^Q72L mice over

B6 and B6.TC controls which was consistently reversed upon the inactivation of Rab4A in T cells of B6/Rab4A$^{Q72L}$-KO and B6.TC/Rab4A$^{Q72L}$-KO mice. Prior to the onset of autoimmunity, mTORC1 was activated in CD4$^+$ and CD8$^+$ T cells of 20-week-old B6.TC/Rab4A$^{Q72L}$ mice relative to B6/Rab4A$^{Q72L}$ controls. This suggests that Rab4A-mediated mTOR activation is a driver of autoimmunity via expansion of CD4$^+$ over CD8$^+$ T cells in SLE. A positive feed-back loop between Rab4A and mTOR was further substantiated by the reversal of mito-chondrial changes in CD4$^+$ and CD8$^+$ T cells of rapamycin-treated B6.TC/Rab4A$^{Q72L}$ mice. In mediating such fundamental control of T cell lineage development, Rab4A markedly distorted gene expression into sharply opposite directions between CD4$^+$ T cells and CD8$^+$ T cells, primarily affecting mitochondrial metabolism, endosome traffic, and autophagy pathways. The differential effect by rapamycin on meta-bolic flux between CD4$^+$ and CD8$^+$ T cells may be attributed, at least in part, to the variable reliance of these cells on glycolysis[86] relative to the mitochondrial TCA cycle[87], respectively.

The accumulation of mitochondria in CD4$^+$ T cells of B6/Rab4A$^{Q72L}$ and B6.TC/Rab4A$^{Q72L}$ mice is consistent with a role for Rab4A in depleting Drp1 and thus limiting mitophagy in lupus T cells[13,37]. Rab4A-mediated depletion of Drp1 was found to be mTOR dependent both in vitro and in vivo, as the retention of mitochondria in T cells of B6.TC/Rab4A$^{Q72L}$ mice was reversed by rapamycin treatment. The accumula-tion of oxidative stress-generating mitochondria in lupus T cells[13,46] has been widely confirmed[88,89] and recently extended to myeloid cells, such as neutrophils[90,91] and erythroid cells of patients with SLE[92]. While mitochondrial dysfunction of erythroid cells was also attributed to defective mitophagy, the involvement of Rab4A and mTOR activation have not been addressed in this system[92].

Beyond trafficking organelles and intracellular proteins to lysosomes[93], a systematic analysis of surface receptors that recycle via endosomes unveiled a mosaic of Rab4A dependency. Canonical PDBu-induced internalization[27,94] and subsequent recycling and expression CD71 were enhanced by Rab4A in CD4$^+$ T cells of B6.TC/Rab4A$^{Q72L}$ mice (Fig. S18). Most recently, overexpression of CD71 was associated with enhanced iron uptake into CD4$^+$ lupus T cells[95], which might contribute to mitochondrial dysfunction in SLE independent of mTOR pathway activation. As unveiled by this study, the surface expression and traffic of CD98 followed a different pattern, as it failed to internalize after PDBu treatment, and lupus itself promoted the recycling and expres-sion of CD98 in B6.TC mice as compared to non-autoimmune B6 controls (Fig. S18). Expression of CD98 was synergistically promoted by Rab4A activation and SLE on all T cells of B6.TC/Rab4A$^{Q72L}$ mice; these coordinate changes were consistently reversed by the inactiva-tion of Rab4A in B6.TC/Rab4A$^{Q72L}$-KO mice. Increased expression of CD98 was mechanistically connected to a robust positive feed-back loop within a Rab4A-mTOR-CD98 axis in mice and patients with SLE. While Rab4A promoted mTOR activation and CD98 expression, mTOR blockade reduced the expression of Rab4A and CD98. Furthermore, knockdown of CD98 in primary human T cells blocked CD3/CD28-induced activation of mTORC1, suggesting that CD98 transmits critical signals required for mTOR activation and T-cell function. As revealed by this study, CD98 supports the uptake of branched-chain (VAL, LEU, ILE) and aromatic amino acids, TRP and its metabolite KYN. While KYN itself activated mTORC1 and mTORC2 in CD4$^+$ and CD8$^+$ T cells, it preferentially stimulated CD98 expression and depletion of CD8$^+$ T cells. Recently, KYN uptake was attributed to SLC7A5 (LAT1) and associated with mTORC1 activation in dendritic cells in SLE[96]. There-fore, it is conceivable that both the CD98 heavy chain (SLC3A2) and the LAT1 light chain (SLC7A5) of the heterodimer can independently reg-ulate KYN uptake. These findings support earlier observations that the accumulation of KYN may contribute to mTOR activation in patients with SLE[21].

mTORC1 and mTORC2 were activated in CD19$^+$, CD19$^+$CD38$^+$, and CD19$^+$CD11c$^+$ ABCs of B6.TC/Rab4A$^{Q72L}$ mice. These changes were all reversed by the inactivation of Rab4A in T cells of B6.TC/Rab4A$^{Q72L}$-KO mice. Rapamycin and NAC also reversed the activation of mTORC1 and mTORC2 and the expansion of ABCs in B6.TC/Rab4A$^{Q72L}$ mice. Of note, elevated production of ANA and aPL occurred with the accumulation of KYN in sera of B6.TC/Rab4A$^{Q72L}$ mice. KYN activated mTORC1 and mTORC2 and augmented the expression of CD98 in CD19$^+$, CD19$^+$CD38$^+$ B cells, CD19$^+$CD11c$^+$ ABCs, and CD138$^+$plasma cells. Moreover, KYN also expanded plasma cells with concurrent LPS sti-mulation. These findings suggest that population shifts in the B-cell compartment are driven by the accumulation of pro-inflammatory KYN metabolites in sera of B6.TC/Rab4A$^{Q72L}$ mice.

The overexpression of Rab4A preceded mTOR activation in sev-eral spontaneously lupus-prone mouse strains, including B6.TC, NZB/WF1, MRL, lpr, and MRL/lpr mice[13]. DN T cells, preceding the later stages of CD4$^+$CD8$^+$ DP and CD4$^+$ or CD8$^+$ SP T-cell development in the thymus, were depleted in B6/Rab4A$^{Q72L}$ females but expanded in B6.TC/Rab4A$^{Q72L}$ females along with coordinate skewing in activation of mTORC1 and mTORC2. These results suggest that Rab4A and lupus-driven inflammation exert distinct influences of T-cell development that warrant further studies. It has been firmly established that DN T cells may derive from activated CD8$^+$ T cells in patients[97] and mice with SLE[98,99]. Given that the expansion of DN T cells is mTOR-dependent[15,100–102], Rab4A-mediated mTOR activation may contribute to the depletion of CD8$^+$ T cells and reciprocal expansion of DN T cells in B6.TC/Rab4A$^{Q72L}$ mice. This mechanism is supported by the effective reversal of DN T-cell expansion in rapamycin-treated B6.TC/Rab4A$^{Q72L}$ mice. Our results also indicates the heterogeneity of DN T cells that have been considered both drivers[103] and inhibitors of renal inflammation[104]. 1) DN T cells were depleted in 20-week-old B6.TC/Rab4A$^{Q72L}$ mice before the onset of SLE, as shown in Fig. 2; however, DN T cells were expanded in B6.TC/Rab4A$^{Q72L}$ mice and depleted in B6.TC/Rab4A$^{Q72L}$-KO mice after the onset of SLE. 2) While rapamycin restrained DN T cells in B6.TC/Rab4A$^{Q72L}$ mice with therapeutic effi-cacy, DN T cells were expanded in rapamycin-treated B6.TC/Rab4A$^{Q72L}$-KO mice over rapamycin-treated B6.TC/Rab4A$^{Q72L}$ mice. These results reveal that the abundance of DN T cells is controlled by two different mechanisms in SLE: i) expansion via Rab4A-dependent mTOR activa-tion as noted in B6.TC/Rab4A$^{Q72L}$ mice; and ii) contraction via Rab4A-independent mTOR activation in B6.TC/Rab4A$^{Q72L}$-KO mice, both of which can be reversed by treatment with rapamycin. Therefore, these two types of DN T cells may play divergent roles in disease pathogenesis[103,104], which warrant further investigations.

Importantly, this study identifies Rab4A as a promoter of mTOR activation, ANA production, proteinuria, and GN during lupus patho-genesis in vivo. mTOR blockade by rapamycin or NAC reduced GN of B6.TC/Rab4A$^{Q72L}$ mice. These findings are consistent with the promis-ing efficacy of mTOR blockade in SLE patients[15,19], including those with GN[16,17,56,105]. Similar to SLE patients[15], rapamycin blocked thrombocy-topenia in B6.TC and B6.TC/Rab4A$^{Q72L}$ mice. Rapamycin increased hemoglobin in B6.TC/Rab4A$^{Q72L}$ and B6.TC/Rab4A$^{Q72L}$-KO mice, sug-gesting that mTOR activation contributed to anemia in SLE. NAC blocked lupus GN and thrombocytopenia in B6.TC/Rab4A$^{Q72L}$ mice, which have not yet been observed in human participants[19].

C alleles of the rs451401 single nucleotide polymorphism (SNP) in the HRES-1/Rab4 human genomic locus predispose to anti-DNA and aPL production and GN in patients with SLE[32]. This SNP has also been linked to another autoimmune disease, MS[33,84]. The predisposition of B6/Rab4A$^{Q72L}$ mice to MOG-induced EAE indicates that Rab4A activa-tion may broadly enhance CD4$^+$ T cell-dependent autoimmune dis-eases, including MS[106]. Another MS susceptibility gene[107], C-type lectin-like domain family 16 member A, CLEC16A, has been recently mapped to Rab4A$^+$ recycling endosome in human T cells[108]. rs451401 C alleles of the transcriptional enhancer facilitate the expression of Rab4A and mTOR activation both in healthy and SLE participants[35]. As docu-mented in this study, mTOR is trafficked by Rab4A to the lysosome,

where it gets activated within the cell[109,110]. However, mTOR activation and downstream lineage development also depend on Rab4A-directed traffic of surface receptors, such as CD98. In CD4$^+$, CD8$^+$, and DN T cells, Rab4A promoted the recycling of CD98 that transports KYN which activates mTOR. Along this line, Rab4A and mTOR-dependent production of KYN is identified as pro-inflammatory metabolite that may transmit activation signals from CD8$^+$ T cells to CD19$^+$ and CD19$^+$CD38$^+$ B cells and CD138$^+$ plasma cells. Thus, signal transduction along the Rab4A-CD98-KYN-mTOR axis is hereby established as a positive feed-back loop underlying pro-inflammatory lineage specification in the immune system with considerable impact on the development of SLE. The contribution of this axis is likely to be central for disease pathogenesis and offer regulatory checkpoints as targets for therapeutic interventions in autoimmune disease beyond SLE.

Overexpression of Rab4A has been detected in T cells of SLE patients as well as T cells, B cells, macrophages, thymocytes and hepatocytes in multiple lupus-prone mouse strains[13,111]. Therefore, we created B6/Rab4A$^{Q72L}$ and B6.TC/Rab4A$^{Q72L}$ mice to examine the global impact of Rab4A activation both in B6 control and lupus-prone B6.TC mice. Accelerated lupus in B6.TC/Rab4A$^{Q72L}$ mice is consistent with the notion that the genetic polymorphism, which influences lupus susceptibility[35], operates via activation of Rab4A. In accordance with the prominence of Rab4A overexpression in CD4$^+$ T cells of SLE patients[6], autoimmunity, lupus GN and EAE were all blocked in B6.TC/Rab4A$^{Q72L}$–KO mice lacking Rab4A in T cells. Since CD4 is expressed during the CD4$^+$CD8$^+$ double-positive stage of T-cell development[38], Rab4A was deleted both in CD4$^+$ and CD8$^+$ T cells. Although the results unveil a critical role of Rab4A in T cell development that controls the abundance of CD4$^+$ over CD8$^+$ T cells, its involvement in other relevant types of cells within and outside the immune system has not been evaluated[91,112]. In vivo and in vitro mechanistic studies identified the Rab4A-mTOR-CD98 positive feedback loop as a driver of pro-inflammatory lineage specification both in patients and mice with SLE. However, we have not individually targeted other trafficked receptors for impact on mitochondrial metabolism, mTOR activation, and disease pathogenesis. We identified increased production of KYN as a central metabolite that may spread mTOR activation and inflammation through the bloodstream. Although KYN accumulation in CD8$^+$ T cells and sera of B6.TC/Rab4A$^{Q72L}$ mice occur with increased demand for pyridine nucleotides during lupus pathogenesis, all of which are mTOR-dependent, further mechanistic studies that connect KYN metabolism with mitochondrial dysfunction and metabolic flux through the TCA cycle seem warranted. These findings and limitations open up unexplored avenues for investigating metabolic control of immune cell development and the pathogenesis of autoimmunity.

## Methods

These studies were conducted in compliance with all ethical regulations and study protocols approved by Institutional Review Boards of the State University of New York Upstate Medical University. The biomarker studies of human participants were conducted in the setting of a clinical trial (NCT00779194)[15].

### Mice

Autoimmunity-resistant C57BL/6 (B6) and lupus-prone B6.Sle1.2.3 triple congenic (B6.TC) mice[113] were obtained from Jackson Laboratory (Bar Harbor, ME). Since *Rab4A* is overexpressed in T cells of SLE patients[6] as well as T and B cells of MRL, lpr, NZB/W(F1), and B6.TC mice prior to the onset of ANA production or any sign of disease, we created mice with constitutively active *Rab4A*. We replaced exons 3 of the *Rab4A* genomic lupus in 129SvJ-derived TC1 embryonic stem (ES) cells using a targeting vector with 5 kb upstream and 3 kb downstream genomic fragments harboring exon 2 and exons 4 and 5 respectively, of the genomic locus (Figure S1). Intron 3 contained a neomycin resistance cassette flanked by Frt recognition motifs. Intron 2 and

intron 3 downstream of the neomycin cassette harbored LoxP sites to allow the removal of exon 3 by crossing with transgenic mice expressing Cre recombinase (Figure S1A). Exon 3 was altered via site-directed A → T mutagenesis of codon CAG to CTG thus replacing amino acid Q$^{72}$ with L$^{72}$ (Fig. S1B, C). The targeting vector was electroporated into 129SvJ-derived TC1 ES cells. ES cells heterozygous for the disrupted allele were selected in culture medium supplemented with neomycin and ganciclovir. ES cells with targeted alleles were identified by PCR (Fig. S1A) and microinjected into C57BL/6 blastocysts to generate chimeric mice. Chimeric males were mated with C57Bl/6 females, and tail DNA of the offspring was tested for transmission of the targeted allele by PCR and sequencing of genomic DNA (Fig. 1C). The neomycin resistance cassette was removed by mating with *Rosa26-flp* recombinase knock-in mice[114]. Rab4A$^{Q72L}$ heterozygotes were back-crossed onto the C57Bl/6 strain for ten generations, as earlier described[115]. Heterozygotes were bred and wildtype (WT), heterozygote, and homozygote mice with floxed *Rab4A*$^{Q72L}$ alleles were tracked by PCR genotyping (Fig. 1D). Homozygote Rab4A$^{Q72L}$ mice were mated with heterozygote CD4$^{Cre}$ transgenic mice to generate offspring lacking Rab4A expression in T cells (Fig. 1E). Expression of Rab4A was absent in CD4$^+$ and CD8$^+$ T cells of Rab4A$^{Q72L}$-KO mice.

Rab4A$^{Q72L}$ mice have constitutive activation of *Rab4A* due to the elimination of GTPase activity that locks the Rab4A in a GTP-bound or active state. The mutation is ubiquitously expressed throughout the mice. The Rab4A$^{Q72L}$–KO mice are the crossbreed of the floxed Rab4A$^{Q72L}$ mice with CD4$^{Cre}$ mice where the *Cre* expression is dependent on the expression of CD4, which confers deletion of floxed alleles at the CD4$^+$CD8$^+$ double-positive state in the thymus[38]. These results in the deletion of *Rab4A* in T cells of Rab4A$^{Q72L/CD4Cre}$ mice, termed Rab4A$^{Q72L}$-KO mice (Table S1).

To evaluate the role of Rab4A in lupus pathogenesis, Rab4A$^{Q72L}$, Rab4A$^{Q72L}$-KO and CD4$^{Cre}$ control mice were crossed with the B6.TC strain. The retention of three NZM2410-derived lupus susceptibility intervals (*Sle1*: chr1, ~76 cM; *Sle2*:chr4, ~39 cM; and *Sle3*:chr7, ~15 cM) have been confirmed in collaboration with Dr. Laurence Morel and monitored through each generation, as earlier described[113].

Animal experimentation has been approved by the Committee on the Human Use of Animals in accordance with NIH Guide for the Care and Use of Laboratory Animals. Fatal Plus (390 mg/ml pentobarbital sodium, 0.01 mg/ml propylene glycol, 0.29 mg/ml ethyl alcohol, 0.2 mg/ml benzyl alcohol preservative) was provided by the Upstate Department of Laboratory Animal Resources. Tenfold diluted Fatal Plus (100 μl per 20 g mouse) was used prior to perfusion with 2% paraformaldehyde and 1% glutaraldehyde in PBS for histological studies of the spinal cord. Cervical dislocation was used for metabolic studies to be performed in vitro. This method of euthanasia is necessary to prevent $CO_2$ inhalation-induced respiratory acidosis or pentobarbital-induced metabolic acidosis[116]. $CO_2$ or Fatal Plus administration would result in acidosis that could compromise the assessment of oxidative stress which is associated intracellular acidosis[115]. $CO_2$ inhalation was used for euthanasia of moribund animals.

### Flow cytometry of human peripheral blood lymphocytes (PBL)

We examined unstimulated cells and cells stimulated with CD3/CD28 for 16 h[15]. T-cell subsets were analyzed by staining with antibodies to CD4, CD8, CD25, CD27, CD197, CD98, CD45RA, CD45, and CD62L. For detection of mTOR activity, cells were permeabilized with Cytofix/CytopermPlus (eBiosciences) and stained with AlexaFluor-488-conjugated antibody to pS6RP (Cell Signaling; Beverly, MA; Cat. No. 4851). Each patient's cells were freshly isolated, stained and analyzed in parallel with a matched control. Mean channel fluorescence intensity (MFI) values of patient samples were normalized to controls set at 1.0 for each analysis and expressed as fold changes. Frequencies of cell populations were compared as absolute values. Relative fluorescence intensity (RFI) was calculated by comparison of MFI values of patients'

cells to healthy participants' cells, which were analyzed in parallel and normalized to 1.0.

## Transfection of PBL with siRNA

$10^7$ freshly isolated lymphocytes were electroporated with 300 pmol of siRNA specific for SLC3A2 (Qiagen Hs_Slc3a2_1 Cat No. S100301826; 5′-AATCCTGAGCCTACTCGAATC-3′) or scrambled control siRNA (Qiagen Hs_Slc3a2_1 Cat No. 1027293) in 100 μl of nucleofection solution using the transfection protocol U-014 for primary human T cells (Amaxa, Gaithersburg, MD). SLC3A2 and scrambled siRNA were conjugated with 3′Alexafluor-647 and 3′Alexafluor-546. Expression of CD98 and mTORC1 were assayed 48 h after transfection by flow cytometry.

## Treatment of mice with rapamycin (Rapa) and N-acetylcysteine (NAC)

Female lupus-prone mice were treated with rapamycin (Biotica, Cambridge, UK), solvent control (carboxymethylcellulose (CMC, Millipore Sigma Cat No C5013), or NAC (Spectrum Chemical Company, New Brunswick, NJ) for 12 weeks beginning at $27 \pm 1.4$ weeks of age. Fresh rapamycin solution was prepared daily by making a 10 mg/ml stock solution in dimethyl sulfoxide (DMSO, Millipore Sigma Cat No D2650) and making final solution in phosphate-buffered saline (PBS) with 0.2% CMC warmed to 37 °C and mixed thoroughly by vortexing. Rapamycin solution is an emulsion that precipitates out of solution when cooled, so the solution was prepared immediately and kept at 37 °C before injection. 3 mg/kg rapamycin was administered in CMC solvent intraperitoneally (ip) in the left lower quadrant of the abdomen three times weekly, while 10 g/l of NAC was provided in drinking water. Control mice were treated ip three times weekly with 0.2% CMC solvent control alone. Age-matched female B6.TC and B6.TC/Rab4A$^{Q72L}$-KO mice were also treated with rapamycin or solvent control.

## Disease monitoring in spontaneously lupus-prone B6.TC mice

Development of nephritis was monthly monitored by measurement of proteinuria from >4 weeks of age through >50 weeks of age. Urine was collected from the bottom of cages after housing mice in individual cages without bedding for 4 hours. Urine protein was measured by Bradford protein assay (Bio-Rad, catalog #500-006) using 5 μl of urine. In parallel, serum was collected by submandibular bleeding for measurement anti-nuclear auto-antibodies (ANA) and antiphospholipid autoantibodies (aPL). To assess for potential treatment toxicities, mice were weighed weekly.

## Renal pathology

Kidney tissues were fixed in 10% formalin. Samples were paraffin-embedded, sectioned, and stained with Periodic Acid-Schiff (PAS) and hematoxylin dyes. Histology was assessed by scoring for glomerulonephritis (GN), glomerulosclerosis (GS), interstitial nephritis (IN) on a 0–4 scale, as well as determining the percentage of sclerotic and crescentic glomeruli[13,117]. Slides were scored independently by an expert pathologist (MH) blinded to genotype and treatment groups.

## Induction of MOG 35-55 peptide-induced EAE

On Day 0, ~12-week-old female mice were immunized subcutaneously with 200 μg of MOG 35-55 peptide (MEVGWYRSPFSRVVHLYRNGK)[118] and 400 μg of Mycobacterium tuberculosis H37Ra (MTb, Difco, Detroit, MI). Incomplete Freund's Adjuvant (IFA; Gibco, Grand Island, NY) and dried, heat-killed MTb were emulsified with an equal volume of MOG 35-55 dissolved in PBS using a VirTishear Cyclone homogenizer with a 10 mm micro shaft impeller (VirTis, Gardiner, NY). The mice were immunized with 150 μl of the MOG-containing emulsion divided among three subcutaneous sites on the flank (between the shoulder blades and over each hip). Additionally, on Days 0 and 2, the mice were injected intraperitoneally with 200 ng pertussis toxin (PTx) (List Biological Laboratories, Campbell, CA) in 200 μl PBS[119].

## Clinical evaluation of EAE

Individual animals were observed daily and clinical scores were assessed on a 0–5 scale as follows: decreased tail tone: 0.5. limp tail: 1, limp tail and hind limb weakness: 1.5, waddling gait with limp tail (ataxia): 2 ataxia with partial limb paralysis (legs slip through cage top): 2,5, ataxia with full hind limb paralysis: 3 hind limb paralysis, and forelimb weakness: 3.5, full paralysis of hind limbs and partial paralysis of forelimbs: 4, moribund: 4.5, and death: 5[119]. The animal protocol was approved by the Collaborative Institutional Training Initiative (CITI) Animal Care and Use Working Group. Data are reported as the mean clinical score ± SEM for all animals in a particular group. Mice were age and sex-matched for all experiments. Significance of clinical score of experimental over control mice was assessed by the Student's $t$ test.

## Histologic evaluation of EAE

Mice were anesthetized with 3 mg of nembutal, harvested for lymph nodes and spleen, and sacrificed by total body perfusion intracardially through the left ventricle with 25 ml of 10% formalin in 1x PBS. Brain and spinal cord were dissected out and fixed in 10% formalin. 1–2 mm thick transverse serial segments were taken from the cerebral hemispheres, cerebellum/brainstem and spinal cord cervical, thoracic, lumbar and sacral regions for embedding in paraffin. Paraffin sections were stained with hematoxylin/eosin and analyzed at ×400 and ×1000 magnification for the presence of inflammatory lesions with infiltrating mononuclear cells in the meninges and parenchyma and for the presence of demyelination[120].

## Analysis of ANA and aPL by ELISA

For measurement of ANA, nuclear chromatin extract was prepared from chicken red blood cells[121], which were pelleted at $1500 \times g$ for 10 min and washed twice with Buffer A (80 mM NaCl, 20 mM EDTA, and 20 mM Tris-HCl pH7.5). Erythrocytes were then resuspended in buffer A with 1.5% Triton X, mixed, then placed on ice for 10 minutes and repeated once more after centrifugation at $1500 \times g$ for 10 min. Erythrocytes were then resuspended in buffer A plus 0.25 M sucrose and pelleted at $1000 \times g$ for 10 min. The pellet was washed twice with buffer A. The final pellet was resuspended in 10 mL of 10 mM EDTA pH 8.0 and sonicated using a Fisher Model 100 Sonic Dismembrator (Fisher Scientific, Pittsburgh, PA) at 10 Watts for 60 s, cooled on ice for 1 min the repeated three more times. Protein concentration was estimated at $OD_{280}$, with 1.0 $OD_{280}$ equal to 1.42 mg of protein. For ELISA assay, flat-bottom 96-well polystyrene plates were coated with nuclear extract (50 μg/well), cardiolipin (100 ng/well, Sigma cat. no. c1649), beta-2 glycoprotein I ($\beta_2$GPI or apolipoprotein H, 100 pg/well, R&D Systems Cat. No. 6575-AH-050), phosphatidylethanolamine (PE, 100 ng/well; Sigma cat. no. P7693), or phosphatidylserine (PS, 100 ng/well; Sigma cat.no. P7769) in 0.01 M NaHCO3 (pH 9.55) overnight[13,122]. After coating, plates were washed six times with 0.1% Tween-20 in PBS (Tween-20/PBS), the plates were blocked for 1 h at room temp in 10% normal goat serum in Tween-20/PBS. Antibodies were measured in 1 μl of mouse serum which was diluted 100-fold in PBS with 10% normal goat serum and 0.1% Tween-20 (Tween-20/PBS) and incubated on antigen-coated ELISA plates at room temperature for 1 h. Then, plates were washed six times with 0.1% Tween-20/PBS, and incubated with 2000-fold diluted, HRP-conjugated secondary goat antibody directed against mouse IgG (heavy and light chain) from Jackson Immuno Research Laboratories s (Cat. no. 115-035-146). After washing six times with 0.1% Tween-20/PBS, plates were developed with 3,3′,5,5′-tetramethylbenzidine (TMD, Alpha Diagnostic International, Cat No 5210) and optical density (OD) was read at 450 nm and 630 nm using a Biotek Synergy II plate reader equipped with Gen5 software. Absorbances of 630 nm were subtracted from the 450 nm measurement for background reduction. In each experiment negative and positive control sera from B6 and MRL/lpr mice were used as negative and positive controls, as earlier described[13,111].

## Isolation of splenocyte subsets

Single-cell suspension was made by crushing a freshly isolated spleen through a 70-μm mesh filter (Corning Cat 431751) in a sterile Petri dish containing 10 mL of complete RPMI 1640 culture medium with 10% of fetal calf serum (FCS), 100 U/mL penicillin, 100 μg/mL streptomycin, 10 μg/mL amphotericin B, 2 mM L-glutamine, 1 mM sodium pyruvate, 50 μM β-mercapto-ethanol, and 10 mM HEPES. The suspension was washed once in 10 ml of culture medium and stored on ice until all mice were harvested to be processed together. The cell suspensions were spun down at $300 \times g$ for 8 min at 4 °C. The supernatant were removed, and the pelleted cells were resuspended in 5 mL of ACK (150 mM $NH_4Cl$, 10 mM $KHCO_3$, 0.1 mM EDTA, pH 7.4) lysis buffer for 5 min, after which the solution was diluted with 5 mL of complete media and spun down at $300 \times G$ for 8 min at 4 °C. The resulting pellet was resuspended in isolation buffer ($Ca^{2+}$ and $Mg^{2+}$ free PBS supplemented with 2% fetal bovine serum and 2 mM EDTA, pH 7.4) and processed for flow cytometry or further isolation of splenocyte subsets. Cell isolations were carried out in the following sequence: i) positive CD4 isolation (ThermoFisher Scientific Cat 11461D), ii) positive CD8 positive isolation (ThermoFisher Cat No 11462D), iii) B cell negative isolation (ThermoFisher Cat No11422D).

## Cell culture

Using tissue culture plates pre-coated overnight at 4 °C with 5 μg/ml of anti-CD3 (BioLegend Cat 100253), CD4$^+$ and CD8$^+$ T cells were cultured in complete medium containing 0.5 μg/ml soluble anti-CD28 (BioLegend Cat 102121) at 37 °C. B cells were stimulated with 10 μg/ml of lipopolysaccharide (LPS, Sigma Cat No L2630). Splenocytes were cultured with kynurenine (KYN) at or below concentrations previously employed to activate mTOR[21].

## Complete blood counts (CBC)

Blood was collected from the submandibular vein. 100 μl blood was transferred into Eppendorf tubes containing 2 μl of 0.5 M EGTA for anticoagulation. The blood was stored on ice, and 20 μl of each blood sample was analyzed within 1 hour using an automated Hemavet 950 instrument (Drew Scientific, Miami Lakes, FL).

## Flow cytometry

Surface receptors on live cells were stained with fluorochrome-conjugated antibodies described in Table S2. Parallel detection of mitochondrial mass and potential was carried out by incubating surface-stained cells with fluorescent molecular probes (see below). Intracellular antigens, such as mTOR substrates pS6RP and pAkt, and transcription factors FoxP3 and Helios, were detected following permeabilization and fixation using FixPerm kit (eBioscience Cat Nos 00-5123-43, 005223-56, 008333-56). In agreement with a recent study[123], we have noted far greater sensitivity and accuracy and less variability of flow cytometry as compared to western blot detection of mTORC1 and mTORC2 activities[13,15,100–102]. Intracellular cytokine production was assessed following 3 h stimulation with phorbol 12-myristate 13-acetate (PMA, 5 ng/ml; Sigma, cat. no. P-8139) and ionomycin (500 ng/ml; Sigma, cat. no. I-0634) in the presence of brefeldin A (10 ng/ml, Sigma cat no. B6542). Subsequently, cells were i) stained with antibodies directed to surface antigens; ii) fixed in 1% paraformaldehyde; iii) permeabilized with the eBioscience FixPerm kit; and iv) stained with antibodies directed to cytokines (Table S2).

## Receptor recycling

Freshly isolated splenocytes were stained with fluorochrome-conjugated antibodies directed to surface receptors CD71[6,27], CD38[59,60], CD68[61,62], CD98[63,64], SERT[65,66], and CD152 which are regulated through endocytic recycling[67,68], as well as CD3, CD4, and CD8. Freshly stained cells were kept at 4 °C and used reference point for flow cytometry Unstained splenocytes were stimulated with 100 nM PDBu (Millipore Sigma, Cat. No. P1269) at 37 °C for 1 h to initiate receptor internalization[27]. PDBu was removed by washing three times in 1 ml of ice-cold PBS. To inhibit de novo protein synthesis, cycloheximide (50 μg/ml, Sigma-Aldrich cat no O1810) was added to the cells, and they were then incubated at 4 °C for 10 min. An aliquot was kept on ice, and the rest of the cells were incubated at 37 °C to allow recycling for 30 min, 60 min, and 120 min. Subsequently, the cells were stained for the expression of surface antigens 4 °C for 30 min and analyzed by flow cytometry.

## Flow cytometric analysis of mitochondrial transmembrane potential ($\Delta\Psi_m$) and mitochondrial mass

$\Delta\Psi_m$ was estimated by staining for 15 min at 37 °C in the dark with 1 μM tetramethylrhodamine methyl ester (TMRM, ThermoFisher Cat No T668, excitation: 543 nm, emission: 567 nm recorded in FL-2) and 20 nm 3,3′-dihexyloxacarbocyanine iodide (DiOC$_6$, Thermo-Fisher Cat No D273, excitation: 488 nm, emission: 525 nm recorded in FL-1). Co-treatment with a protonophore, 5 μM carbonyl cyanide m-chlorophenylhydrazone (mClCCP, Sigma-Aldrich Cat No 215911) for 15 min at 37 °C resulted in decreased TMRM and DiOC$_6$ fluorescence and served as a positive control for disruption of $\Delta\Psi m$[124]. Mitochondrial mass was monitored by staining with 100 nM Mito-Tracker Green-FM (MTG, ThermoFisher Cat No M7514, excitation: 490 nm, emission: 516 nm recorded in FL-1) or 100 nM MitoTracker Deep Red (MTDR, ThermoFisher Cat No M22426, MTDR, excitation: 644 nm, emission: 665 nm). Samples were analyzed using a Becton Dickinson LSRII flow cytometer equipped with 20 mW solid-state Ng-YAG (emission at 355 nm), 20 mW argon (emission at 488 nm), 10 mW diode-pumped solid-state yellow-green (emission at 535 nm), and 16 mW helium-neon lasers (emission at 634 nm). Data were analyzed with Flow Jo software (Version 10, TreeStar Corporation, Ashland, OR). Dead cells and debris were excluded from the analysis by electronic gating of forward (FSC) and side scatter (SSC) measurements. Each measurement was carried out on ≥10,000 cells. In each experiment, freshly isolated cells from control and lupus-prone mice were analyzed in parallel. Detailed protocols for assessment of mitochondrial dysfunction in SLE have been earlier described[125].

## Measurement of mitochondrial electron transport chain (ETC) activity and glycolysis in live cells

After 3 days of CD3/CD28 stimulation, CD4$^+$ T cells and CD8$^+$ T cells were seeded at $4 \times 10^5$ cells/well in 40 μl volume of either lymphocyte glycolysis medium (Seahorse XF Base Medium (Catalog No. 102353-100, Seahorse Bioscience and 2 mM glutamine) or mitochondrial stress test medium (Seahorse XF Base Medium, 10 mM glucose, 2 mM glutamine,1 mM sodium pyruvate) in XF 96-well culture plates coated with 450 ng/well Cell-Tak (Catalog No. 354240, Corning). The plates were centrifuged at $300 \times g$ for 10 s with no brake to accelerate cell adherence to the plate. The cells were then equilibrated at 37 °C in ambient atmosphere (no $CO_2$) for 30 min. After the cells had fully adhered, 135 μl of media was added to each well to bring the total volume up to 175 μl. The plates were equilibrated for another 15 min at 37 °C and then loaded onto the Seahorse XFe96 Analyzer (North Billerica, MA). For the glycolysis assay, we injected 10 mM glucose, 1 μM oligomycin, and 50 mM 2-deoxyglucose. For the mitochondrial stress test, we injected 1 μM oligomycin, 0.5 μM carbonyl cyanide-4- trifluoromethoxyphenylhydrazone (FCCP), and 500 nM rotenone/antimycin A. Measurements were performed in 3–5 replicates and their means were used as the result for each experiment[126].

## Confocal immunofluorescence microscopy

For labeling and tracking of mTOR to organelles, Jurkat cell lines were incubated at 37 °C with 5 μM Lysotracker Red (catalog no. L-7528; Invitrogen) for 30 min in complete RPMI 1640 medium. Cells were

washed twice and resuspended in RPMI 1640 medium followed by pipetting onto poly-L-lysine-coated (0.1 mg/ml poly-L-lysine; Sigma-Aldrich Cat No A-005-M) coverslips for 10 min at room temperature. The cells were fixed in a 4% paraformaldehyde-PBS solution for 15 min and permeabilized with a 0.1% saponin/1% FBS/HBSS mixture for 30 min. Normal goat serum (10%) was used to block the cells followed by mTOR primary antibody (catalog no. 2983 S; Cell signaling) staining. Anti-mTOR antibody was directly conjugated with Alexa Flour 405 using Zenon Alexa Fluor 405 kit (ThermoFisher Cat No. Z25313). Primary antibody conjugated with Alexa Flour 405 was directly applied to the permeabilized cells for 45 min. Cells were washed twice with permeabilization buffer and twice with HBSS, mounted on slides and visualized with a Zeiss LSM 780 inverted laser scanning confocal microscope (Carl Zeiss, Oberkochen, Germany) using a 40×/1.3 Plan-NeoFluar oil-immersion objective at 0.45 µm z-step intervals with lateral pixel dimensions of 0.22 µm. Images were acquired with transmission photomultiplier tube detector. Signal intensity gain was calibrated on cells that were unstained. Sequential scanning was used to record GFP (excitation: 395 nm, emission: 509 nm), mTOR-Alexa647 (excitation: 650 nm, emission: 670 nm), and LTR (excitation: 577 nm, emission: 590 nm); the RGB images were converted to 8-bit grayscales and pseudo-colored in green, blue, and red, respectively. Captured z-series were imported and analyzed using Image J. Mean intensity values were taken in each channels for each pixel, as earlier described[127,128]. Manders' overlap coefficient was obtained using JACOP plugin (available at http://rsb.info.nih.gov/ij/plugins/track/jacop.html). Colocalized signal between mTOR and LTR was quantified and divided by the mean value of the corresponding channel to determine colocalization ratio[129]. Colocalization between mTOR and LTR was assessed using an in-house developed macro. Briefly, the FIJI macro converted the red and green channel to binary images followed by calculating the total number of overlapping pixels that could not be spatially resolved between the two channels in each z-slice for the entire z-series. Sub-resolution beads were imaged with the same acquisition parameters and used to collect z-series to validate the FIJI-macro[130].

### Western blot analyses

Liver tissue protein lysates were prepared by sonication using a Fisher Model 100 Sonic Dismembrator (Fisher Scientific, Pittsburgh, PA) in 300 µl of 1 x cell lysis buffer (20 mM Tris-HCl, 150 mM NaCl, 1 mM $Na_2$EDTA, 1 mM EGTA, 1% Triton, 2.5 mM sodium pyrophosphate, 1 mM β-glycerophosphate, 1 mM $Na_3VO_4$, 1 µg/mL leupeptin; catalog no. 9803, Cell Signaling Technology, Danvers, MA) with 1 mM phenylmethanesulfonyl fluoride (PMSF) (catalog no. P7626, Sigma-Aldrich). Splenocytes, CD4+ T cells, CD8+ T cells, and B cells were lysed via pipette with 1x cell lysis buffer with 1 mM PMSF. 40 µg of protein lysates, unless otherwise indicated, were resuspended in Laemmli buffer (20% glycerol, 125 mM Tris-HCl, 4% sodium dodecyl sulfate, 10% β-mercaptoethanol, 0.075% bromophenol blue)[131], separated in a 12% SDS-polyacrylamide gel electrophoreses (SDS-PAGE), and electro-blotted to nitrocellulose. Rabbit monoclonal Rab4A antibody was obtained from Abcam (Cat No. ab108974; Cambridge, MA). Mouse monoclonal p70S6 kinase (p70S6K) (Catalog No. sc-8418), and mouse monoclonal phospho-p70S6K antibodies (Catalog No. sc-8416) were obtained from Santa Cruz Biotechnology (Santa Cruz, CA). Rabbit monoclonal p70S6K (Catalog No. 2708), AKT (Catalog No. 4685), rabbit polyclonal phospho-p70S6K (Catalog No. 9205), and phospho-AKT (Catalog No. 4058) were purchased from Cell Signaling Technology. Drp1 was detected with phospho-DRP1 (Ser637) (D3A4) Rabbit mAb #6319, phospho-DRP1 (Ser616) (D9A1) Rabbit mAb #4494, and DRP1 (D6C7) Rabbit mAb #8570 from Cell Signaling Technology (Beverly, MA). β-actin mouse monoclonal antibody (Catalog No. Mab1501R) and anti-Cre recombinase rabbit polyclonal antibody were obtained from Sigma Millipore (Burlington, MA, Cat No 69050-3). A complete list of antibodies with dilutions employed for western blots is provided in Supplementary Table S3.

### Metabolome analysis by LC-MS/MS

$3 \times 10^6$ million cells were washed in 1 ml of PBS, pelleted at 1500 rcf in an Eppendorf centrifuge at 4 °C, and resuspended in 100 µl of 80% methanol (−80 °C) and 10 µl of 0.3 mM 5-thio-glucose, used as internal standard to allow correction for sample recovery. After freezing at −80 °C and thawing once, the sample was centrifuged at 13,000 × g for 30 min at 4 °C, and 100 µl of supernatant was saved. A 2nd aliquot of 100 µl of 80% methanol (−80 °C) was added to the pellet, the sample was vortexed, centrifuged at 13,000 × g for 30 min at 4 °C, and the 2nd 100 µl of supernatant was saved. The two 100-µl supernatants were combined, dried in a SpeedVac (Savant AS160, Farmingdale, NY), and stored −80 °C until analysis. Each sample was resuspended in 20 µl of LC/MS grade water, and 10 µl per sample was injected into a Thermo Scientific Vanquish HPLC coupled to a Thermo Scientific Q Exactive hybrid quadrupole-orbitrap MS. The metabolites were separated using a hydrophilic interaction liquid-chromatography (HILIC) method on a Waters Xbridge BEH Amide column (3.5 µm, 2.1 × 100 mm, P/N: 186004860) kept at 30 °C during the analysis. Mobile phase component A was 10 mM ammonium acetate and 7.5 mM ammonium hydroxide in water with 3 % (v/v) acetonitrile (pH 9.0) while mobile phase component B was 100% acetonitrile. The 25-min-long gradient was as follows: 0 min, 85% B; 1.5 min, 85% B; 5.5 min, 35% B; 14.5 min, 35% B; 15.0 min, 85 % B; 25.0 min, 85% B. The mobile phase flow rate was the following: 0 minutes, 0.150 ml/min; 10.0 minutes, 0.150 ml/min; 10.5 minutes, 0.300 ml/min; 14.5 minutes, 0.300 ml/min; 15.0 minutes, 0.150 ml/min, 25.0 minutes, 0.150 ml/min. The sample injection volume was 10 µL. The Thermo Scientific Q Exactive MS was operated in polarity switching mode throughout the acquisition run to maximize metabolite coverage. The heated electrospray ionization (HESI) probe parameters were the following in both polarity modes: sheath gas flow rate 30, aux gas flow rate 10, sweep gas flow rate 0, spray voltage 3.60 kV, aux gas heater temp 120 °C, S-lens RF level 55, ion transfer capillary temp 320 °C. In positive mode, the instrument acquired Full Scan spectra with a m/z range of 61-915, while in negative mode, the Full Scan mass range was m/z 70-920. The resolution of the scans was 70000, the AGC target was 3e6, while maximum IT was 200 ms. Mass spectroscopy data were collected with the following softwares: Thermo Q Exactive Tune version 2.9, Thermo TraceFinder version 4.1, and Thermo Scientific Xcalibur version 4.1. We used a quantitative polar metabolomics profiling platform with selected reaction monitoring (SRM) that covers all major metabolic pathways to confirm the identity of targeted metabolites. The platform uses hydrophilic interaction liquid chromatography with positive/negative ion switching to analyze ∼ 500 metabolites from a single 25-min acquisition run with a 3-ms dwell time and a 1.55-s duty cycle time (Supplementary Table S4).

### Metabolite steady-state, pathway, and flux analyses

Quantitative enrichment of detected metabolites was utilized for pathway analysis employing the web-based MetaboAnalyst 5.0 software[132]. Samples from compared mice were matched for age and gender and were injected in the same LC-MS/MS run. The signal stability was assured by normalizing the controls between runs to the sum of all signals between separate runs using Metaboanalyst. The enrichment analysis was based on global analysis of covariance (Ancova). A Google-map style interactive visualization system was utilized for data exploration and creation of a 3-level graphical output: metabolome view, pathway view, and compound view. The "metabolome view" shows all metabolic pathways arranged according to the scores from enrichment analysis (y axis: −log p) and from topology analysis (x axis: impact: number of detected metabolites with significant p value). The pathway topology analysis used two well-

established node centrality measures to estimate node importance: degree centrality and betweenness centrality. Degree centrality depends on the number of links connected to a given node. For directed pathway graphs, there are two types of degrees: in-degree for links came from other nodes, and out-degree for links initiated from the current node. Here, we only considered the out-degree for node importance measure. Upstream nodes are considered to have regulatory roles for the downstream nodes, and not vice versa. The betweenness centrality measures the number of shortest paths going through the node. Since metabolic networks are directed, we used relative-betweenness centrality for a metabolite importance measure based on metabolite topology weighed by relative-betweenness centrality[132]. The degree centrality measures focus more on local connectivity, while the betweenness centrality measures focus more on global network topology. The node importance values calculated from centrality measures were further normalized by the sum of the importance of the pathway. Therefore, the total/maximum importance of each pathway reflects the importance measure of each metabolite node that is actually the percentage relative to the total pathway importance, and the pathway impact value is the cumulative percentage from the matched metabolite nodes. The altered compounds have been grouped and presented together for each pathway.

Metabolite concentrations were evaluated for their ability to discriminate between WT, Rab4AQ72L and Rab4AQ72L-KO mice by partial least squares-discriminant analysis (PLS-DA)[133]. Contribution of individual metabolites to PLS-DA was assessed by variable importance in projection (VIP) and coefficient scores. Individual compounds were also compared between B6 and lupus-prone mice by 2-way ANOVA paired and Tukey's correction for multiple comparisons using Prism Software Version 10 (GraphPad, San Diego, CA).

For metabolic flux analyses, stable isotope-labeled compounds [1,2-$^{13}$C]- glucose (CLM-504-0.5), [U-$^{13}$C]-glucose (CLM-1396-0.5), [U-$^{13}$C]-glutamine (CLM-1822-H-PK), [2-$^{2}$H]-glucose (DLM-1271-0.5), [3-$^{2}$H]-glucose (DLM-3557-PK), [4-$^{2}$H]-glucose (DLM-9294-PK), [11-$^{13}$C]-tryptophan (CLM-4290-H-PK), [10-$^{13}$C]-kynurenine (CLM-9884-PK), and [1,2,3-$^{13}$C]-3-hydroxykynurenine (CNLM-10399-PK) were obtained from Cambridge Isotope Laboratories (Cambridge, MA) and [1-$^{2}$H]-glucose was obtained from Omicron Biochemicals (South Bend IN, Cat No GLC-032). Following CD3/CD28 stimulation for 72 h, the culture medium was replaced with glucose-free medium supplemented with $^{13}$C or $^{2}$H stable isotope labeled glucose or glutamine-free medium replaced with $^{13}$C-labeled glutamine for 30 min. Metabolites of stable isotope labeled compounds were tracked through the mitochondrial TCA cycle, glycolysis, and the PPP (Fig. S13)[126]. % enrichment of labeled metabolites was compared to unlabeled compounds.

## RNA sequencing
RNA was extracted from $3 \times 10^{6}$ CD4$^{+}$ or CD8$^{+}$ T cells after 3-day stimulation with CD3/CD28 using the Qiagen miRNeasy kit (Qiagen, Hilden, Germany). RNA quality and quantity were determined using the RNA 6000 Nano Kit on the Agilent 2100 Bioanalyzer (Agilent, Santa Clara, CA). Sequencing library preparation was done with the Illumina TruSeq Stranded Total RNA with RiboZero Gold kit (Illumina, San Diego, CA). Sequencing libraries were quantified using the KAPA Library Quantification Complete Kit Universal (KAPA Biosystems, Wilmington, MA). The pooled library (1.4pM) was loaded onto the NextSeq 500 instrument, using the NextSeq 500/550 High Output v2 Kit for 75 cycles (Illumina, San Diego, CA). RNAseq data have been securely transferred, stored, and analyzed in the Illumina BaseSpace Sequence Hub. All RNA-sequencing data have been uploaded to the NCBI Gene Expression Omnibus (Accession GSE245413). Express software was used to assign aligned reads to genes and perform differential gene expression analysis. Cufflinks software was used to profile gene expression and to detect transcript isoforms. Leveraging KEGG, Ingenuity, and Panther GeneOntology databases, integrated analysis of

individually matched metabolome and RNAseq results was carried out with MetaboAnalyst 5.0. Pathway analyses were performed using the Partek Bioinformatic server (St. Louis, MO).

## Statistical analysis and reproducibility
Statistical analyses were performed using the GraphPad Prism 5.0 Software (San Diego, CA). Data were expressed as the mean ± standard error of the mean (SEM) of individual experiments. Pair-wise repeated measures analysis of variance (ANOVA), two-way ANOVA, and Student's $t$ tests were used for analysis of results. For hypothesis testing, changes were considered significant at $p$ value < 0.05. 4 digits after the decimal point over the connecting bars are displayed between groups compared. Broken connecting lines are displayed for two-tailed $p$ values between 0.05 and 0.1; which reflect one-tailed significance. The absence of connecting bars reflects $p > 0.05$. Figures and all panels source data files include raw data and 4-digit exact $p$ values obtained by ANOVA. Two-way ANOVA was used when comparing WT, Rab4A$^{Q72L}$ and Rab4A$^{Q72L}$-KO mice between control B6 and lupus-prone B6.TC strains. One-way ANOVA was used when comparing WT, Rab4A$^{Q72L}$ and Rab4A$^{Q72L}$-KO mice within control B6 or lupus-prone B6.TC strains. Three-way ANOVA was performed to address the impact of three independent variables in Fig. 6B: 1) CD4 versus CD8; 2) Control versus KYN stimulation; and 3) Control versus CD3/CD28 co-stimulation. Post-hoc test $p$ values displayed in figures have been corrected for multiple comparisons via the recommended Tukey or Sidak methods[134-137] in GraphPad Prism Version 10. All the multiple comparison tests offered by Prism are valid even if the overall ANOVA did not find a significant difference among means. These tests are more focused, so have the power to find differences between groups even when the overall ANOVA is not significant[138]. Our experimental design has been aimed at having 4–5 mice in each experimental group matched for genotype, sex, and age. However, this goal has been difficult to achieve when breeding mice with 6 different genotypes on two different backgrounds, B6 and lupus-prone B6.TC. This difficulty has been further compounded when treating mice with medications that required additional parallel control groups treated with solvents alone. Given that two data points in a group are suitable for statistical analysis[139-142], simultaneous rather than successive testing of large numbers of experimental groups has been preferred for the reliability of metabolic studies. Microscopy images represent the numbers of scored slides from the indicated numbers of mice in each figure legend.

For Fig. 1 the numbers ($n$) of mice in each experimental group were as follows:

Panel A, females: B6 ($n = 5$), B6/Rab4A$^{Q72L}$ ($n = 13$), B6/Rab4A$^{Q72L}$-KO ($n = 6$), B6.TC ($n = 7$), B6.TC/Rab4A$^{Q72L}$ ($n = 22$), B6.TC/Rab4A$^{Q72L}$-KO ($n = 12$); 2-way ANOVA $p = 0.0225$; Sidak's post-hoc test $p$ values corrected for multiple comparisons: B6 vs B6.TC $p = 0.0251$, B6/Rab4A$^{Q72L}$ vs B6.TC/Rab4A$^{Q72L}$ $p < 0.0001$; B6.TC/Rab4A$^{Q72L}$ vs B6.TC/Rab4A$^{Q72L}$-KO $p = 0.0008$.

Panel A, males: B6 ($n = 15$), B6/Rab4A$^{Q72L}$ ($n = 8$), B6/Rab4A$^{Q72L}$-KO ($n = 10$), B6.TC ($n = 7$), B6.TC/Rab4A$^{Q72L}$ ($n = 14$), B6.TC/Rab4A$^{Q72L}$-KO ($n = 9$); 2-way ANOVA $p = 0.0917$; Sidak's post-hoc test $p$ values corrected for multiple comparisons: B6 vs B6/Rab4A$^{Q72L}$-KO $p = 0.0021$, B6 vs B6.TC $p < 0.0001$, B6/Rab4A$^{Q72L}$ vs B6.TC/Rab4A$^{Q72L}$ $p < 0.0001$.

Panel B, females: B6 ($n = 5$), B6/Rab4A$^{Q72L}$ ($n = 13$), B6/Rab4A$^{Q72L}$-KO ($n = 6$), B6.TC ($n = 7$), B6.TC/Rab4A$^{Q72L}$ ($n = 22$), B6.TC/Rab4A$^{Q72L}$-KO ($n = 12$);

2-way ANOVA $p = 0.0794$; Sidak's post-hoc test $p$ values corrected for multiple comparisons: B6 vs B6.TC $p = 0.0141$, B6/Rab4A$^{Q72L}$ vs B6.TC/Rab4A$^{Q72L}$ $p < 0.0001$; B6.TC vs B6.TC/Rab4A$^{Q72L}$-KO, $p = 0.0199$, B6.TC/Rab4A$^{Q72L}$ vs B6.TC/Rab4A$^{Q72L}$-KO $p = 0.0441$.

Panel B, males: B6 ($n = 15$), B6/Rab4A$^{Q72L}$ ($n = 8$), B6/Rab4A$^{Q72L}$-KO ($n = 10$), B6.TC ($n = 7$), B6.TC/Rab4A$^{Q72L}$ ($n = 14$), B6.TC/Rab4A$^{Q72L}$-KO ($n = 9$); 2-way ANOVA $p = 0.0791$; Sidak's post-hoc test

*p* values corrected for multiple comparisons: B6 vs B6/Rab4A$^{Q72L}$-KO *p* = 0.0437, B6 vs B6.TC *p* = 0.0019, B6/Rab4A$^{Q72L}$ vs B6.TC/Rab4A$^{Q72L}$ *p* < 0.0001.

Panel C, females: B6 (*n* = 5), B6/Rab4A$^{Q72L}$ (*n* = 13), B6/Rab4A$^{Q72L}$-KO (*n* = 6), B6.TC (*n* = 7), B6.TC/Rab4A$^{Q72L}$ (*n* = 22), B6.TC/Rab4A$^{Q72L}$-KO (*n* = 12);

2-way ANOVA *p* = 0.0164; Sidak's post-hoc test *p* values corrected for multiple comparisons: B6 vs B6.TC *p* = 0.0001, B6/Rab4A$^{Q72L}$ vs B6.TC/Rab4A$^{Q72L}$ *p* < 0.0001; B6.TC vs B6.TC/Rab4A$^{Q72L}$-KO, *p* = 0.0023, B6.TC/Rab4A$^{Q72L}$ vs B6.TC/Rab4A$^{Q72L}$-KO *p* = 0.0064.

Panel C, males: B6 (*n* = 15), B6/Rab4A$^{Q72L}$ (*n* = 8), B6/Rab4A$^{Q72L}$-KO (*n* = 10), B6.TC (*n* = 7), B6.TC/Rab4A$^{Q72L}$ (n = 14), B6.TC/Rab4A$^{Q72L}$-KO (*n* = 9); 2-way ANOVA p = 0.0873; Sidak's post-hoc test *p* values corrected for multiple comparisons: B6 vs B6/Rab4A$^{Q72L}$-KO *p* = 0.0193, B6 vs B6.TC *p* = 0.0006, B6/Rab4A$^{Q72L}$ vs B6.TC/Rab4A$^{Q72L}$ *p* = 0.0001, B6,TC vs B6.TC/Rab4A$^{Q72L}$ p = 0.0398.

Panel D, females: B6 (*n* = 5), B6/Rab4A$^{Q72L}$ (*n* = 8), B6/Rab4A$^{Q72L}$-KO (*n* = 2), B6.TC (*n* = 10), B6.TC/Rab4A$^{Q72L}$ (*n* = 28), B6.TC/Rab4A$^{Q72L}$-KO (*n* = 9);

Panel D, males: B6 (*n* = 11), B6/Rab4A$^{Q72L}$ (*n* = 12), B6/Rab4A$^{Q72L}$-KO (*n* = 6), B6.TC (*n* = 10), B6.TC/Rab4A$^{Q72L}$ (*n* = 22), B6.TC/Rab4A$^{Q72L}$-KO (*n* = 5);

Panel E, females, GN: B6 (*n* = 3), B6/Rab4A$^{Q72L}$ (*n* = 4), B6/Rab4A$^{Q72L}$-KO (*n* = 4), B6.TC (*n* = 11), B6.TC/Rab4A$^{Q72L}$ (*n* = 8), B6.TC/Rab4A$^{Q72L}$-KO (*n* = 7);

2-way ANOVA *p* = 0.0024; Sidak's post-hoc test p values corrected for multiple comparisons: B6 vs B6.TC p = 0.0230, B6/Rab4A$^{Q72L}$ vs B6.TC/Rab4A$^{Q72L}$ *p* < 0.0001; B6.TC vs B6.TC/Rab4A$^{Q72L}$ *p* = 0.0295, B6.TC vs B6.TC/Rab4A$^{Q72L}$-KO, *p* = 0.0045, B6.TC/Rab4A$^{Q72L}$ vs B6.TC/Rab4A$^{Q72L}$-KO *p* < 0.0001.

Sclerosis/hyalinosis: B6 (*n* = 4), B6/Rab4A$^{Q72L}$ (*n* = 6), B6/Rab4A$^{Q72L}$-KO (*n* = 1), B6.TC (*n* = 8), B6.TC/Rab4A$^{Q72L}$ (*n* = 13), B6.TC/Rab4A$^{Q72L}$-KO (*n* = 9); 2-way ANOVA p = 0.8688.

Glomerulosclerosis: B6 (*n* = 4), B6/Rab4A$^{Q72L}$ (*n* = 6), B6/Rab4A$^{Q72L}$-KO (*n* = 1), B6.TC (*n* = 8), B6.TC/Rab4A$^{Q72L}$ (*n* = 13), B6.TC/Rab4A$^{Q72L}$-KO (*n* = 9); 2-way ANOVA *p* = 0.7980.

Panel F, males, GN: B6 (*n* = 11), B6/Rab4A$^{Q72L}$ (*n* = 6), B6/Rab4A$^{Q72L}$-KO (*n* = 2), B6.TC (*n* = 14), B6.TC/Rab4A$^{Q72L}$ (*n* = 21), B6.TC/Rab4A$^{Q72L}$-KO (*n* = 13);

Sclerosis/hyalinosis: B6 (*n* = 11), B6/Rab4A$^{Q72L}$ (*n* = 6), B6/Rab4A$^{Q72L}$-KO (*n* = 2), B6.TC (*n* = 14), B6.TC/Rab4A$^{Q72L}$ (*n* = 21), B6.TC/Rab4A$^{Q72L}$-KO (*n* = 13);

Glomerulosclerosis: B6 (*n* = 11), B6/Rab4A$^{Q72L}$ (*n* = 6), B6/Rab4A$^{Q72L}$-KO (*n* = 2), B6.TC (*n* = 14), B6.TC/Rab4A$^{Q72L}$ (*n* = 21), B6.TC/Rab4A$^{Q72L}$-KO (*n* = 13);

Figure 3: The numbers (*n*) of mice in each experimental group were as follows: B6.TC Veh (*n* = 5), B6.TC Rapa (*n* = 4), B6.TC/Rab4A$^{Q72L}$ Veh (*n* = 3), B6.TC/Rab4A$^{Q72L}$ NAC (*n* = 5), B6.TC/Rab4A$^{Q72L}$ Rapa (*n* = 6), B6.TC/Rab4A$^{Q72L}$-KO Veh (*n* = 3), B6.TC/Rab4A$^{Q72L}$-KO Rapa (*n* = 2); CD19$^{+}$pS6RP$^{+}$ B cells one-way ANOVA *p* < 0.0001, Sidak's post-hoc test *p* values corrected for multiple comparisons: B6.TC (Veh) vs B6.TC (Rapa) *p* = 0.0029, B6.TC/Rab4A$^{Q72L}$ (Veh) vs B6.TC/Rab4A$^{Q72L}$ (NAC) *p* = 0.0201, B6.TC/Rab4A$^{Q72L}$ (Veh) vs B6.TC/Rab4A$^{Q72L}$ (Rapa) *p* < 0.0001, B6.TC/Rab4A$^{Q72L}$ (Veh) vs B6.TC/Rab4A$^{Q72L}$-KO (Veh) *p* = 0.0276; CD19$^{+}$Akt$^{+}$ B cells one-way ANOVA *p* = 0.0016, Sidak's post-hoc test *p* values corrected for multiple comparisons: B6.TC/Rab4A$^{Q72L}$ (Veh) vs B6.TC/Rab4A$^{Q72L}$ (NAC) *p* = 0.0112, B6.TC/Rab4A$^{Q72L}$ (Veh) vs B6.TC/Rab4A$^{Q72L}$ (Rapa) *p* = 0.0106, B6.TC/Rab4A$^{Q72L}$ (Veh) vs B6.TC/Rab4A$^{Q72L}$-KO (Veh) *p* = 0.0077; CD19$^{+}$pS6RP$^{+}$Akt$^{+}$ B cells one-way ANOVA *p* = 0.0084, Sidak's post-hoc test *p* values corrected for multiple comparisons: B6.TC/Rab4A$^{Q72L}$ (Veh) vs B6.TC/Rab4A$^{Q72L}$ (NAC) *p* = 0.0156, B6.TC/Rab4A$^{Q72L}$ (Veh) vs B6.TC/Rab4A$^{Q72L}$ (Rapa) *p* = 0.0110, B6.TC/Rab4A$^{Q72L}$ (Veh) vs B6.TC/Rab4A$^{Q72L}$-KO (Veh) *p* = 0.0129; CD19$^{+}$CD38$^{+}$ B cells one-way ANOVA *p* = 0.0617, Sidak's post-hoc test *p* values corrected for multiple comparisons: B6.TC (Veh)

vs B6.TC/Rab4A$^{Q72L}$-KO (Veh) *p* = 0.0001; CD19$^{+}$ CD38$^{+}$pS6RP$^{+}$ B cells one-way ANOVA *p* = 0.0007, Sidak's post-hoc test *p* values corrected for multiple comparisons: B6.TC (Veh) vs B6.TC/Rab4A$^{Q72L}$ (Veh) *p* = 0.0195, B6.TC/Rab4A$^{Q72L}$ (Veh) vs B6.TC/Rab4A$^{Q72L}$ (NAC) *p* = 0.0171, B6.TC/Rab4A$^{Q72L}$ (Veh) vs B6.TC/Rab4A$^{Q72L}$ (Rapa) *p* = 0.0004, B6.TC/Rab4A$^{Q72L}$ (Veh) vs B6.TC/Rab4A$^{Q72L}$-KO (Veh) *p* = 0.0154; CD19$^{+}$CD38$^{+}$Akt$^{+}$ B cells one-way ANOVA *p* = 0.0023, Sidak's post-hoc test p values corrected for multiple comparisons: B6.TC (Veh) vs B6.TC/Rab4A$^{Q72L}$ (Veh) *p* = 0.0027, B6.TC/Rab4A$^{Q72L}$ (Veh) vs B6.TC/Rab4A$^{Q72L}$ (NAC) *p* = 0.0018, B6.TC/Rab4A$^{Q72L}$ (Veh) vs B6.TC/Rab4A$^{Q72L}$ (Rapa) *p* = 0.0187, B6.TC/Rab4A$^{Q72L}$ (Veh) vs B6.TC/Rab4A$^{Q72L}$-KO (Veh) *p* = 0.0031; CD19$^{+}$ CD11c$^{+}$ B cells one-way ANOVA *p* = 0.0007, Sidak's post-hoc test *p* values corrected for multiple comparisons: B6.TC/Rab4A$^{Q72L}$ (Veh) vs B6.TC/Rab4A$^{Q72L}$ (NAC) *p* = 0.0009, B6.TC/Rab4A$^{Q72L}$ (Veh) vs B6.TC/Rab4A$^{Q72L}$ (Rapa) *p* = 0.0003, B6.TC/Rab4A$^{Q72L}$ (Veh) vs B6.TC/Rab4A$^{Q72L}$-KO (Veh) *p* = 0.0025. **G** Dimensionality reduction analyses by tSNE and UMAP of the expansion of mTORC1$^{+}$mTORC2$^{+}$ CD4$^{+}$ T cells and mTORC1$^{+}$mTORC2$^{+}$CD19$^{+}$CD38$^{+}$ plasma cells in B6.TC/Rab4A$^{Q72L}$ mice, which were reversed by NAC and rapamycin treatment or the inactivation of Rab4A B6.TC/Rab4A$^{Q72L}$-KO mice. Depletion of CD8$^{+}$ T cells in B6.TC/Rab4A$^{Q72L}$ mice over B.TC and B6.TC/Rab4A$^{Q72L}$-KO mice is clearly visualized. Upper panels and dimensionality reduction analyses of metabolic and cell surface markers were performed by using t-SNE and UMAP plug-in programs integrated with the FlowJo version 10 software. Color axis: blue (0) → red (max). Lower panels, integrated t-SNE and UMAP analyses both unveiled potential expansions of mTORC1$^{+}$mTORC2$^{+}$ CD4$^{+}$ T cells in B6.TC mice and mTORC1$^{+}$mTORC2$^{+}$CD19$^{+}$CD38$^{+}$ B cells in B6.TC/Rab4A$^{Q72L}$ mice. Depletion of CD8 T cells in B6.TC/Rab4A$^{Q72L}$ mice over B6.TC and B6.TC/Rab4A$^{Q72L}$-KO mice is clearly visualized. These changes were reversed by NAC and rapamycin treatment (marked with red arrows) or the inactivation of Rab4A in B6.TC/Rab4A$^{Q72L}$-KO mice (marked with green arrows). Boolean functions were used to sum up the animals of each experimental group into single files arrayed for t-SNE and UMAP analyses in top and bottom rows, respectively. Color scale from blue to red: −1622 to 262,856. **H** Flow cytometry analysis of the impact by Rab4A activation and treatment by rapamycin and NAC on mitochondrial metabolism in freshly isolated splenocyte subsets from lupus-prone mice. Dimensionality reduction analyses by t-SNE and UMAP highlighted unveiled the expansion of metabolically active TMRM$^{+}$ DN T cells and MTG$^{+}$CD19$^{+}$CD11c$^{+}$ ABC B cells in B6.TC/Rab4A$^{Q72L}$ mice, which were reversed by NAC and rapamycin treatment or the inactivation of Rab4A B6.TC/Rab4A$^{Q72L}$-KO mice. Dimensionality reduction analyses of metabolic and cell surface markers were performed by t-SNE and UMAP plug-in programs using FlowJo version 10 software. Upper panels, **c**olor axis: blue (0) → red (max). Lower panels, Integrated analyses by t-SNE and UMAP both unveiled the expansion of metabolically active MTG$^{+}$ and MTG$^{+}$TMRM$^{+}$ DN (CD3$^{+}$CD4$^{-}$CD8$^{-}$) T cells and MTG$^{+}$CD19$^{+}$CD11c$^{+}$ ABC B cells in B6.TC/Rab4A$^{Q72L}$ mice, which were reversed by NAC and rapamycin treatment (marked with red arrows) or the inactivation of Rab4A in B6.TC/Rab4A$^{Q72L}$-KO mice (marked with green arrows). Boolean functions were created to sum up the animals of each experimental group into single files arrayed with t-SNE and UMAP in top and bottom rows, respectively. Color scale from blue to red: −1622 to 262,856.

## Reporting summary

Further information on research design is available in the Nature Portfolio Reporting Summary linked to this article.

## Data availability

RNA-sequencing data have been uploaded to the NCBI Gene Expression Omnibus (Accession GSE245413). All other data are available in the article and its Supplementary files or from the corresponding author upon request. Source data are provided with this paper.

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

## Acknowledgements

This work was supported in part by grants AI072648 and AI122176 from the National Institutes of Health, the Phillips Lupus and Autoimmunity Center of Excellence, the Research Foundation of the American College of Rheumatology, and the Central New York Community Foundation.

## Author contributions

A.P. (Andras Perl) designed the study, oversaw the design, execution, and analyses of each experiment. N.H. and T.W. designed experiments and analyzed data. B.W., Z.O., T.F., G.C., Z.-W.L., J.L., M.B., M.D., D.K., A.P. (Akshay Patel), J.P., T.C., M.S., L.M., M.H., F.M., and K.B. performed experiments and analyzed data. N.H., T.W., and A.P. (Andras Perl) drafted the manuscript, and all authors revised the manuscript.

## Competing interests

The authors declare no competing interests.

## Additional information

Nick Huang [1,2,8], Thomas Winans[1,2,8], Brandon Wyman [1,2], Zachary Oaks[1,2], Tamas Faludi [1], Gourav Choudhary[1,2], Zhi-Wei Lai[1], Joshua Lewis[1], Miguel Beckford[1], Manuel Duarte[1], Daniel Krakko [1], Akshay Patel [1,2], Joy Park [1,2], Tiffany Caza[1], Mahsa Sadeghzadeh[1,2], Laurence Morel[3], Mark Haas[4], Frank Middleton [5], Katalin Banki[6] & Andras Perl [1,2,7] ✉

[1]Department of Medicine, State University of New York, Upstate Medical University, Norton College of Medicine, Syracuse, New York, NY 13210, USA. [2]Department of Biochemistry and Molecular Biology, State University of New York, Upstate Medical University, Norton College of Medicine, Syracuse, New York, NY 13210, USA. [3]Department of Pathology, Immunology, and Laboratory Medicine, University of Florida, Gainesville, FL 32610, USA. [4]Department of Pathology and Laboratory Medicine, Cedars-Sinai Medical Center, Los Angeles, CA 90048, USA. [5]Department of Neuroscience and Physiology, State University of New York, Upstate Medical University, Norton College of Medicine, Syracuse, New York, NY 13210, USA. [6]Department of Pathology, State University of New York, Upstate Medical University, Norton College of Medicine, Syracuse, New York, NY 13210, USA. [7]Department of Microbiology and Immunology, State University of New York, Upstate Medical University, Norton College of Medicine, Syracuse, New York, NY 13210, USA. [8]These authors contributed equally: Nick Huang, Thomas Winans. ✉e-mail: perla@upstate.edu

