## [Peer Review File · Nature Communications]

RAB4A-DIRECTED ENDOSOME TRAFFIC SHAPES PRO-INFLAMMATORY MITOCHONDRIAL METABOLISM IN T CELLS VIA MITOPHAGY, CD98 EXPRESSION, AND KYNURENINE-SENSITIVE MTOR ACTIVATIONREVIEWER COMMENTS

Reviewer #1 T cells, inflammation (Remarks to the Author):

Multiple and profound T cell abnormalities in SLE are intertwined with disease expression. In the manuscript, authors linked Rab4A mediated metabolic alterations in T cells with lupus pathogenesis. Major findings include 1) Activation of Rab4A in T cells promotes autoimmunity in mouse models of either lupus or multiple sclerosis. 2) The distinct and specific regulations of activated Rab4A on cell metabolisms in different types of T cells. 3) Identification of dysregulated KYN metabolism and altered CD98 expression in T cells from B6.TC/Rab4AQ72L mice with their contributions to lupus pathogenesis. The paper is conceptually novel and interesting and extensive data have been presented.

Specific comments:

1. Transient thymic involution is frequently found during inflammation. Considering the fact that overexpression of Rab4A has been detected in thymocytes, it would be interesting to know whether the activation of Rab4A interferes T cell development in thymus which contributes to altered ratio of CD4/CD8 in the peripheral.
2. Considering the remarkable splenomegaly in B6.TC/Rab4AQ72L mice, the percentage reduction of CD8 T cells is not equivalent to the decrease of cellularity. To prove that Rab4A activation depleted CD8+ T cells, it would be important for authors to provide cell counting as well.
3. Elevated serological KYN in B6.TC/Rab4AQ72L mice were diminished in mice with T cell targeted deletion of Rab4A, which indicated T cells especially CD8 T cells are main source of serological KYN in B6.TC/Rab4AQ72L mice. It would be interesting to know whether live or dead CD8 T cells contribute most to the serum KYN pool.
4. Self-reactive CD8+ T cells are one of the main sources of expanded DNT cells in SLE and the CD8-DN T cell transition is possibly regulated by mTOR, it should be discussed if the CD8-DN T cell transition partially explain the reduction of CD8 T cells and the expansion of DNT cells in B6.TC/Rab4AQ72L mice.
5. Mitochondrial dysfunction has been linked with T cells exhaustion and it would be interesting to know whether CD8 T cells in B6.TC/Rab4AQ72L display exhausted phenotype. This might be another key pathogenesis mechanism since reduced cytolytic function of CD8 T cells promotes autoimmunity.

6. Different from the amino acid transport function of the light chain, the CD98 heavy chain amplifies integrin signaling and is more essential for cell proliferation and cell survival. It would be helpful to evaluate the expression of CD98 heavy chain as well.
7. The indexes of color axis of all t-SNE and UMAP plots are missing. The induction or enhanced expression of pAKT is not obvious since generally blue and green are considered as negative or low expression. A signal amplification with secondary antibodies might help.
8. The discussion is a little wordy.

Reviewer #2 mTOR, T cells, metabolism (Remarks to the Author):

In this manuscript, Huang and colleagues generated novel Rab4a constitutive active knockin and Rab4a T cell conditional knockout mouse models. These mice were bred with SLE1.2.3 lupus mouse model to study the potential contribution of Rab4a in lupus disease. They propose that Rab4a promotes mTOR activation and mitochondrial metabolism partly by upregulation of CD98 and accumulation of kynurenine. This study follows up on their previous observation that lupus patient T cells have increased Rab4a expression and mTORC1 activation, which attempts to provide in-depth mechanistic insight on how Rab4a may contribute to lupus pathophysiology. The authors generated a large body of data that are partly innovative and partly confusing. I don't think I'm fully convinced that the data support their many conclusions, particularly in regard to the mechanisms. I have following comments.

1. Throughout the figures, there are numerous multi-parameter comparisons. Many p value bars are empty (Fig. 1A-1D, entire Fig. 2, Fig. 3A-3F, Fig. 5B, D, Fig. 7D, Fig. 8E, Fig. S2, Fig. S3, Fig. S6A, Fig. S7D). It is unclear why the authors picked any two of the parameters for comparison (or not comparing others), what the p values are (they are empty). Without all the numbers, it is not possible to understand the statistical significances. They gave an impression of random cherry picking for statistical comparisons. In addition, many lack details about the statistical methods, such as Fig. 2, Fig. 3. Some statistic methods may not be proper, such as Fig. 1, Fig. 3. Furthermore, some parameters had only two data points, such as Fig. 1A-1C, 2A-2D and some panels in Fig. 5G, Fig. S2, Fig. S6). I don't think two data points are sufficient for any statistical analyses. Finally, I'm not sure what are the blue p

values bars in Fig. 7E. What do they compare? Is it appropriate to do the statistical analysis this way?

2. Fig. S1 validates the genetic mouse model. The immunoblot in Fig. S1E should have WT CD4T cells side by side with the Rab4A KI and Rab4A KO T cells to confirm the deletion. It is not fair to compare T cells to liver cells.

3. Some of the disease presentations shown in Fig. 1 and Fig. S2 may need some explanation. Why was the elevated proteinuria phenotype in TC/Rab4a KI restricted to 20-29 weeks, but not later time points (Fig. S2B), while the GN phenotype was present at 50 weeks (Fig. 1E)? The proteinuria phenotype in TC vs B6 was also quite inconsistent and weak. The T cell phenotypes appear to be subtle in Fig. 2 and not convincing (see above points on statistics). Have the investigators performed any histology on spleen tissues from these mice?

4. The central hypothesis is that Rab4a overactivation promote mTOR signaling and mitochondrial metabolism, which contributes to lupus development. There are several issues on the data supporting this hypothesis. First, TC mice themselves showed increased mTORC1 activation in T cells based on their previous publications. But I don't see expansion of CD4 T cells (Fig. 2B), or increased MTG or TMRM staining (Fig. 2C), or even pS6RP staining (Fig. S4) comparing TC to B6 control. Second, there is no clear increase of pS6RP or p-AKT in CD4 T cells (Fig. 3E) and possibly a stronger increase of pS6RP and p-AKT in CD8 T cells (Fig. 3E) comparing TC/Rab4A KI to TC alone, although Fig. S4 did show increase of pS6RP in CD4 T cells from TC/Rab4A KI mice (it is unclear how they quantified pS6RP in the flow data). In fact, Fig. 3G seems to be show that TC/Rab4 KI T cells have reduced pS6RP and pAKT. However, Fig. 2 showed that CD4 T cells expanded at the expense of CD8 T cells in TC/Rab4A KI mice. Shouldn't we expect the opposite? Third, the Seahorse data in Fig. 5 and Fig.S6 are a little confusing too. If the increased OCR in TC/Rab4KI in Fig. 5 A is a consequence of increased mTORC1, shouldn't we expect to see rapamycin rectifying it? But Fig. 5B shows this is not the case. In fact, rapamycin seems to increase OCR in CD8 T cells and ECAR in CD4 T cells, which seems to be a little contradictory. Fig. 5E lacks legends. Lastly, the isotope tracing experiment is very nice and comprehensive. But again, we would expect that rapamycin can reverse the increased glutamate, a-ketoglutarate, malate, fumarate flux in TC/Rab4A KI CD4 T cells, but it did not. Also, why rapamycin can increase a-ketoglutarate and fumarate flux over TC/Rab4A KO CD4 T cells is unclear. The glucose tracing results (Fig.

S6C) seem to contradict the investigator's early results that showed increased glucose metabolism in TC T cells over B6 T cells. Can the investigators provide an explanation? In fact, one would expect to see increased glycolysis in TC/Rab4A KO over TC because Tsc1 KO T cells have increased ECAR.

5. Following on the previous point, the mTORC2 activity is measured by flow cytometry of p-AKT. The pS6RP and p-AKT staining data in Fig. 6C, 6D and Fig. S8 was very weak and not convincing at all. Immunoblot of p-AKT is needed to confirm the conclusion. This also brings question on the conclusion in Fig. 3E-3G. Not sure if one can expect to see 20% of resting B6 CD4 T cells are p-AKT+. How did the authors gate? Some of the key conclusions on mTORC2 should be confirmed by immunoblot. The conclusion that kynurenine promotes mTOR activity in T cells is not convincing by the presented data.

6. Fig. 3H does not seem to be consistent with Fig. 2. I don't see increased MTG+TMRM+ in TC/Rab4a KI CD4 T cells vs TC. But there seemed to be different in Fig. 2D. If CD4-CD8- double negative T cells have the highest MTG/TMRM increase in TC/Rab4a KI mice, shouldn't we expect to see increased DN T cells in TC/Rab4a KI mice? But in fact, we see the opposite phenotypes (Fig. 2B).

7. Some panels of Fig. 4 may be moved to supplementary figures. They do not provide a lot of information.

8. The authors should present Cd98 staining data for Fig. 6B. The CD98 flow data in Fig. S8 was also quite weak.

9. I think some of the data Fig. S7 should be moved to main figure because it is a key link to the authors' hypothesis. The serum kynurenine level is not altered in TC mice compared to B6 mice, which contradicts their previous publication. Can the authors provide an explanation? When we look at kynurenine level in T cells, it remains largely the unaltered in TC/Rab4a KI CD4 T cells, but increased in TC/Rab4a KI CD8 T cells, which was reversed by rapamycin. But if mTORC1 and mitochondrial activity are more highly elevated in TC/Rab4a KI CD4 T cells, as shown in Fig. 2, Fig. 3 and Fig. 5, shouldn't we expect kynurenine level is higher in TC/Rab4a KI CD4 T cells?

10. It is challenging to read Fig. 7. Most of the panels are so small, with a lot of texts in Fig. 7A are too blurred to read. Fig. 7C showed that the alteration of CD98 and CD71 is mostly restricted to DN T cells, not CD4 and CD8 T cells, which does not fit their overall hypothesis very well. The conclusion that CD71 and CD98 are elevated in TC/Rab4a KI T cells is not well

supported. Then, the CD98+CD71+ cells form a distinct population in TC/Rab4a KI DN T cells. Are these cells a distinct cell subset or an indication of cell activation (mTOR activation presumably)? For the recycling experiment in Fig. 7G and H, if Rab4a is constitutively active in TC/Rab4a KI T cells, shouldn't we see stronger internalization of CD71 and Cd98, and impaired internalization in Rab4a KO T cells (but not clear in Fig. 7G and 7H)? It is unclear how reliable the assay is.

11. Fig. 8 tried to further establish a causal link between CD98 and mTOR and lupus patients. CD98 is known to be required for optimal mTOR activity. Fig. 8A and 8B showed that CD98 expression is restricted to DN T cell, but pS6RP increase is observed in CD4+CD98+ and CD8+CD98+ T cells from lupus patients. These data are not completely consistent either. The authors' previous publication showed lupus T cells have modest increase of pS6RP. How about mTOR activity in CD98- T cells? Fig. 8C aims to show reduced CD98 on human T cells can reduce pS6. But the effects are very small. CD98 increase in Fig. 8E is not very convincing either. Overall, it is not clear such a small effect to promote CD98 expression by Rab4a can explain the disease phenotypes in the TC/Rab4a KI mice.

12. The title indicates that Rab4A controls mitophagy in lupus T cells. The supporting data are Fig. S3A and S3B. Such a weak change of Drp1 expression is not sufficient to support mitophagy is regulated by Rab4a.

13. There are several places where the texts are mistaken or wrong figure is referred to (e.g., While both CD71 and CD98 were most abundantly expressed on DN T cells, the majority of CD98+ T cells did not co-express CD71 in B6.TC mice (Figure 6C)). It should be Figure 7C.

Reviewer #3 SLE (Remarks to the Author):

In this paper, Huang et al. revealed that the constitutive activation of Rab4A exerts dominant control over pro-inflammatory signal transduction networks and mechanistically in vivo and in vitro. There are many concerns in this study.

There are significant issues in statistics, especially multiple comparisons. The authors should show the statistical analysis in each figure legend. The authors didn't show the p-value in many Figures.

Since previous reports revealed the importance of Rab4A in patients with SLE and lupus models, the novelty of this paper is not so significant.

1. Why the authors used Rab4A Q72L mice when they made the mice using loxP systems? Is something wrong if they used B6 mice or B6 TC mice for using the loxP system? The authors should explain why they chose Q72L alleles and how the differences between B6 and Rab4A Q72L occurred.

2. In Figure S1E, how about the expression of Rab4A in CD8 T cells?

3. In Figure 1A-D and some other figures, the significant difference or p-value should be shown.

ANA in the male are increased compared to female? Fold change in male mice should be calculated using a female mouse to compare sex.

Is the number of some groups (e.g., B6 or B6.TC) only two? Why didn't the authors use ANOVA or multiple comparisons in Figures 1A-F, 2S, and other Figures ?

4. In Figure S2, there is no data in some groups.

The authors described that Non-autoimmune B6 males also had greater proteinuria ($1.01 \pm 0.14 \mu\text{g}/\mu\text{l}$) than female controls ($0.49 \pm 0.07 \mu\text{g}/\mu\text{l}$, $p=0.0299$). Is this in the normal range or pathologic ?

5. Why the inactivation of Rab4A in T cells abrogated proteinuria in male B6.TC/Rab4AQ72L-KO mice did not have significant differences in comparison to B6.TC/Rab4AQ72L female controls?

6. About the sentence "Interestingly, male B6.TC/Rab4AQ72L-KO mice developed severe glomerulosclerosis with greater percentage of glomeruli with sclerosis or hyalinosis relative to B6.TC mice with normal Rab4A alleles", the authors should discuss and explain the differences between genders. Do the authors think the Rab4A target therapy is ineffective or harmful for male patients with lupus nephritis?

7. The authors should describe the statistical method in all the Figure legends. If the authors would like to have significant differences, the number of each group should be more than three.

8. In Figure 2, the DN T cell population increases in B6.TC/Rab4AQ72L-KO mice. Basically, the population of DN T cells reflects the disease activity of the lupus model. The authors should explain and discuss the discrepancy.

9. In Figure S3, the differences of Drp1 is not so significant even though the Rab4A is knocked down. Is this the main mechanism?

10. In the manuscript of Figure 3S, the reasons why the authors focused on Drp1, TSC1, and mLST8, should be clearly explained.

11. A dot plot should be shown in some Figures (e.g., Figure 3S CD).

12. "Rapamycin expanded DN T cells in B6.TC mice while it depleted DN T cells in B6.TC/Rab4AQ72L mice; NAC also depleted DN T cells in B6.TC/Rab4AQ72L mice." The authors should explain the reasons.

13. In Figure 5, the space capacity of the mito stress test should be shown. In Figure 5E and Figure S6, it is unclear which comparison has significant differences.

14. In Figure 5G and Figure 6SC and D, if the authors revealed the results in the pathway map, it is easily understood. Again the number of some groups is only two. Why does rapamycin's effect on metabolic flux differ between CD4 and CD8 T cells?

15. In Figure S7, pyruvate is significantly reduced in B6TC/Rab4a KO compared to B6TC/Rab4a. On the other hand, the glycolysis stress test does not have significant differences. Why this discrepancy occurs? How about lactate in this setting?

REVIEWER 1 COMMENTS

Reviewer #1 T cells, inflammation (Remarks to the Author)

General Comments: Multiple and profound T cell abnormalities in SLE are intertwined with disease expression. In the manuscript, authors linked Rab4A mediated metabolic alterations in T cells with lupus pathogenesis. Major findings include 1) Activation of Rab4A in T cells promotes autoimmunity in mouse models of either lupus or multiple sclerosis. 2) The distinct and specific regulations of activated Rab4A on cell metabolisms in different types of T cells. 3) Identification of dysregulated KYN metabolism and altered CD98 expression in T cells from B6.TC/Rab4AQ72L mice with their contributions to lupus pathogenesis. The paper is conceptually novel and interesting and extensive data have been presented.

Response: We very much appreciate the positive overall comments on the conceptual novelty of the paper and the robustness of supporting data.

Specific comments:

Comment 1. Transient thymic involution is frequently found during inflammation. Considering the fact that overexpression of Rab4A has been detected in thymocytes, it would be interesting to know whether the activation of Rab4A interferes T cell development in thymus which contributes to altered ratio of CD4/CD8 in the peripheral.

Response: We agree with the discerning comment that the influence of Rab4A on thymic development is expected to be complex and likely to be affected by the systemic inflammation in the setting of lupus. Indeed, Rab4A influenced T cell development in the thymus. We performed additional experiments to characterize major stages of intra-thymic T cell development in 23 age-matched female mice carrying WT or constitutively active Rab4^{Q72L} alleles, or lacking Rab4A in T cells in the B6 control and B6.TC lupus-prone backgrounds (**Figures S7-S10**). These mice were of 25 ± 1.7 weeks of age when B6.TC/Rab4A^{Q72L} females already showed elevated autoantibody production and proteinuria (**Figure 1**). This analysis indicates that CD4⁻CD8⁻ double-negative (DN) T cells, which precede the later stages of CD4⁺CD8⁺ double-positive and CD4⁺ or CD8⁺ single-positive T-cell development in the thymus, are depleted in B6/Rab4A^{Q72L} females but expanded in B6.TC/Rab4A^{Q72L} females (**Figure S7**). Intracellular staining demonstrated coordinate changes in mTORC1 and mTORC2, which underlay the skewed abundance of DN thymocytes (**Figure S8**). These new results suggest that Rab4A and lupus synergistically activate mTORC1 and mTORC2 and expand DN T cells in the thymus of B6.TC/Rab4A^{Q72L} mice (**Figures S7 and S8**). This information has been added to the Results section of the revised manuscript. These findings also support the notion that DN T cells are heterogeneous as further addressed in responses to comments 8 and 12 of Reviewer 3.

Comment 2. Considering the remarkable splenomegaly in B6.TC/Rab4A^{Q72L} mice, the percentage reduction of CD8 T cells is not equivalent to the decrease of cellularity. To prove that Rab4A activation depleted CD8⁺ T cells, it would be important for authors to provide cell counting as well.

Response: Following the reviewer's suggestion, we newly included absolute CD4⁺ and CD8⁺ T-cell counts in **Figure S3**. The inactivation of Rab4A in T cells reduced the relative (**Figure 2B**) and absolute numbers of CD4⁺ T cells in B6/Rab4A^{Q72L}-KO and B6.TC/Rab4A^{Q72L}-KO mice in comparison to B6/Rab4A^{Q72L} and B6.TC/Rab4A^{Q72L} controls, respectively (**Figure S3**). In contrast, relative rather than absolute numbers of CD8⁺ T cells were expanded by the inactivation of Rab4A in T cells of B6/Rab4A^{Q72L}-KO and B6.TC/Rab4A^{Q72L}-KO mice over B6/Rab4A^{Q72L} and B6.TC/Rab4A^{Q72L} controls, respectively (**Figures 2B and S3**). Thus, the following statement has been included in the revised Results section: ...these findings indicate that skewing of CD4:CD8 T-cell abundance by Rab4A inactivation may be driven by the absolute depletion of CD4⁺ T cells rather than the expansion of CD8⁺ T cells in B6/Rab4A^{Q72L}-KO and B6.TC/Rab4A^{Q72L}-KO mice.

Comment 3. Elevated serological KYN in B6.TC/Rab4A^{Q72L} mice were diminished in mice with T cell targeted deletion of Rab4A, which indicated T cells especially CD8 T cells are main source of serological KYN in B6.TC/Rab4A^{Q72L} mice. It would be interesting to know whether live or dead CD8 T cells contribute most to the serum KYN pool.

Response: Following CD3/CD28 co-stimulation for 72 hours in vitro, CD4⁺ and CD8⁺ T cells used for metabolomic studies had similar, nearly 100% viability. However, these findings do not exclude the possibility that dying CD8⁺ T cells may serve as source of KYN accumulation in sera of B6.TC/Rab4A^{Q72L} mice.

Comment 4. Self-reactive CD8⁺ T cells are one of the main sources of expanded DNT cells in SLE and the CD8-DN T cell transition is possibly regulated by mTOR, it should be discussed if

the CD8-DN T cell transition partially explain the reduction of CD8 T cells and the expansion of DNT cells in B6.TC/Rab4A^{Q72L} mice.

Response: Indeed, it has been documented that DN T cells may derive from activated CD8⁺ T cells in patients¹ and mice with SLE^{2,3}. Given that the expansion of DN T cells is mTOR-dependent⁴⁻⁷, Rab4A-mediated mTOR activation may contribute to the depletion of CD8⁺ T cells and reciprocal expansion of DN T cells in B6.TC/Rab4A^{Q72L} mice. This mechanism is supported by the effective reversal of DN T-cell expansion in rapamycin-treated B6.TC/Rab4A^{Q72L} mice. The above considerations have been included in the Discussion of the revised manuscript. Please, note that DN T cells are heterogeneous, which is further discussed below in response to comments 8 and 12 from Reviewer 3.

Comment 5. Mitochondrial dysfunction has been linked with T cells exhaustion and it would be interesting to know whether CD8 T cells in B6.TC/Rab4A^{Q72L} display exhausted phenotype. This might be another key pathogenesis mechanism since reduced cytolytic function of CD8 T cells promotes autoimmunity.

Response: In order to address the reviewer's question, we performed additional studies to evaluate markers of exhaustion, T-bet, Eomes, TIGIT, PD-1, PL-L2, and CTLA-4⁸ along with mTORC1 (pS6RP), mTORC2 (pAkt), and CD98 in CD4⁺, CD8⁺, and DN T cells from the spleen and DN, DP, CD4⁺, and CD8⁺ SP T cells from the thymus of B6.TC, B6.TC/Rab4A^{Q72L}, and B6.TC/Rab4A^{Q72L}-KO mice, using five age-matched females per genotype. In the spleen, we noted a moderate expansion of pAkt⁺T-bet⁺ DN T cells but failed to identify consistent changes in markers of exhaustion in CD4⁺ or CD8⁺ T cells (**Figure S9A**). Following CD3/CD28 co-stimulation, we observed marked expansions of PD-1⁺T-bet⁺, CTLA4⁺T-bet⁺, Eomes⁺T-bet⁺, TIGIT⁺T-bet⁺, TIGIT⁺CTLA4⁺, and TIGIT⁺Eomes⁺ DN T cells in the thymus of B6.TC/Rab4A^{Q72L} mice over those in B6.TC/Rab4A^{Q72L}-KO mice (**Figure S9B**). These DN thymocytes displayed mTORC1 (**Figure S10A**) and mTORC2 activation (**Figure S10B**) and increased expression of CD98 (**Figure S10C**), indicating that the Rab4A/CD98/mTOR axis may cause exhaustion in the early DN stage of T-cell development. This information has been added to the Results section of the revised manuscript.

Comment 6. Different from the amino acid transport function of the light chain, the CD98 heavy chain amplifies integrin signaling and is more essential for cell proliferation and cell survival. It would be helpful to evaluate the expression of CD98 heavy chain as well.

Response: CD98 (SLC3A2) forms a heterodimer with LAT1 (SLC7A5). The CD98 (SLC3A2, ~630 amino acids) protein exists as the heavy chain of a heterodimer, covalently bound through di-sulfide bonds to a light chain, such as LAT1 (SLC7A5, ~507 amino acids in humans and 512 amino acids in mice)⁹. Therefore, we have assumed the reviewer meant that we should evaluate the expression of LAT1, the other component of the heterodimer. LAT1 is a transmembrane protein^{8,9}, and thus its expression can be reliably detected by western blot analysis. Since mouse (45 kD) and human LAT1 (55 kD) migrate at different molecular weights, we used α -tubulin and β -actin as loading controls. As shown in **Figure S15**, LAT1 protein levels were not affected by Rab4A in CD4⁺ or CD8⁺ T cells of age-matched female B6.TC, B6.TC/Rab4A^{Q72L}, and B6.TC/Rab4A^{Q72L}-KO mice (**Figure S15A**) or Jurkat human CD4⁺ T cell lines with altered expression of Rab4A (**Figure S15B**). This information has been included in the Results section of the revised manuscript. Both CD98 and LAT1 are ubiquitously expressed. As we have shown in the original manuscript (**Figure S10**), knockdown of CD98 failed to influence LAT1 levels in HeLa cells (**Figure S14** in revised manuscript).

Comment 7. The indexes of color axis of all t-SNE and UMAP plots are missing. The induction or enhanced expression of pAKT is not obvious since generally blue and green are considered as negative or low expression. A signal amplification with secondary antibodies might help.

Response: As suggested by the reviewer, all t-SNE and UMAP figure panels have been updated with color axis scales. The intensity of fluorescence varied amongst different reagents. We only used primary antibodies directly labelled with fluorochromes to avoid cross reactivity amongst different antigens.

Comment 8. The discussion is a little wordy.

Response: We have appreciated the reviewer's suggestion and made every effort to shorten the Discussion. However, we have been also asked to carry out and interpret additional experiments and provide further mechanistic discussions, which have been kept as succinct as possible.

REVIEWER 2 COMMENTS

Reviewer #2 mTOR, T cells, metabolism (Remarks to the Author)

General Comments: In this manuscript, Huang and colleagues generated novel Rab4a constitutive active knockin and Rab4a T cell conditional knockout mouse models. These mice were bred with SLE1.2.3 lupus mouse model to study the potential contribution of Rab4a in lupus disease. They propose that Rab4a promotes mTOR activation and mitochondrial metabolism partly by upregulation of CD98 and accumulation of kynurenine. This study follows up on their previous observation that lupus patient T cells have increased Rab4a expression and mTORC1 activation, which attempts to provide in-depth mechanistic insight on how Rab4a may contribute to lupus pathophysiology. The authors generated a large body of data that are partly innovative and partly confusing. I don't think I'm fully convinced that the data support their many conclusions, particularly in regard to the mechanisms. I have following comments.

Response: We have appreciated the general comments that a large body of innovative data has been generated and the need for clarity and conservative interpretation of the mechanisms by which Rab4A controls lupus pathogenesis.

Comment 1. Throughout the figures, there are numerous multi-parameter comparisons. Many p value bars are empty (Fig. 1A-1D, entire Fig. 2, Fig. 3A-3F, Fig. 5B, D, Fig. 7D, Fig. 8E, Fig. S2, Fig. S3, Fig. S6A, Fig. S7D). It is unclear why the authors picked any two of the parameters for comparison (or not comparing others), what the p values are (they are empty). Without all the numbers, it is not possible to understand the statistical significances. They gave an impression of random cherry picking for statistical comparisons. In addition, many lack details about the statistical methods, such as Fig. 2, Fig. 3. Some statistic methods may not be proper, such as Fig. 1, Fig. 3. Furthermore, some parameters had only two data points, such as Fig. 1A-1C, 2A-2D and some panels in Fig. 5G, Fig. S2, Fig. S6). I don't think two data points are sufficient for any statistical analyses. Finally, I'm not sure what are the blue p values bars in Fig. 7E. What do they compare? Is it appropriate to do the statistical analysis this way?

Response: The "empty" connecting bars reflected two-tailed p values < 0.05 without displaying exact numbers, as indicated in figure legends of the original manuscript. This was

intended to avoid overcrowding. By contrast, the absence of connecting bars between experimental groups reflected p values > 0.05.

In the revised manuscript, we have consistently included exact p values with 4 digits after the decimal point above all connecting bars within each figure. Broken connecting lines are displayed for two-tailed p values between 0.05 and 0.1; which reflect one-tailed significance. The absence of connecting bars reflects two-tailed p values > 0.05. This information has been added to the statistical section of the Methods in the revised manuscript.

In agreement with the reviewer, we very much prefer having 4-5 mice in each experimental group matched for genotype, sex, and age. However, this is exceedingly difficult when breeding mice with 6 different genotypes on two different backgrounds, B6 and lupus-prone B6.TC. This difficulty is further compounded by treating mice with medications that require control groups exposed in parallel to solvents alone. It has been taking several years of breeding, genotyping, and constant maintenance of a large colony of 700-900 mice to prepare matched sets for the studies. The results of this paper have taken eight years to accumulate. When available, additional data points normalized to the mean of B6 WT controls have been added to the figure panels. Thus, we have included additional data points in Figures 1A-C.

Nevertheless, two data points in a group are minimally suitable for statistical analysis according to peer-reviewed literature¹⁰⁻¹³ and the Graphpad Software used in our studies (<https://www.graphpad.com/support/faqid/591/>: “The equations that calculate the SD, SEM and CI all work just fine when you have only duplicate (N=2) data.”... ” Is it valid to compute a t test or ANOVA with only two replicates in each group? Sure. You get more power with more data. But n=2 is enough for the results to be valid. (But of course t tests and ANOVA cannot be done with n=1.”) Two data points can also be used in the statistical package of Microsoft excel. This information has been included in the Methods Section of the revised manuscript, as follows: “Our experimental design has been aimed at having 4-5 mice in each experimental group matched for genotype, sex, and age. However, this goal has been difficult to achieve when breeding mice with 6 different genotypes on two different backgrounds, B6 and lupus-prone B6.TC. This difficulty has been further compounded when treating mice with medications that required additional parallel control groups treated with solvents alone. Given that two data points in a group are suitable for statistical analysis¹⁰⁻¹³, simultaneous rather than successive testing of large numbers of experimental groups has been preferred for reliability of metabolic studies.”

P values in blue reflect comparison between CD4⁺ and DN T cells or CD8⁺ DN T cells in B6.TC control mice in Figure 7E, which corresponds to Figure 8C in the revised manuscript.

Comment 2. Fig. S1 validates the genetic mouse model. The immunoblot in Fig. S1E should have WT CD4T cells side by side with the Rab4A KI and Rab4A KO T cells to confirm the deletion. It is not fair to compare T cells to liver cells.

Response: In accordance with the reviewer’s suggestion, we have included western blots of both CD4⁺ and CD8⁺ T cells from B6, B6/Rab4A^{Q721}, and B6/Rab4A^{Q721}-KO mice in Figure S1E. The liver extracts have been removed.

Comment 3. Some of the disease presentations shown in Fig. 1 and Fig. S2 may need some explanation. Why was the elevated proteinuria phenotype in TC/Rab4a KI restricted to 20-29 weeks, but not later time points (Fig. S2B), while the GN phenotype was present at 50 weeks (Fig. 1E)? The proteinuria phenotype in TC vs B6 was also quite inconsistent and weak. The T cell phenotypes appear to be subtle in Fig. 2 and not convincing (see above points on statistics). Have the investigators performed any histology on spleen tissues from these mice?

Response: The question “why was the elevated proteinuria phenotype in TC/Rab4a KI restricted to 20-29 weeks, but not later time points (Fig. S2B)” may have been an oversight, as proteinuria was also elevated in B6.TC/Rab4A^{Q72L} mice relative to B6.TC/Rab4A^{Q72L}-KO mice at ages of 40-49 weeks and > 50 weeks, as shown in **Figure S2B**. This indicated a consistent impact of activated Rab4A in T cells at the onset of nephritis. It is correct that the severity of proteinuria between B6.TC and B6.TC/Rab4A^{Q72L} mice has become equilibrated after 30 weeks of age. This suggests that Rab4A accelerates the onset of GN. However, the deletion of Rab4A in T cells effectively blocked the development of proteinuria in B6.TC/Rab4A^{Q72L}-KO mice over B6.TC/Rab4A^{Q72L} lupus-prone mice at ages of 20-29 weeks, 40-49 weeks and > 50 weeks (**Figure S2B**). These findings indicate that Rab4A exerts a robust effect on lupus pathogenesis within T cells. Please, note that both the inactivation of Rab4A in T cells and treatment with rapamycin completely blocked GN in B6.TC/Rab4A^{Q72L} mice (**Figure 3A**). We did perform histology of spleen tissues from these mice. Our preliminary results indicate that Rab4A affects the overall splenic architecture.

Comment 4. The central hypothesis is that Rab4a overactivation promote mTOR signaling and mitochondrial metabolism, which contributes to lupus development. There are several issues on the data supporting this hypothesis. First, TC mice themselves showed increased mTORC1 activation in T cells based on their previous publications. But I don't see expansion of CD4 T cells (Fig. 2B), or increased MTG or TMRM staining (Fig. 2C), or even pS6RP staining (Fig. S4) comparing TC to B6 control. Second, there is no clear increase of pS6RP or p-AKT in CD4 T cells (Fig. 3E) and possibly a stronger increase of pS6RP and p-AKT in CD8 T cells (Fig. 3E) comparing TC/Rab4A KI to TC alone, although Fig. S4 did show increase of pS6RP in CD4 T cells from TC/Rab4A KI mice (it is unclear how they quantified pS6RP in the flow data). In fact, Fig. 3G seems to be show that TC/Rab4 KI T cells have reduced pS6RP and pAKT. However, Fig. 2 showed that CD4 T cells expanded at the expense of CD8 T cells in TC/Rab4A KI mice. Shouldn't we expect the opposite? Third, the Seahorse data in Fig. 5 and Fig.S6 are a little confusing too. If the increased OCR in TC/Rab4KI in Fig. 5 A is a consequence of increased mTORC1, shouldn't we expect to see rapamycin rectifying it? But Fig. 5B shows this is not the case. In fact, rapamycin seems to increase OCR in CD8 T cells and ECAR in CD4 T cells, which seems to be a little contradictory. Fig. 5E lacks legends. Lastly, the isotope tracing experiment is very nice and comprehensive. But again, we would expect that rapamycin can reverse the increased glutamate, a-ketoglutarate, malate, fumarate flux in TC/Rab4A KI CD4 T cells, but it did not. Also, why rapamycin can increase a-ketaglutarate and fumarate flux over TC/Rab4A KO CD4 T cells is unclear. The glucose tracing results (Fig. S6C) seem to contradict the investigator's early results that showed increased glucose metabolism in TC T cells over B6 T cells. Can the investigators provide an explanation? In fact, one would expect to see increased glycolysis in TC/Rab4A KO over TC because Tsc1 KO T cells have increased ECAR.

Response: We previously identified mTORC1 activation in T cells of patients 5, 7, 14 and mice with SLE¹⁵. mTORC1 was robustly activated in 6-month-old NWB/W(F1) mice¹⁵. This study also shows activation of mTORC1 in B6.TC mice that have a delayed onset of SLE, which is less severe than that in the parental strain¹⁶. Nevertheless, the current study has demonstrated increased mTORC1 activation in B6.TC mice over B6 controls in CD3⁺, CD4⁺, CD8⁺ and DN T cells alike, using 4-6 mice per experimental group (**Figure S5**). To clarify the method of quantification, the percentage of pS6RP⁺ cells were compared between different genotypes. As an example, pS6RP⁺ cells constituted 4.32% of CD4⁺ T cells in B6/Rab4A^{Q72L} mice and 12.3% of CD4⁺ T cells in B6.TC/Rab4A^{Q72L} mice. Thus, the prevalence of CD4⁺pS6RP⁺ T cells was 2.86-

fold greater in lupus-prone B6.TC/Rab4A^{Q72L} mice over B6/Rab4A^{Q72L} mice (**Figure S5**). Although we did not have preconceived notions about population changes, we have been working with the RAB4KI (B6/Rab4A^{Q72L}) and RAB4KO (B6/Rab4A^{Q72L}-KO) mice since 2015 and consistently seeing the expansion of CD4⁺ over CD8⁺ T cells in B6.TC/Rab4A^{Q72L} mice and the reversal of such skewing in B6.TC/Rab4A^{Q72L}-KO mice. With respect to the Seahorse data in Figure 5 (Figure 4 in the revised manuscript), the Rab4A-imposed OCR changes in CD8⁺ T cells were indeed more responsive to treatment with rapamycin *in vivo*. **Figure 4E** has had the following legend, which may have been missed by the Reviewer, “Opposite effects of Rab4A activation on basal respiration and mitochondrial ATP production between CD4⁺ and CD8⁺ T cells. Normalized values in B6.TC mice were compared to those of B6.TC/Rab4A^{Q72L} mice with two-way ANOVA.” The mechanism underlying why rapamycin can increase α -ketoglutarate flux in CD4⁺ T cells of B6.TC/Rab4A^{Q72L}-KO mice certainly warrant further investigations. Our lab has previously not reported stable isotope tracing studies of TCA or glucose metabolism in B6.TC mice. As described in the Results section of the manuscript, [M1-¹³C]-pyruvate was accumulated in CD4⁺ T cells of B6.TC/Rab4A^{Q72L}-KO mice upon [1,2-¹³C]-glucose labeling (**Figure S12C** in the revised manuscript). As measured by ECAR, glycolysis and glycolytic flux were both increased in CD4⁺ T cells of rapamycin-treated B6.TC/Rab4A^{Q72L}-KO mice (**Figure S12A** in the revised manuscript). These findings suggest that Rab4A regulates carbon flux between the non-oxidative branch of the PPP, glycolysis, and the TCA cycle in CD4⁺ T cells even when mTOR is inactivated. Regulation of CD71 expression by Rab4A could certainly play a role in controlling mitochondrial metabolism independent of mTOR pathway activation¹⁷. Therefore, we included the following statement in the Discussion: Most recently, overexpression of CD71 was associated with enhanced iron uptake into CD4⁺ lupus T cells¹⁷, which might contribute to mitochondrial dysfunction in SLE independent of mTOR pathway activation.

Comment 5. Following on the previous point, the mTORC2 activity is measured by flow cytometry of p-AKT. The pS6RP and p-AKT staining data in Fig. 6C, 6D and Fig. S8 was very weak and not convincing at all. Immunoblot of p-AKT is needed to confirm the conclusion. This also brings question on the conclusion in Fig. 3E-3G. Not sure if one can expect to see 20% of resting B6 CD4 T cells are p-AKT+. How did the authors gate? Some of the key conclusions on mTORC2 should be confirmed by immunoblot. The conclusion that kynurenine promotes mTOR activity in T cells is not convincing by the presented data.

Response: As noted above, we originally discovered mTORC1 activation in T cells of patients 4-7, 14, 18, 19 and mice with SLE^{15, 20}. These findings have been widely confirmed, cited > 1,500 times, and extended to other cell types in patients²¹⁻³⁰ and mice in SLE³¹⁻³⁶. mTORC1 activation has been shown by flow cytometry based on phosphorylation of S6RP, a downstream target of phosphorylated S6K, a substrate of mTORC1, and by western blot detection of pS6K in patients 4, 7, 14 and mice with SLE, as shown earlier¹⁵. mTORC1 was robustly activated in 6-month-old NWB/W(F1)¹⁵. This study shows activation of mTORC1 in B6.TC mice that have a delayed onset of SLE, which is somewhat less severe than the mixed NWB/W(F1) parental strains¹⁶. Nevertheless, the current study has also demonstrated increased mTORC1 activation in B6.TC mice over B6 controls in CD3⁺, CD4⁺, CD8⁺ and DN T cells alike, using 4-6 mice per experimental group (**Figure S5**).

We have appreciated the suggestion to confirm mTOR pathway activation by western blot, which has been detected by flow cytometry. We originally discovered that kynurenine stimulated mTORC1 in Jurkat T cells and primary human peripheral blood lymphocytes, using western blot and flow cytometry³⁷. However, we have noted far greater sensitivity and accuracy and less

variability of flow cytometry as compared to western blot detection of mTORC1 and mTORC2 activities 4-7, 15. This is in good agreement with a recent study that has been dedicated to comparing the sensitivity and reproducibility of mTOR pathway activation via flow cytometry and western blot. This latter study found that phospho-flow was more sensitive and showed less inter-sample variation compared with the western blotting method³⁸. Nevertheless, to address the reviewer's comment, we performed western blot analysis, demonstrating activation of mTORC1 by enhanced expression of pS6K in CD4⁺ T cells B6.TC/Rab4A^{Q72L} mice over B6.TC controls (**Figure S20x**). Dose response and time course studies detected increased pS6K/actin and pAkt/actin protein levels in splenocytes stimulated with CD3/CD28 and 1 mM kynurenine for 30 min to 4 hours in comparison to control splenocytes stimulated with CD3/CD28 alone. (**Figure S21x**). Figures S20x and S21x have been submitted for review but not necessarily meant to be included in the manuscript.

Comment 6. Fig. 3H does not seem to be consistent with Fig. 2. I don't see increased MTG+TMRM⁺ in TC/Rab4a KI CD4 T cells vs TC. But there seemed to be different in Fig. 2D. If CD4-CD8⁻ double negative T cells have the highest MTG/TMRM increase in TC/Rab4a KI mice, shouldn't we expect to see increased DN T cells in TC/Rab4a KI mice? But in fact, we see the opposite phenotypes (Fig. 2B).

Response: Indeed, the studies unveiled an expansion of CD4⁺MTG⁺TMRM⁺ T cells in B6.TC/Rab4A^{Q72L} mice in Figure 2 as opposed to Figure 3H. As noted in the Results section, mice in Figure 2 were studied at 20 weeks of age before the onset of GN and proteinuria, while those in Figure 3 were studied at 40 weeks of age to evaluate the therapeutic impact of rapamycin and NAC. To delineate this difference, the ages of the mice have been clearly stated in the legends of the revised manuscript. Moreover, we added the following statement to the Results section: As opposed to 20-week-old mice (**Figure 2C**), MTG⁺TMRM⁺ CD4⁺ T cells were not expanded in B6.TC/Rab4A^{Q72L} mice over B6.TC controls following disease onset at 40 weeks of age (**Figure 3H**). By contrast, MTG⁺TMRM⁺ DN T cells were expanded in B6.TC/Rab4A^{Q72L} mice after the onset of SLE onset at 40 weeks of age (**Figure 3H**), which may reflect similar expansions of metabolically activate DN T cells in patients with SLE 5, 6, 19.

Comment 7. Some panels of Fig. 4 may be moved to supplementary figures. They do not provide a lot of information.

Response: Following the reviewer's suggestion, we have moved Figure 4 as **Figure S11** to Supplementary Materials.

Comment 8. The authors should present Cd98 staining data for Fig. 6B. The CD98 flow data in Fig. S8 was also quite weak.

Response: We added flow data to Figure S9 which show robust induction of CD98 expression by KYN in CD4⁺ and CD8⁺ T cells with or without concurrent CD3/CD28 co-stimulation (**Figure 6B** in the revised manuscript). Although the KYN-induced shift of CD98 expression is less robust in B cells, it's unequivocal and statistically highly significant (**Figure S13**).

Comment 9. I think some of the data Fig. S7 should be moved to main figure because it is a key link to the authors' hypothesis. The serum kynurenine level is not altered in TC mice compared to B6 mice, which contradicts their previous publication. Can the authors provide an explanation? When we look at kynurenine level in T cells, it remains largely the unaltered in TC/Rab4a KI CD4

T cells, but increased in TC/Rab4a KI CD8 T cells, which was reversed by rapamycin. But if mTORC1 and mitochondrial activity are more highly elevated in TC/Rab4a KI CD4 T cells, as shown in Fig. 2, Fig. 3 and Fig. 5, shouldn't we expect kynurenine level is higher in TC/Rab4a KI CD4 T cells?

Response: Following the reviewer's recommendation, the original Figure S7 has become main Figure 5 in the revised manuscript. The differences in serum kynurenine levels between the current and earlier studies may be attributed to diet variability between institutions. As stated in the Results section of the original manuscript, the extent of CD98 expression was greater on CD8⁺ T cells that occurred with the contraction CD8⁺ T cells relative to CD4⁺ T cells upon treatment with KYN and concurrent CD3/CD28 co-stimulation (**Figure 6B**). Along these lines, the following statement has been added to the Results section: "The cell type-specific differences in KYN accumulation may be attributed the markedly elevated expression of CD98 on CD8⁺ T cells over CD4⁺ T cells upon CD3/CD28 co-stimulation (ANOVA p=0.0037; **Figure 6B**)".

Comment 10. It is challenging to read Fig. 7. Most of the panels are so small, with a lot of texts in Fig. 7A are too blurred to read. Fig. 7C showed that the alteration of CD98 and CD71 is mostly restricted to DN T cells, not CD4 and CD8 T cells, which does not fit their overall hypothesis very well. The conclusion that CD71 and CD98 are elevated in TC/Rab4a KI T cells is not well supported. Then, the CD98⁺CD71⁺ cells form a distinct population in TC/Rab4a KI DN T cells. Are these cells a distinct cell subset or an indication of cell activation (mTOR activation presumably)? For the recycling experiment in Fig. 7G and H, if Rab4a is constitutively active in TC/Rab4a KI T cells, shouldn't we see stronger internalization of CD71 and Cd98, and impaired internalization in Rab4a KO T cells (but not clear in Fig. 7G and 7H)? It is unclear how reliable the assay is.

Response: Following the reviewer's comment, the original Figure 7 was restructured and divided into two separate figures, **Figure 7** and **Figure 8**, in the revised manuscript. Both the representative flow cytometry dot plots and the cumulative statistical analyses clearly show that CD71⁺CD98⁺ cells are expanded within each T-cell subset of B6.TC/Rab4A^{Q72L} (Rab4AKI) mice, and that these changes are completely abrogated by treatment with rapamycin *in vivo*, as shown in revised Figure 8A and 8B. Our laboratory has been employing recycling assays for almost 20 years^{14, 39}. The comparison of receptor traffic patterns between experimental groups was carried out simultaneously under strictly controlled conditions, as described in the methods section.

Comment 11. Fig. 8 tried to further establish a causal link between CD98 and mTOR and lupus patients. CD98 is known to be required for optimal mTOR activity. Fig. 8A and 8B showed that CD98 expression is restricted to DN T cell, but pS6RP increase is observed in CD4⁺CD98⁺ and CD8⁺CD98⁺ T cells from lupus patients. These data are not completely consistent either. The authors' previous publication showed lupus T cells have modest increase of pS6RP. How about mTOR activity in CD98⁻ T cells? Fig. 8C aims to show reduced CD98 on human T cells can reduce pS6. But the effects are very small. CD98 increase in Fig. 8E is not very convincing either. Overall, it is not clear such a small effect to promote CD98 expression by Rab4a can explain the disease phenotypes in the TC/Rab4a KI mice.

Response: As noted in response to comment 5, our lab originally discovered mTORC1 activation in T cells of patients^{5, 7, 14} and mice with SLE¹⁵. These findings have been widely confirmed and extended to other cell types in patients²¹ and mice in SLE²⁰. mTORC1 activation has been shown by flow cytometry based on phosphorylation of S6RP, and downstream target of phosphorylated S6K, a substrate of mTORC1, and by western blot detection of pS6K in patients⁴,

7, 14 and mice with SLE¹⁵. The current study provides evidence that Rab4A-mediated changes in expression of CD98 influence mTOR activation in human and mouse T cells. Nevertheless, we presume and agree with the reviewer that the molecular bases of mTOR activation are multifactorial. As examples, branched chain amino acids, such as valine, and glutamine, also activate mTOR, which senses amino acid accumulation in the lysosome. Along these lines, the impact of Rab4A on mTOR activation is also likely to be multifactorial involving the traffic of receptors that transport metabolites as well as the traffic of mTOR itself. These complex mechanisms have been addressed in the Discussion of the original manuscript.

Comment 12. The title indicates that Rab4A controls mitophagy in lupus T cells. The supporting data are Fig. S3A and S3B. Such a weak change of Drp1 expression is not sufficient to support mitophagy is regulated by Rab4a.

Response: Given that Rab4A was found to be overexpressed in T cells of patients with SLE and lupus-prone strains before the onset of SLE, the primary goal of these studies was to test the hypothesis that Rab4A can influence the development of SLE *in vivo*. The present study clearly shows that inactivation of Rab4A in T cells blocks ANA production, proteinuria, and GN in lupus-prone mice. We do agree that the impact of Rab4A on disease development involves complex mechanisms. This study has not been initiated to prove that Rab4A controls mitophagy via depletion of Drp1, which has been previously documented in control and SLE subjects and mice¹⁵ and model cell lines⁴⁰. Here, we show that Drp1 is depleted in Rab4AKI (B6/Rab4A^{Q72L}, -28%; and B6.TC/Rab4A^{Q72L} mice; -44%). Similar changes in Drp1 may have significant impact on mitochondrial turnover via mitophagy⁴¹⁻⁴³.

Comment 13. There are several places where the texts are mistaken or wrong figure is referred to (e.g., While both CD71 and CD98 were most abundantly expressed on DN T cells, the majority of CD98+ T cells did not co-express CD71 in B6.TC mice (Figure 6C)). It should be Figure 7C.

Response: We very much appreciate the careful review and thoughtful comments. Indeed, these data were shown in Figure 7C of the original manuscript. This figure corresponds to **Figure 8A** in the revised manuscript.

REVIEWER 3 COMMENTS

Reviewer #3 SLE (Remarks to the Author):

General Comments: In this paper, Huang et al. revealed that the constitutive activation of Rab4A exerts dominant control over pro-inflammatory signal transduction networks and mechanistically *in vivo* and *in vitro*. There are many concerns in this study. There are significant issues in statistics, especially multiple comparisons. The authors should show the statistical analysis in each figure legend. The authors didn't show the p-value in many Figures.

Since previous reports revealed the importance of Rab4A in patients with SLE and lupus models, the novelty of this paper is not so significant.

Response: As described in our response to Reviewer 2, connecting bars between experimental groups without displaying exact numbers reflected $p < 0.05$, as indicated in figure legends. This was intended to avoid overcrowding. We have included a corresponding statement to the methods section of the original manuscript. However, following the recommendations by the

Editor and Reviewer 1, p values < 0.05 have been consistently shown as exact p values with 4 digits displayed after the decimal point in all figures of the revised manuscript.

With regards to novelty and significance, our earlier studies showed increased expression of Rab4A in patients and mice with SLE. By contrast, 1) this study newly demonstrates that Rab4A plays a causative role in lupus pathogenesis which is accelerated in B6.TC/Rab4A^{Q72L} mice and blocked in female B6.TC/Rab4A^{Q72L}-KO mice; and moreover, 2) this causative role is mTOR dependent, as treatment with rapamycin *in vivo* completely abrogated mTOR activation and GN in B6.TC/Rab4A^{Q72L} mice.

Comment 1. Why the authors used Rab4A Q72L mice when they made the mice using loxP systems? Is something wrong if they used B6 mice or B6 TC mice for using the loxP system? The authors should explain why they chose Q72L alleles and how the differences between B6 and Rab4A Q72L occurred.

Response. As described in the abstract, in the introduction, and in the first paragraph of the Results section, we created the floxed Rab4A^{Q72L} knock-in mice as a model for constitutive activation of Rab4A. These mice were developed to test the hypothesis that the activation of Rab4A, which was intended to mimic overexpression and increased endosome recycling in patients¹⁴ and mice¹⁵, exerted an influence on disease development, such as GN in SLE. Once such effect was observed, we deleted Rab4A in T cells by crossing the floxed B6/Rab4A^{Q72L} and B6.TC/Rab4A^{Q72L} knock-in mice with CD4Cre mice that in turn blocked the development of GN in lupus-prone B6.TC/Rab4A^{Q72L} mice.

Comment 2. In Figure S1E, how about the expression of Rab4A in CD8 T cells?

Response. We revised Figure S1E to document the deletion of Rab4A both in CD4⁺ and CD8⁺ T cells. We also stated in the revised manuscript that Cre was not expressed in peripheral CD8⁺ T cells as deletion of Rab4A had occurred at the developmental stage of double-positive T cells in the thymus, as originally described in this targeted deletion model⁴⁴.

Comment 3. In Figure 1A-D and some other figures, the significant difference or p-value should be shown.

ANA in the male are increased compared to female? Fold change in male mice should be calculated using a female mouse to compare sex.

Is the number of some groups (e.g., B6 or B6.TC) only two? Why didn't the authors use ANOVA or multiple comparisons in Figures 1A-F, 2S, and other Figures?

Response: We routinely used t-test for hypothesis testing when comparing two experimental groups. We have been using two-way ANOVA when comparing interventions between two experimental groups. For each comparison, we have included exact p values in each figure of the revised manuscript.

Comment 4. In Figure S2, there is no data in some groups.

The authors described that Non-autoimmune B6 males also had

greater proteinuria (1.01±0.14 jtg/jtl) than female controls (0.49±0.07 jtg/jtl, p=0.0299. Is this in the normal range or pathologic?

Response: Indeed, non-autoimmune B6 males had greater proteinuria (1.01±0.14 jtg/jtl) than female B6 controls at 20-29 weeks of age (0.49±0.07 jtg/jtl, p=0.0299; **Figure S2**). These results were consistent with earlier findings of sex and age-related differences⁴⁵⁻⁴⁷. As stated in the Results section of the manuscript, proteinuria was increased in young adult B6 males (1.12±0.18

jig/jil) over female B6 mice at 10-19 weeks of age (0.65 ± 0.96 jig/jil; $p=0.0395$). Proteinuria declined in B6 males to 0.54 jig/jil by 50 weeks of age ($p=0.0395$; **Figure S2A**), which was consistent with earlier findings of sex and age-related differences ⁴⁵⁻⁴⁷.

With the onset of ANA at 20-29 weeks of age (**Figure S2B**), there was a robust increase in proteinuria in B6.TC/Rab4A^{Q72L} females (1.11 jig/jil) relative to B6/Rab4A^{Q72L} (0.75 jig/jil; $p=0.04$) and B6.TC females (0.58 jig/jil; $p=0.012$) (**Figure 1D**). Inactivation of Rab4A in T cells abrogated proteinuria in female B6.TC/Rab4A^{Q72L}-KO mice (0.56 ± 0.11 jig/jil) in comparison to B6.TC/Rab4A^{Q72L} female controls (1.11 ± 0.19 jig/jil; $p=0.0178$; **Figure 1D**). Proteinuria was reduced with remarkable consistency in B6.TC/Rab4A^{Q72L}-KO females relative to B6.TC/Rab4A^{Q72L} females across all ages investigated, such as 20-29, 30-39, 40-49, and >50 weeks (**Figure S2B**).

Comment 5. Why the inactivation of Rab4A in T cells abrogated proteinuria in male B6.TC/Rab4A^{Q72L}-KO mice did not have significant differences in comparison to B6.TC/Rab4A^{Q72L} female controls?

Response: This question appears to represent an oversight, since the inactivation of Rab4A in T cells abrogated proteinuria in female but not in male B6.TC/Rab4A^{Q72L}-KO mice in comparison to B6.TC/Rab4A^{Q72L} controls. Such sex-specific differences were noted consistently in female mice at ages of 20-29 weeks, 40-49 weeks, and >50 weeks (**Figure S2B**). The abrogation of proteinuria by Rab4A deletion in T cells in female but not male mice is strongly suggestive of clinical relevance given the 9 to 10-fold greater prevalence of SLE in women over men.

Comment 6. About the sentence “Interestingly, male B6.TC/Rab4A^{Q72L}-KO mice developed severe glomerulosclerosis with greater percentage of glomeruli with sclerosis or hyalinosis relative to B6.TC mice with normal Rab4A alleles”, the authors should discuss and explain the differences between genders. Do the authors think the Rab4A target therapy is ineffective or harmful for male patients with lupus nephritis?

Response: As stated in the Results section, “given the overexpression of Rab4A in SLE patients, predisposition to GN in female B6.TC/Rab4A^{Q72L} mice, and blockade of GN upon inactivation of Rab4A in T cells, its role in immune system activation and disease pathogenesis were further investigated in female mice. Importantly, estrogen regulates the biosynthesis of geranylgeranyl isoprene units which are required for posttranslational modification of Rab GTPases, such as Rab4 ⁴⁸. While geranylgeranylation is required for binding of Rab4A to endosome membranes, pharmacological blockade of this enzymatic process inhibits the development of SLE in female mice ¹⁵. Therefore, we very much agree with the reviewer’s conclusion that Rab4A-directed therapy may have different outcomes in males relative to females. These considerations have been added to the Results section of the revised manuscript.

Comment 7. The authors should describe the statistical method in all the Figure legends. If the authors would like to have significant differences, the number of each group should be more than three.

Response: As suggested by the Reviewer, the statistical approach has been included in each figure legend. The statistical validity of having a minimum number of two independent replicates has been addressed in our response to comment 1 from Reviewer 1. This was occasionally unavoidable when attempting to breed sufficient numbers of mice for parallel testing of six genotypes matched for age and sex and providing parallel groups for treatment and solvent controls.

Comment 8. In Figure 2, the DN T cell population increases in B6.TC/Rab4A^{Q72L}-KO mice. Basically, the population of DN T cells reflects the disease activity of the lupus model. The authors should explain and discuss the discrepancy.

Response: In agreement with the reviewer, our results support the heterogeneity of DN T cells that have been considered both drivers⁴⁹ and inhibitors of renal inflammation⁵⁰. 1) DN T cells were depleted in 20-week-old B6.TC/Rab4A^{Q72L} mice before the onset of SLE, as shown in Figure 2; however, DN T cells were expanded in B6.TC/Rab4A^{Q72L} mice and depleted in B6.TC/Rab4A^{Q72L}-KO mice after the onset of SLE. 2) While rapamycin restrained DN T cells in B6.TC/Rab4A^{Q72L} mice with therapeutic efficacy, DN T cells were expanded in rapamycin-treated B6.TC/Rab4A^{Q72L}-KO mice over rapamycin-treated B6.TC/Rab4A^{Q72L} mice. These results indicate that the abundance of DN T cells is controlled by two different mechanisms in SLE: i) expansion via Rab4A-dependent mTOR activation as noted in B6.TC/Rab4A^{Q72L} mice; and ii) contraction via Rab4A-independent mTOR activation in B6.TC/Rab4A^{Q72L}-KO mice, both of which can be reversed by treatment with rapamycin. Therefore, these two types of DN T cells may play divergent roles in disease pathogenesis^{49, 50}, which warrant further investigations. These considerations have been added to the Discussion of the revised manuscript.

Comment 9. In FigureS3, the differences of Drp1 is not so significant even though the Rab4A is knocked down. Is this the main mechanism?

Response: While the impact of Rab4A on lupus pathogenesis is exerted via endosome traffic, the depletion of Drp1 is one of several mechanisms that promote lupus pathogenesis, as illustrated in **Figure S18** of the revised manuscript.

Comment 10. In the manuscript of Figure 3S, the reasons why the authors focused on Drp1, TSC1, and mLST8, should be clearly explained.

Response: As stated in the Results section, the studies shown in Figure S4 were initiated to evaluate the role of the Rab4A-mTOR positive feedback loop as a mechanism underlying the depletion of Drp1. Similar to SLE patients¹⁵, the accumulation of mitochondrial mass was attributed to Rab4A-mediated Drp1 depletion in CD4⁺ T cells of B6/Rab4A^{Q72L} (**Figure S3A**) and B6.TC/Rab4A^{Q72L} mice (**Figure S4B**). Involvement of mTOR in this positive feedback loop was supported by the increased expression of Rab4A and the depletion of Drp1 and pDrp1^{S616}, which restrain mitochondrial fission and mitophagy^{51, 52} and thus elicit the accumulation of mitochondria, in mouse embryonic fibroblasts (MEFs) lacking a tuberous sclerosis complex 1 (TSC1-/-; **Figure S4B**). Of note, patients with genetically enforced mTOR activation due to TSC mutations were found to develop fulminant SLE⁵³⁻⁵⁵, including severe nephritis^{24, 56-58}. Alternatively, Drp1 and pDrp1 were accumulated and mitochondrial mass was reduced in MEFs lacking mLST8, a component shared by mTORC1 and mTORC2 (**Figure S4C**). Along these lines, Drp1 and pDrp1 were also accumulated and mitochondrial mass was reduced in MEFs lacking mTORC2 component Rictor and mTORC1 effector 4E-BP1 (**Figure S4C**). These findings indicated the involvement of the Rab4A-mTOR positive feedback loop in the depletion of Drp1 and the accumulation of mitochondrial mass.

The activation of mTOR is regulated by tuberous sclerosis proteins 1 (TSC1) and 2 (TSC2)⁵⁹. After discovering a role for mTOR activation in lupus pathogenesis^{14, 18}, patients with genetically enforced mTOR activation due to TSC mutations were reported to develop SLE. Severe SLE with class IV nephritis has been documented in four case reports of patients with tuberous sclerosis^{24, 56-58}. Mutations of TSC1 and TSC2 have also been associated with

lymphangioliomyomatosis (LAM) and concurrent SLE⁵³⁻⁵⁵. In two of these cases, a diagnosis of SLE preceded the onset of LAM by more than a decade^{58, 60}. Thus, patients with long-standing SLE should be monitored for the development of LAM and other conditions with mTOR activation. To evaluate the role of the Rab4-mTOR positive feedback loop, we examined the impact of genetically enforced mTOR activation on expression of Rab4A and depletion of Drp1. While we refrained from expanding the Discussion, we added the following sentence to the rationale in the Results section: Of note, patients with genetically enforced mTOR activation due to TSC mutations may develop SLE⁵³⁻⁵⁵, including severe nephritis^{24, 56-58}.

Comment 11. A dot plot should be shown in some Figures (e.g., Figure 3S CD).

Response: Bart charts have been replaced with dot plots in panels C and D of Figure S4.

Comment 12.. “Rapamycin expanded DN T cells in B6.TC mice while it depleted DN T cells in B6.TC/Rab4AQ72L mice; NAC also depleted DN T cells in B6.TC/Rab4AQ72L mice.” The authors should explain the reasons.

Response: Rapamycin indeed expanded DN T cells in B6.TC mice, while both rapamycin and NAC depleted DN T cells in B6.TC/Rab4A^{Q72L} mice. Albeit the depletion by rapamycin and NAC of DN T cells in B6.TC/Rab4A^{Q72L} mice occurred with therapeutic efficacy, the expansion by rapamycin of DN T cells in B6.TC mice did not occur with disease flare. These observations also support the notion that DN T cells are heterogeneous, as discussed in the response to comment 8.

Comment 13. In Figure 5, the space capacity of the mito stress test should be shown. In Figure 5E and Figure S7, it is unclear which comparison has significant differences.

Response: Under space capacity, we have assumed that the reviewer may have meant spare respiratory capacity, which has now been included for both CD4⁺ and CD8⁺ T cells; p values < 0.05 have been included in **Figures 4B and 4D** in the revised manuscript. This analysis revealed that spare respiratory capacity was increased in CD8⁺ T cells of B6.TC/Rab4A^{Q72L} mice as compared to B6.TC controls upon rapamycin treatment *in vivo* (**Figure 4D**), which has also been included in the Results section of the revised manuscript.

Comment 14. In Figure 5G and Figure 6SC and D, if the authors revealed the results in the pathway map, it is easily understood. Again the number of some groups is only two. Why does rapamycin's effect on metabolic flux differ between CD4 and CD8 T cells?

Response: The revised schematic pathway map in **Figure 4F** of the revised manuscript illustrates that metabolic flux in the TCA cycle was increased in CD4⁺ T cells but it was reduced in CD8⁺ T cells; these changes were highlighted by red circular arrows and red TCA designation in CD4⁺ T cells and blue circular arrows and blue TCA designation in CD8⁺ T cells, respectively, within mitochondria. This description has also been added to the legend of **Figure 4**. The schematic pathway map in **Figure S12C** illustrates that Rab4A regulates carbon flux between the non-oxidative branch of the PPP and glycolysis in CD4⁺ but not in CD8⁺ T cells. The schematic pathway map in **Figure S12D** illustrates that enhanced glucose metabolism through the non-oxidative branch of the PPP supports greater proliferative capacity of CD4⁺ T cells of B6/Rab4A^{Q72L} and B6.TC/Rab4A^{Q72L} mice. Hereby, the reviewer also raised the important question why rapamycin's effect on metabolic flux differed between CD4⁺ and CD8⁺ T cells. Given the complexity of metabolic cues which are sensed and regulated by mTOR in CD4⁺ and CD8⁺ T cells⁶¹, many of which are profoundly skewed in SLE^{6, 7, 59, 62}, we added the following

succinct statement to the Discussion: The differential effect by rapamycin on metabolic flux between CD4⁺ and CD8⁺ T cells may be attributed, at least in part, to the variable reliance of these cells on glycolysis⁶³ relative to the mitochondrial TCA cycle⁶⁴, respectively.

Comment 15. In Figure S7, pyruvate is significantly reduced in B6TC/Rab4a KO compared to B6TC/Rab4a. On the other hand, the glycolysis stress test does not have significant differences. Why this discrepancy occurs? How about lactate in this setting?

Response: This comment may have reflected an oversight since pyruvate was not shown in Figure S8. We assume the reviewer meant Figure S6 (Figure S12 in the revised manuscript), which presents functional glycolysis data. As stated in the Results section, upon [1,2-¹³C]-glucose labeling, [M1-¹³C]-pyruvate was accumulated in CD4⁺ T cells but not in CD8⁺ T cells of B6.TC/Rab4A^{Q72L}-KO mice (**Figure S12C**). This suggests that Rab4A regulates carbon flux between the non-oxidative branch of the PPP and glycolysis in CD4⁺ but not in CD8⁺ T cells. [M1/M3-¹³C]-lactate could not be effectively measured in [1,2-¹³C]-glucose labeled CD4⁺ and CD8⁺ T cells. This was consistent with the extrusion of lactate from glycolytic cells^{65, 66} which allows for the assessment of glycolysis via measurement of extracellular acidification^{25, 65, 66}. We are aware of the limitations of individual methodologies for assessment of metabolism, and therefore, we carried out independent measurements from the same samples, using flow cytometry, Seahorse functional assays, and LC/MS-MS tracing of metabolites from isolated CD4⁺ and CD8⁺ T cells labeled with stable isotopes.

REFERENCES

1. Crispin, J.C. & Tsokos, G.C. Human TCR- $\alpha\beta$ CD4⁻ CD8⁻ T cells can derive from CD8⁺ T cells and display an inflammatory effector phenotype. *J. Immunol.* **183**, 4675-4681 (2009).
2. Hedrich, C.M. *et al.* cAMP responsive element modulator (CREM) α mediates chromatin remodeling of CD8 during the generation of CD3⁺CD4⁻CD8⁻ T cells. *J. Biol. Chem.* **289**, 2361-2370 (2013).
3. Rodriguez-Rodriguez, N. *et al.* TCR- $\alpha\beta$ CD4⁻CD8⁻ double negative T cells arise from CD8⁺ T cells. *J. Leukoc. Biol.* **108**, 851-857 (2020).
4. Kato, H. & Perl, A. mTORC1 expands Th17 and IL-4⁺ DN T cells and contracts Tregs in SLE. *J. Immunol.* **192**, 4134-4144 (2014).

0. Lai,Z.-W. *et al.* mTOR activation triggers IL-4 production and necrotic death of double-negative T cells in patients with systemic lupus erythematosus. *J. Immunol.* **191**, 2236-2246 (2013).
1. Lai,Z. *et al.* Sirolimus in patients with clinically active systemic lupus erythematosus resistant to, or intolerant of, conventional medications: a single-arm, open-label, phase 1/2 trial. *Lancet* **391**, 1186-1196 (2018).
2. Kato,H. & Perl,A. The IL-21-mTOR axis blocks Treg differentiation and function by suppression of autophagy in patients with systemic lupus erythematosus. *Arthritis Rheumatol.* **70**, 427-438 (2018).
3. Saito,Y. & Soga,T. Amino acid transporters as emerging therapeutic targets in cancer. *Cancer Sci.* **112**, 2958-2965 (2021).
4. Scalise,M., Galluccio,M., Console,L., Pochini,L., & Indiveri,C. The Human SLC7A5 (LAT1): The Intriguing Histidine/Large Neutral Amino Acid Transporter and Its Relevance to Human Health. *Front. Chem.* **6**, 243 (2018).
5. Hagopian,L.P. The consecutive controlled case series: Design, data-analytics, and reporting methods supporting the study of generality. *J. Appl. Behav. Anal.* **53**, 596-619 (2020).
6. der Heijde,D.v. *et al.* Effect of different imputation approaches on the evaluation of radiographic progression in patients with psoriatic arthritis: results of the RAPID-PsA 24-week phase III double-blind randomised placebo-controlled study of certolizumab pegol. *Ann. Rheum. Dis.* **73**, 233-237 (2014).
7. Dickson,M. & Baird,D. Significance Testing in *Philosophy of Statistics Handbook of the Philosophy of Science* (eds. Bandyopadhyay,P.S. & Forster,M.R.) 199229 (Elsevier North-Holland, Amsterdam, 2011).
8. Grossman,J. The Likelihood Principle in *Philosophy of Statistics Handbook of the Philosophy of Science* (eds. Bandyopadhyay,P.S. & Forster,M.R.) 553-580 (Elsevier North-Holland, Amsterdam, 2011).
9. Fernandez,D.R. *et al.* Activation of mTOR controls the loss of TCR. in lupus T cells through HRES-1/Rab4-regulated lysosomal degradation. *J. Immunol.* **182**, 20632073 (2009).
10. Caza,T.N. *et al.* HRES-1/RAB4-Mediated Depletion of DRP1 Impairs Mitochondrial Homeostasis and Represents a Target for Treatment in SLE. *Ann. Rheum. Dis.* **73**, 1887-1897 (2014).
11. Morel,L. *et al.* Genetic reconstitution of systemic lupus erythematosus immunopathology with polycongenic murine strains. *Proc. Natl. Acad. Sci. USA* **97**, 6670-6675 (2000).

12. Voss,K. *et al.* Elevated transferrin receptor impairs T cell metabolism and function in systemic lupus erythematosus. *Science Immunology* **8**, eabq0178 .
13. Fernandez,D., Bonilla,E., Mirza,N., Niland,B., & Perl,A. Rapamycin reduces disease activity and normalizes T-cell activation-induced calcium fluxing in patients with systemic lupus erythematosus. *Arthritis Rheum.* **54**, 2983-2988 (2006).
14. Lai,Z.-W. *et al.* N-acetylcysteine reduces disease activity by blocking mTOR in T cells of lupus patients. *Arthritis Rheum.* **64**, 2937-2946 (2012).
15. Oaks,Z. *et al.* Mitochondrial dysfunction in the liver and antiphospholipid antibody production precede disease onset and respond to rapamycin in lupus-prone mice. *Arthritis Rheumatol.* **68**, 2728-2739 (2016).
16. Mao,Z. *et al.* Renal mTORC1 activation is associated with disease activity and prognosis in lupus nephritis. *Rheumatology* **61**, 3830-3840 (2022).
17. Eriksson,P., Wallin,P., & Sjowall,C. Clinical experience of sirolimus regarding efficacy and safety in systemic lupus erythematosus. *Front. Pharmacol.* **10**, (2019).
18. Yap,D.Y.H. *et al.* Longterm data on sirolimus treatment in patients with lupus nephritis. *J. Rheumatol.* **45**, 1663-1670 (2018).
19. Okita,Y., Yoshimura,M., Katada,Y., Saeki,Y., & Ohshima,S. A mechanistic target of rapamycin inhibitor, everolimus safely ameliorated lupus nephritis in a patient complicated with tuberous sclerosis. *Mod. Rheumatol. Case Rep.* **7**, 47-51 (2023).
20. Liu,X. *et al.* Glomerular mTORC1 activation was associated with podocytes to endothelial cells communication in lupus nephritis. *Lupus Sci. Med.* **10**, e00089 (2023).
21. Ding,Y., Luan,Z.Q., Mao,Z.M., Qu,Z., & Yu,F. Association between glomerular mTORC1 activation and crescents formation in lupus nephritis patients. *Clin. Immunol.* **249**, 109288 (2023).
22. Jiang,N. *et al.* Sirolimus versus tacrolimus for systemic lupus erythematosus treatment: Results from a real-world CSTAR cohort study. *Lupus Sci. Med.* **9**, e000617 (2022).
23. Murayama,G. *et al.* Inhibition of mTOR suppresses IFN α production and the STING pathway in monocytes from systemic lupus erythematosus patients. *Rheumatology* **59**, 2992-3002 (2020).
24. Peng,L. *et al.* Clinical efficacy and safety of sirolimus in systemic lupus erythematosus: a real-world study and meta-analysis. *Ther. Adv. Musculoskelet. Dis.* **12**, 1759720X20953336 (2020).
25. Wu,C. *et al.* Lupus-associated atypical memory B cells are mTORC1-hyperactivated and functionally dysregulated. *Ann. Rheum. Dis.* **78**, 1090-1100 (2019).

26. Chen,S. *et al.* Downregulation of miR-633 activated AKT/mTOR pathway by targeting AKT1 in lupus CD4 + T cells. *Lupus* **28**, 510-519 (2019).
27. Wei,S. *et al.* Allogeneic adipose-derived stem cells suppress mTORC1 pathway in a murine model of systemic lupus erythematosus. *Lupus* **28**, 199-209 (2019).
28. Katsuyama,T., Li,H., Comte,D., Tsokos,G.C., & Moulton,V.R. Splicing factor SRSF1 controls T cell hyperactivity and systemic autoimmunity. *J. Clin. Invest.* **129**, 5411-5423 (2019).
29. Zhang,D. *et al.* Regulating T Cell Population Alleviates SLE by Inhibiting mTORC1/C2 in MRL/lpr Mice. *Front Pharmacol* **11**, 579298 (2021).
30. Gao,Y.F., Fan,X.Z., Li,R.S., & Zhou,X.S. Rapamycin relieves lupus nephritis by regulating TIM-3 and CD4+CD25+Foxp3+ Treg cells in an MRL/lpr mouse model. *Cent Eur J Immunol* **47**, 206-217 (2022).
31. Zhang,J. *et al.* Rapamycin-encapsulated costimulatory ICOS/CD40L-bispecific nanoparticles restrict pathogenic helper T-B-cell interactions while in situ suppressing mTOR for lupus treatment. *Biomaterials* **289**, 121766 (2022).
32. Perl,A. *et al.* Comprehensive metabolome analyses reveal N-acetylcysteine-responsive accumulation of kynurenine in systemic lupus erythematosus: implications for activation of the mechanistic target of rapamycin. *Metabolomics* **11**, 1157-1174 (2015).
33. Rip,J., de Bruijn,M.J.W., Kaptein,A., Hendriks,R.W., & Corneth,O.B.J. Phosphoflow Protocol for Signaling Studies in Human and Murine B Cell Subpopulations. *J. Immunol.* **204**, 2852-2863 (2020).
34. Nagy,G. *et al.* Regulation of CD4 Expression via Recycling by HRES-1/RAB4 Controls Susceptibility to HIV Infection. *J. Biol. Chem.* **281**, 34574-34591 (2006).
35. Talaber,G. *et al.* HRES-1/Rab4 promotes the formation of LC3⁺ autophagosomes and the accumulation of mitochondria during autophagy. *PLoS ONE* **9**, e84392. (2014).
36. Gomes,L.C., Benedetto,G.D., & Scorrano,L. During autophagy mitochondria elongate, are spared from degradation and sustain cell viability. *Nat Cell Biol* **13**, 589-598 (2011).
37. Kim,Y.M. *et al.* Redox Regulation of Mitochondrial Fission Protein Drp1 by Protein Disulfide Isomerase Limits Endothelial Senescence. *Cell Rep.* **23**, 3565-3578 (2018).
38. Xie,L. *et al.* Drp1-dependent remodeling of mitochondrial morphology triggered by EBV-LMP1 increases cisplatin resistance. *Sig. Transduct. Target. Ther.* **5**, 56 (2020).
39. Lee,P.P. *et al.* A Critical Role for Dnmt1 and DNA Methylation in T Cell Development, Function, and Survival. *Immunity* **15**, 763-774 (2001).

40. Wicks,L.F. Sex and Proteinuria of Mice. *Proc. Soc. Exp. Biol. Med.* **48**, 395-400 (1941).
41. Sandberg,K. Mechanisms underlying sex differences in progressive renal disease. *Gender Med.* **5**, 10-23 (2008).
42. Barsha,G., Denton,K.M., & Mirabito Colafella,K.M. Sex- and age-related differences in arterial pressure and albuminuria in mice. *Biol. Sex Differ.* **7**, 57 (2016).
43. Bruscalupi,G., Cicuzza,S., Allen,C.M., Di Croce,L., & Trentalance,A. Estrogen Stimulates Intracellular Traffic in the Liver of *Rana esculenta* complexby Modifying Rab Protein Content. *Biochem. Biophys. Res. Commun.* **251**, 301-306 (1998).
44. Kyttaris,V.C., Zhang,Z., Kuchroo,V.K., Oukka,M., & Tsokos,G.C. Cutting Edge: IL-23 Receptor Deficiency Prevents the Development of Lupus Nephritis in C57BL/6-lpr/lpr Mice. *J Immunol* **184**, 4605-4609 (2010).
45. Sadasivam,M. *et al.* Activation and Proliferation of PD-1(+) Kidney Double-Negative T Cells Is Dependent on Nonclassical MHC Proteins and IL-2. *J Am Soc Nephrol* **30**, 277-292 (2019).
46. Taguchi,N., Ishihara,N., Jofuku,A., Oka,T., & Mihara,K. Mitotic Phosphorylation of Dynamin-related GTPase Drp1 Participates in Mitochondrial Fission. *J. Biol. Chem.* **282**, 11521-11529 (2007).
47. Ravirajan,C.T., Rowse,L., MacGowan,J.R., & Isenberg,D.A. An analysis of clinical disease activity and nephritis-associated serum autoantibody profiles in patients with systematic lupus erythematosus: A cross-sectional study. *Rheumatology* **40**, 1405-1412 (2001).
48. Sam,R., Khalid,S., Brecklin,C., Schwartz,M., & Dunea,G. A case of lymphangioliomyomatosis with membranous nephropathy and likely systemic lupus. *Clin. Exp. Nephrol.* **13**, 166-169 (2009).
49. Futami,S. *et al.* Comorbid connective tissue diseases and autoantibodies in lymphangioliomyomatosis: a retrospective cohort study. *Orphanet J Rare Dis* **13**, 182 (2018).
50. Cui,H. *et al.* The etiology of diffuse cystic lung diseases: an analysis of 1010 consecutive cases in a LAM clinic. *Orphanet J Rare Dis* **16**, 273 (2021).
51. Singh,N., Birkenbach,M., Caza,T., Perl,A., & Cohen,P.L. Tuberos sclerosi and fulminant lupus in a young woman. *J. Clin. Rheumatol.* **19**, 134-137 (2013).
52. Carrasco Cubero,C., Bejarano Moguel,V., Fernandez Gil,M.A., & Alvarez Vega,J.L. Coincidence of tuberos sclerosi and systemic lupus erythematosus - a case report. *Rheumatol. Clin.* **12**, 219-222 (2016).

53. Olde Bekkink,M., hmed-Ousenkova,Y.M., Netea,M.G., Van Der Velden,W.J., & Berden,J.H. Coexistence of systemic lupus erythematosus, tuberous sclerosis and aggressive natural killer-cell leukaemia: Coincidence or correlated? *Lupus* **25**, 766-771 (2015).
54. Perl,A. Mechanistic Target of Rapamycin Pathway Activation in Rheumatic Diseases. *Nat. Rev. Rheumatol.* **12**, 169-182 (2016).
55. Verma,V., Paek,A.R., Choi,B.K., Hong,E.K., & You,H.J. Loss of zinc-finger protein 143 contributes to tumour progression by interleukin-8-CXCR axis in colon cancer. *J. Cell. Mol. Med.* **0**, 1-11 (2019).
56. Powell,J.D., Pollizzi,K.N., Heikamp,E.B., & Horton,M.R. Regulation of Immune Responses by mTOR. *Ann. Rev. Immunol.* **30**, 39-68 (2012).
57. Yin,Y. *et al.* Normalization of CD4+ T cell metabolism reverses lupus. *Sci. Transl. Med.* **7**, 274ra18 (2015).
58. Macintyre,A.N. *et al.* The Glucose Transporter Glut1 Is Selectively Essential for CD4 T Cell Activation and Effector Function. *Cell Metab.* **20**, 61-72 (2014).
59. van der Windt,G.J.W. *et al.* Mitochondrial respiratory capacity is a critical regulator of CD8+ T cell memory development. *Immunity* **36**, 68-78 (2012).
60. de la Cruz-Lopez,K., Castro-Munoz,L.J., Reyes-Hernandez,D.O., Garcia-Carranca,A., & Manzo-Merino,J. Lactate in the Regulation of Tumor Microenvironment and Therapeutic Approaches. *Front. Oncol.* **9**, 1143 (2019).
61. Tu,V.Y., Ayari,A., & O'Connor,R.S. Beyond the Lactate Paradox: How Lactate and Acidity Impact T Cell Therapies against Cancer. *Antibodies* **10**, 25 (2021).

REVIEWER COMMENTS

Reviewer #1 (Remarks to the Author):

My main concerns have been well addressed. There is only one minor error to be corrected : CD4-CD8- double-negative Thymocytes but not DNT cells precede DP or SP thymocytes in T cell development. In most literatures, DNT cells are defined as TCR $\alpha\beta$ +CD4-CD8- mature T lymphocytes and most DN thymocytes even did not complete the assembly of CD3-TCR complex so they are not equivalent.

Reviewer #2 (Remarks to the Author):

I would like to commend the authors for the herculean effort to generate such a very large body of data and the effort to address my concerns. While many of my concerns have been addressed, I still have reservations on some of the issues. Regarding the comment 5, I would like to clarify a few issues. I do not question the observation that mTORC1 is elevated in T cells from SLE patient and lupus mouse models (which has been replicated by other groups), nor do I question the association between Rab4a constitutive activation and mTORC1 activation. I also did not try to write off phosflow technique as unreliable because I think the phosflow data in Figure S5 were largely convincing. Thus, the authors' conclusions that mTORC1 overactivation is strongly associated with Rab4a mutant and could be a contribution to the ensuing lupus disease are well supported by their extensive data. However, this does not mean all the phosflow data are equally convincing. As someone who has some experience with flow cytometry, the data in the current Fig. 6C-D do not look particularly convincing. The baselines in Fig. 6C/6D were mostly very low. The increase of p-S6 or p-Akt were from ~380 to ~430 or from ~70 to ~110 with minuscule shift of the staining peak (whereas in Fig S5, you could see clear increase of pS6+ cells in B6.TC/Rab4A KI T cells in the flow plots). Fig. 6D also seems to show kynurenine slightly increases pS6/pAKT double positive cells while reduces pS6 single positive cells. Are these pS6/AKT double positive events real? Again the shift was too small to be comfortable. The new immunoblot data in Figure X21x did not help much either. The increases of pS6K or pAkt were not very clear. Finally, kynurenine level was not elevated in the KI CD4 T cells (Fig.5B). The authors cited a reference that demonstrated the superiority of phosflow over immunoblot. I was not able to

find the reference. If the authors can point out which reference it is, that will be appreciated. Taken together, my reservation remains on the link between kynurenine and mTOR activation in CD4 T cells.

Reviewer #3 (Remarks to the Author):

The authors did not correct statistical errors.

There are still significant issues in statistics for multiple comparisons.

October 15, 2023

This letter is in response to reviews received on September 25, 2023 with follow-up comments on our study entitled "Rab4A-directed endosome traffic shapes pro-inflammatory mitochondrial metabolism in T cells via mitophagy, CD98 expression, and kynurenine-sensitive mTOR activation". We have appreciated that the reviewers found the manuscript improved, while requested to address remaining important points. As summarized in the invitation for submitting a revised version, reviewer #1 was satisfied apart from some textual edits. We have addressed the comments from reviewer #2 about phosphoflow data and the link between kynurenine and mTOR in CD4 T cells and used multiple comparison statistical tests where appropriate, as commented by reviewer #3. Changes made in the manuscript in response to the reviewers' comments are highlighted in yellow in a second copy of the revised manuscript. We provided point-by-point responses to the reviewers' follow-up comments below:

Reviewer #1 (Remarks to the Author):

Comments: My main concerns have been well addressed. There is only one minor error to be corrected : CD4-CD8- double-negative Thymocytes but not DNT cells precede DP or SP thymocytes in T cell development. In most literatures, DNT cells are defined as TCR $\alpha\beta$ +CD4-CD8- mature T lymphocytes and most DN thymocytes even did not complete the assembly of CD3-TCR complex so they are not equivalent.

Response: In agreement with the reviewer, we used the term DN thymocytes rather than DN T cells in experiments involving the thymus.

Reviewer #2 (Remarks to the Author):

Comments: I would like to commend the authors for the herculean effort to generate such a very large body of data and the effort to address my concerns. While many of my concerns have been addressed, I still have reservations on some of the issues. Regarding the comment 5, I would like to clarify a few issues. I do not question the observation that mTORC1 is elevated in T cells from SLE patient and lupus mouse models (which has been replicated by other groups), nor do I question the association between Rab4a constitutive activation and mTORC1 activation. I also did not try to write off phosflow technique as unreliable because I think the phosflow data in Figure S5 were largely convincing. Thus, the authors' conclusions that mTORC1 overactivation is strongly associated with Rab4a mutant and could be a contribution to the ensuing lupus disease are well supported by their extensive data. However, this does not mean all the phosflow data are equally convincing. As someone who has some experience with flow cytometry, the data in the current Fig. 6C-D do not look particularly convincing. The baselines in Fig. 6C/6D were mostly very low. The increase of p-S6 or p-Akt were from ~380 to ~430 or from ~70 to ~110 with minuscule shift of the staining peak (whereas in Fig S5, you could see clear increase of pS6+ cells in B6.TC/Rab4A KI T cells in the flow plots). Fig. 6D also seems to show kynurenine slightly increases pS6/pAKT double positive cells while reduces pS6 single positive cells. Are these pS6/AKT double positive events real? Again the shift was too small to be comfortable. The new immunoblot data in Figure X21x did not help much either. The increases of pS6K or pAkt were not very clear. Finally, kynurenine level was not elevated in the KI CD4 T cells (Fig.5B).

The authors cited a reference that demonstrated the superiority of phosflow over immunoblot. I was not able to find the reference. If the authors can point out which reference it is, that will be appreciated. Taken together, my reservation remains on the link between kynurenine and mTOR activation in CD4 T cells.

Response: We thank the reviewer for acknowledgment of our work and the reproduction of our discoveries by others. While the reviewer was convinced by the robustness of our findings in Figure S5, the extent of changes in Figure panels 6C-D were found less convincing. Although the changes were indeed less robust, they were reproducible and statistically significant. As noted in our previous response, we originally discovered that kynurenine stimulated mTORC1 in Jurkat CD4 T cells and primary human peripheral blood lymphocytes, using western blot and flow cytometry¹. Moreover, we have noted far greater sensitivity and accuracy and less variability of flow cytometry as compared to western blot detection of mTORC1 and mTORC2 activities²⁻⁶. This is in agreement with a recent study that has been dedicated to comparing the sensitivity and reproducibility of mTOR pathway activation via flow cytometry and western blot. This latter study found that phospho-flow was more sensitive and showed less inter-sample variation as compared to the western blotting method⁷, which was included in the revised Methods section. The complete reference is also cited below:

Rip,J., de Bruijn,M.J.W., Kaptein,A., Hendriks,R.W., & Corneth,O.B.J. Phosphoflow Protocol for Signaling Studies in Human and Murine B Cell Subpopulations. *J. Immunol.* **204**, 2852-2863 (2020).

Reviewer #3 (Remarks to the Author):

Comments: The authors did not correct statistical errors. There are still significant issues in statistics for multiple comparisons.

Response: We have extensively revised the statistical approach by displaying exact p values in each figure panel throughout the manuscript. As stated in the statistical analysis section of the revised manuscript, changes were considered significant at p value < 0.05 for hypothesis testing. Exact p values with 4 digits after the decimal point over the connecting bars are displayed between groups compared. We have corrected raw p values for multiple comparison when appropriate, displaying false discovery rate (FDR) p values in Figures S12 and S14.

Source data have been provided in Excel files for each figure in multiple labelled files within a zipped folder named "Source Data". The "Data Availability" section includes the statement "Source data are provided with this paper." Moreover, the RNAseq dataset has been submitted to NCBI Gene Expression Omnibus (GEO) database under Accession #GSE245413. According to OMNIBUS, the RNAseq dataset GSE245413 will be released to the public immediately upon publication (or 6 months from submission on June 1, 2024).

Reference List

1. Perl,A. *et al.* Comprehensive metabolome analyses reveal N-acetylcysteine-responsive accumulation of kynurenine in systemic lupus erythematosus: implications for activation of the mechanistic target of rapamycin. *Metabolomics* **11**, 1157-1174 (2015).
2. Caza,T.N. *et al.* HRES-1/RAB4-Mediated Depletion of DRP1 Impairs Mitochondrial Homeostasis and Represents a Target for Treatment in SLE. *Ann. Rheum. Dis.* **73**, 1887-1897 (2014).
3. Lai,Z.-W. *et al.* mTOR activation triggers IL-4 production and necrotic death of double-negative T cells in patients with systemic lupus erythematosus. *J. Immunol.* **191**, 2236-2246 (2013).
4. Kato,H. & Perl,A. MTORC1 expands Th17 and IL-4⁺ DN T cells and contracts Tregs in SLE. *J. Immunol.* **192**, 4134-4144 (2014).
5. Lai,Z. *et al.* Sirolimus in patients with clinically active systemic lupus erythematosus resistant to, or intolerant of, conventional medications: a single-arm, open-label, phase 1/2 trial. *Lancet* **391**, 1186-1196 (2018).
6. Kato,H. & Perl,A. The IL-21-mTOR axis blocks Treg differentiation and function by suppression of autophagy in patients with systemic lupus erythematosus. *Arthritis Rheumatol.* **70**, 427-438 (2018).
7. Rip,J., de Bruijn,M.J.W., Kaptein,A., Hendriks,R.W., & Corneth,O.B.J. Phosphoflow Protocol for Signaling Studies in Human and Murine B Cell Subpopulations. *J. Immunol.* **204**, 2852-2863 (2020).

REVIEWER COMMENTS

Reviewer #2 (only confidential comments to editor)

Reviewer #3 (Remarks to the Author):

Why did the authors perform multiple analyses only in Figures S12 and S14?

Why didn't they use multiple analyses in Figures 1-6, 8 and 9?

The authors should show the statistical analysis in each figure legend.

RESPONSE TO REVIEWERS

Thank you for your letter of November 7, 2023, inviting another revision of our paper entitled "Rab4A-directed endosome traffic shapes pro-inflammatory mitochondrial metabolism in T cells via mitophagy, CD98 expression, and kynurenine-sensitive mTOR activation".

Reviewer #1: no further comment.

Reviewer #2 (only confidential comments to editor)

Response:

As you indicated, in private comments to the editor, reviewer #2 requested softening of the link between kynurenine and mTOR. Accordingly, we have softened the link between kynurenine and mTOR throughout the manuscript, as follows:

- a. Introduction: "KYN may spread inflammation through the bloodstream by eliciting mTOR activation."
- b. Discussion: "These findings identify CD98-dependent accumulation of KYN, as a pro-inflammatory metabolite that may contribute to Rab4A/mTOR-driven autoimmunity in SLE." ... "KYN is identified as pro-inflammatory metabolite that may transmit activation signals from CD8⁺ T cells to CD19⁺ and CD19⁺CD38⁺ B cells and CD138⁺ plasma cells."
- c. Limitations of the study: "We identified increased production of KYN as a central metabolite that may spread mTOR activation and inflammation through the bloodstream."

Reviewer #3 (Remarks to the Author):

Why did the authors perform multiple analyses only in Figures S12 and S14?

Why didn't they use multiple analyses in Figures 1-6, 8 and 9?

The authors should show the statistical analysis in each figure legend.

Response: As suggested by Reviewer #3 and further specified by the Editor, we provided multiple comparison statistical tests, ANOVA and post-hoc test p values. We performed 1-way, 2-way, or 3-way ANOVA in every figure when appropriate (Figures 1-8 and Supplemental Figures 2-17), i.e., to systematically examine the influence of two independent categorical variables, e.g. 1) lupus by comparing B6 control and B6.TC SLE strains; and 2) Rab4A by comparing Rab4 WT, Rab4A^{Q72L}, and Rab4A^{Q72L}-KO mice within each background strain. We performed 3-way ANOVA to address the impact of three independent variables in Figure 6B: 1) CD4 versus CD8; 2) Control versus KYN stimulation; and 3) Control versus CD3/CD28 co-stimulation. When no such independent variables were available, 1-way ANOVA was performed. Post-hoc tests were performed and exact p values have been provided with 4-digit accuracy beyond the decimal point, as calculated by GraphPad version No. 10. These calculations have been included in the source data files for each panel of Figures 1-6 and 8 and Supplemental Figures S2-S17. Importantly, all panels of the source data files contain i) raw data to allow readers to perform statistical tests of their choosing and ii) results of analyses with 1-way, 2-way, or 3-way ANOVA as appropriate for a given dataset, as well as primary and post-hoc test exact p values for all figures. To reflect these changes, we included the following statement in the statistical paragraph of the Methods section in the revised manuscript: "Figures

and all panels of source data files include raw data and 4-digit exact p values obtained by ANOVA. Two-way ANOVA was used when comparing WT, Rab4A^{Q72L} and Rab4A^{Q72L}-KO mice between control B6 and lupus-prone B6.TC strains. One-way ANOVA was used when comparing WT, Rab4A^{Q72L} and Rab4A^{Q72L}-KO mice within control B6 or lupus-prone B6.TC strains. Three-way ANOVA was performed to address the impact of three independent variables in Figure 6B: 1) CD4 versus CD8; 2) Control versus KYN stimulation; and 3) Control versus CD3/CD28 co-stimulation. Post-hoc test p values displayed in figures have been corrected for multiple comparisons via the recommended Tukey or Sidak methods¹³³⁻¹³⁶ in GraphPad Prism Version 10. All the multiple comparisons tests offered by Prism are valid even if the overall ANOVA did not find a significant difference among means. These tests are more focused, so have power to find differences between groups even when the overall ANOVA is not significant¹³⁷.” As recommended by Reviewer #3, we have also added a description of statistical analysis in each figure legend.

Changes made in the manuscript in response to the reviewers’ comments are highlighted in yellow in a second copy of the revised manuscript.

As requested, revised source data have been provided in a single Excel file that contains all panels for all figures in separate worksheets. The “Data Availability” section includes the statement “Source data are provided with this paper.” Moreover, the RNAseq dataset has been submitted to NCBI Gene Expression Omnibus (GEO) database under Accession #GSE245413. According to OMNIBUS, the RNAseq dataset GSE245413 will be released to the public immediately upon publication (or 6 months from submission on June 1, 2024).

REVIEWERS' COMMENTS

Reviewer #3 (Remarks to the Author):

I think the authors have adequately addressed the reviewers' comments in the manuscript's revised version. Therefore, I have no further comments.